# JarvisArt: Liberating Human Artistic Creativity via an Intelligent Photo Retouching Agent

**Yunlong Lin**[1*]   **Zixu Lin**[1*]   **Kunjie Lin**[1*]   **Jinbin Bai**[5]   **Panwang Pan**[4]   **Chenxin Li**[3]
**Haoyu Chen**[2]   **Zhongdao Wang**[6]   **Xinghao Ding**[1†]   **Wenbo Li**[3♣]   **Shuicheng Yan**[5♠]

[1] Key Laboratory of Multimedia Trusted Perception and Efficient Computing,
Ministry of Education of China, Xiamen University, Xiamen, Fujian, China
[2] The Hong Kong University of Science and Technology (Guangzhou)
[3] The Chinese University of Hong Kong     [4] Bytedance
[5] National University of Singapore     [6] Tsinghua University
Project Page:  https://jarvisart.vercel.app/

## Abstract

Photo retouching has become integral to contemporary visual storytelling, enabling users to capture aesthetics and express creativity. While professional tools such as Adobe Lightroom offer powerful capabilities, they demand substantial expertise and manual effort. In contrast, existing AI-based solutions provide automation but often suffer from limited adjustability and poor generalization, failing to meet diverse and personalized editing needs. To bridge this gap, we introduce Jarvis-Art, a multi-modal large language model (MLLM)-driven agent that understands user intent, mimics the reasoning process of professional artists, and intelligently coordinates over 200 retouching tools within Lightroom. JarvisArt undergoes a two-stage training process: an initial Chain-of-Thought supervised fine-tuning to establish basic reasoning and tool-use skills, followed by Group Relative Policy Optimization for Retouching (GRPO-R) to further enhance its decision-making and tool proficiency. We also propose the Agent-to-Lightroom Protocol to facilitate seamless integration with Lightroom. To evaluate performance, we develop MMArt-Bench, a novel benchmark constructed from real-world user edits. Jarvis-Art demonstrates user-friendly interaction, superior generalization, and fine-grained control over both global and local adjustments, paving a new avenue for intelligent photo retouching. Notably, it outperforms GPT-4o with a **60%** improvement in average pixel-level metrics on MMArt-Bench for content fidelity, while maintaining comparable instruction-following capabilities.

## 1 Introduction

Photo retouching is fundamental to modern photography, enabling users to manipulate exposure, color, contrast, and tone for expressive, high-quality images. Commercial tools such as Adobe Lightroom and PicsArt offer extensive manual controls but demand specialized expertise and significant time investment, creating barriers for non-experts. Existing automated methods—including zero- and first-order optimization [15, 44, 54, 65], reinforcement learning [55, 29, 28], and diffusion-based editing [68, 2, 57]—improve automation yet remain limited in stylistic diversity, fine-grained adjustment, and scene generalization. More recently, instruction-guided multimodal models such as GPT-4o [24] and Gemini-2-Flash [52] have enabled natural-language–driven editing but frequently compromise content fidelity, intricate attribute control, and high-resolution support.

---

* Equal Contributions.     ♣ Project Leader     † Corresponding Author.     ♠ Supervised the research.

39th Conference on Neural Information Processing Systems (NeurIPS 2025).

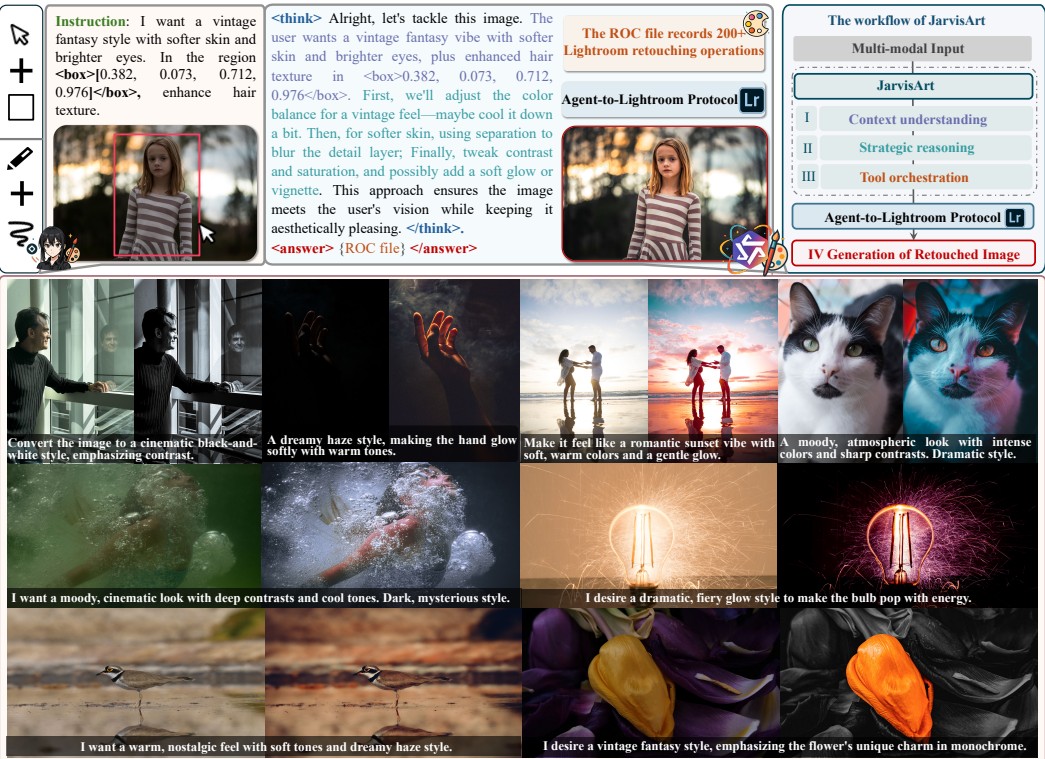

Figure 1: JarvisArt supports multi-granularity retouching goals, ranging from scene-level adjustments to region-specific refinements. Users can perform intuitive, free-form edits through natural inputs such as text prompts, bounding boxes, or brushstrokes. Furthermore, users can edit any-resolution images with JarvisArt. Purple: multi-modal context understanding. Green: retouching strategy reasoning. Orange: decision-making in tool orchestration.

LLM [13, 4, 60, 33]-powered agents have driven breakthroughs in autonomous task execution and problem solving, inspiring us to explore a novel photo-retouching paradigm: *an intelligent, user-friendly artist agent that interprets the user's intent and delivers professional-level edits*. To this end, we introduce JarvisArt, which (1) accurately parses visual inputs and natural-language instructions, (2) embeds professional retouching expertise to emulate an artist's reasoning, (3) efficiently manages over 200 Lightroom operations, and (4) supports both global and local adjustments through an intuitive interface. All planning and tool invocations are fully transparent, allowing users to interactively refine the retouching workflow to suit their individual preferences.

To translate this vision into practice, we must tackle three core challenges:the scarcity of high-quality data (source/target images, textual instructions, and editing configurations), the need for expert-level reasoning strategies, and the absence of a standardized Agent-to-Lightroom integration protocol. To overcome these, we first design a data-generation pipeline that yields the MMArt-55K dataset, comprising 5K standard and 50K Chain-of-Thought–enhanced multi-granularity samples. Next, we employ a two-stage post-training regime: (1) supervised fine-tuning (SFT) to instill a workflow of "understanding → reasoning → decision-making (recording Lightroom operations into a ROC file)", as illustrated in Figure 1, and (2) Group Relative Policy Optimization for Retouching (GRPO-R) augmented with multi-dimensional tool-use rewards—namely, retouching-operation accuracy (evaluating both global and region-specific parameter prediction) and perceptual quality (assessing the visual fidelity of retouched outputs)—to refine decision-making and generalization. Finally, we introduce the Agent-to-Lightroom (A2L) protocol to enable seamless, automated Lightroom editing with bidirectional feedback. Consequently, JarvisArt deeply understands the intent of the user, generates diverse stylistic renditions, and seamlessly executes global and region-specific adjustments to produce visually compelling results (see Figure 1).

Our contributions can be summarized as follows:

- We introduce JarvisArt, an intelligent artist agent powered by an MLLM and linked to over 200 Lightroom operations, capable of producing diverse, user-driven stylistic edits that surpass current automated methods and rival professional human retouchers.

- We design a scalable data-synthesis pipeline to construct the MMArt dataset, comprising 5K standard instruction-based and 50K Chain-of-Thought–enhanced multi-granularity samples for detailed retouching tasks.

- We develop a two-stage post-training regime: SFT followed by GRPO-R with tailored tool-use rewards to enhance the agent's reasoning, tool proficiency, and generalization.

- We establish an Agent-to-Lightroom communication protocol that enables seamless collaboration between JarvisArt and Lightroom, facilitating fully automated editing workflows.

## 2 Related Work

**Photo Retouching.** Existing automated pipelines have been proposed to streamline manual retouching. Zeroth- and First-order optimizations [15, 44, 43, 6, 54, 65, 53] were early attempts, but they are constrained by limited parameter prediction and reliance on pre-trained proxies. RL-based methods [55, 29, 28, 19, 45] attempt to mimic human workflows and offer some transparency but fail to capture artistic vision and lack deeper user interaction. Diffusion models [68, 2, 57, 75, 7, 59, 10] dominate high-fidelity image synthesis but rely on static prompts and lack multi-turn reasoning or flexible language alignment, limiting open-ended editing. Additionally, recent unified image editing models have achieved dual breakthroughs in comprehension and generation. Notable examples include closed-source models like GPT-4o [24] and Gemini-2-Flash [52], as well as open-source models such as Bagel [9], Janus-Pro [8], UniTok [41], QLIP [73], and VARGPT-v1.1 [76]. Despite these breakthroughs, three key limitations remain: (1) destructive editing by regenerating all pixels, compromising content preservation; (2) lack of interactive and interpretable local attribute control (e.g., softening or brightening skin); and (3) the absence of arbitrary-resolution editing due to generative model architectural constraints. Conversely, our study presents an interactive and interpretable retouching paradigm that integrates multimodal understanding with expert-level editing tools for non-destructive photo retouching. JarvisArt empowers users through a human-agent collaboration loop, enabling scene-level edits alongside precise region-specific tweaks-blending creative flexibility with the rigor of a professional workflow.

**Reinforcement Fine-Tuning.** Rule-based reinforcement fine-tuning, as demonstrated by OpenAI's o1 [25] and Deepseek-R1 [13], has shown impressive performance in tasks such as mathematical reasoning [4, 25, 61, 64], and code generation [22, 26, 67, 70]. Subsequent research has extended this approach to multimodal models, designing task-specific reward functions for visual perception tasks. These include correct class prediction in image classification [46, 5, 42], Intersection-over-Union (IoU) metrics in image localization and detection [39, 21, 62, 49], accurate click position prediction in GUI grounding tasks [40, 56], and effective interaction with search engines to leverage up-to-date external information [27]. However, unlike these tasks with a single correct answer, our task involves tool-integrated retouching, which requires predicting multiple tools and their parameters. Designing effective reward signals to support learning in this setting remains an open and underexplored challenge. In this paper, we propose customized tool-use rewards, enabling JarvisArt to equip advanced artistic reasoning and tool invocation capabilities.

**LLM-Empowered Agent.** LLM-powered agents have revolutionized AI systems due to three key developments: 1) unprecedented reasoning capabilities of LLMs [13, 4, 60, 16, 17]; 2) advancements in tool manipulation and environmental interaction [31, 51, 18, 20, 38, 34, 35, 11] and 3) sophisticated memory architectures that support longitudinal experience accumulation [14, 71, 58, 63]. Despite these advancements, three fundamental limitations persist when applying LLM agents to professional photo retouching: 1) the absence of a domain-specific retouching knowledge base, which hinders accurate interpretation of user intent, 2) limited decision-making abilities in selecting suitable tools and determining precise parameter values, and 3) absence of standardized protocols to ensure compatibility with professional retouching software integrations. To address these limitations, we propose JarvisArt, a powerful artistic agent that integrates three core capabilities: (1) professional retouching expertise for precise understanding of user instructions, (2) proficiency with commercial retouching tools in Lightroom, and (3) standardized communication protocols for seamless Lightroom integration.

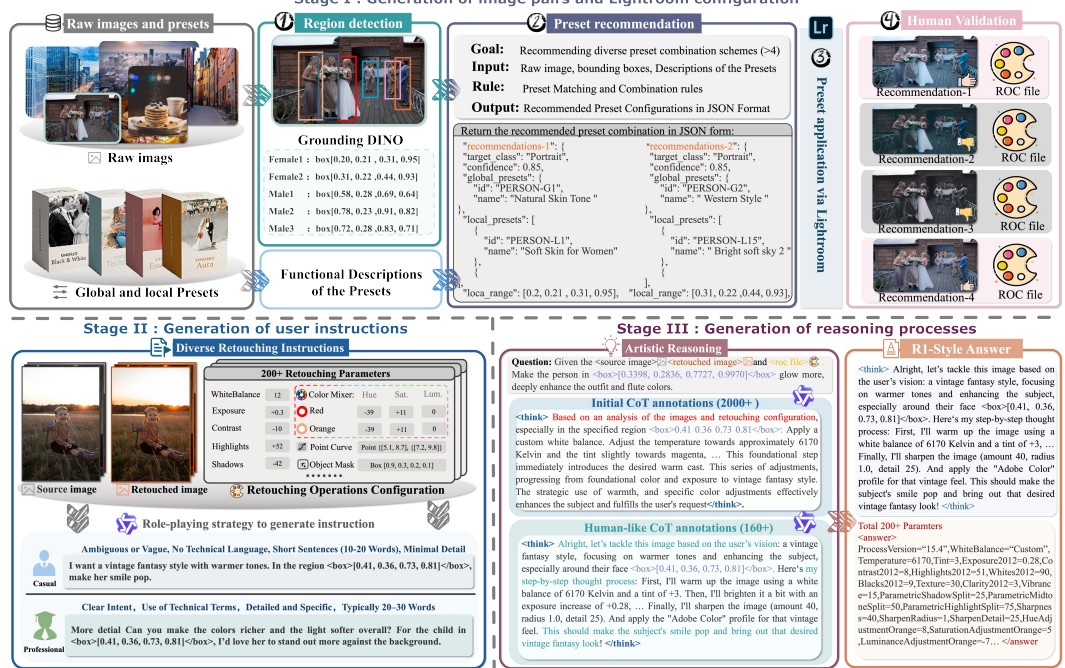

Figure 2: The data generation pipeline comprises three main stages: 1) Curation of diverse source–target examples covering varied scenes and styles with corresponding Lightroom configurations; 2) Generation of diverse user instrcutions that reflects different creative intents; 3) Production of Chain-of-Thought traces that simulate a human artist reasoning process.

## 3 Method

We begin by outlining the overall workflow of JarvisArt (Sec. 3.1). Next, we introduce a comprehensive data generation pipeline that constructs MMArt, a high-quality dataset comprising instruction and reasoning samples for agentic photo retouching tasks (Sec. 3.2). Finally, we investigate the core components of JarvisArt (Sec. 3.3), including a two-stage post-training pipeline and the Agent-to-Lightroom (A2L) protocol, which allows seamless collaboration between JarvisArt and Lightroom.

### 3.1 Overview

JarvisArt is an interactive, MLLM-based photo-retouching system that supports both scene-level and region-level edits. In addition to textual instructions, users can specify local areas via free-form brushstrokes or draggable bounding boxes. In Figure 1, JarvisArt's pipeline comprises three stages: 1) Multi-modal context understanding to parse user directives, image content, and regions of interest; 2) Strategic reasoning grounded in photographic principles to formulate a retouching plan; and 3) Tool orchestration to select appropriate Lightroom operations and parameters. These operations are executed automatically through the A2L protocol. Formally, JarvisArt implements a function:

$$f(Q, I_{\text{src}}) \rightarrow \mathcal{T} = \{t_1, t_2, \ldots, t_n\},$$

where $Q$ is the user query, $I_{\text{src}}$ the source image, and each $t_i$ denotes a specific Lightroom edit (*e.g.*, exposure+0.03). The final output is obtained by $I_{\text{edit}} = g(I_{\text{src}}, \mathcal{T})$, with $g(\cdot)$ representing Lightroom's execution environment.

### 3.2 Data Generation Pipeline

We design a three-stage data-generation pipeline (Figure 2) to construct MMArt with explicit Chain-of-Thought (CoT) annotations. Each sample is a five-tuple $\langle I_{\text{src}}, I_{\text{tgt}}, Q, \mathcal{C}, O \rangle$, where $I_{\text{src}}$ and $I_{\text{tgt}}$ are the before-/after-retouch images, $Q$ the user's instruction, $\mathcal{C}$ the CoT reasoning wrapped in `<think>` tags, and $O$ the retouching operation configuration (ROC) file of tool invocations and parameters

within `<answer>` tags. The pipeline proceeds as follows: 1) Curation of diverse source–target examples covering varied scenes and styles, and the corresponding Lightroom configurations; 2) Generation of natural-language instructions that reflect user intents; 3) Production of step-by-step reasoning traces. Further statistics and examples of MMArt can be found in Appendix A.

**Stage I: Generation of image pairs and Lightroom configuration.** We source raw images from PPR10K [32], the Adobe Lightroom community, and licensed open-source collections, then curate a diverse library of global and local artistic presets[2]. Leveraging Qwen2.5-VL-72B [60] for multimodal role-playing and Grounding DINO [37] for precise region localization, we simulate expert-level edits in four steps: 1) *Region detection*, in which Grounding DINO [37] identifies regions of interest (confidence $> 0.8$); 2) *Preset recommendation*, where Qwen2.5-VL-72B [60] proposes global and local presets based on image aesthetics; 3) *Preset application*, applying each recommendation in Lightroom to generate five candidate retouched images; and 4) *Human-in-the-loop validation*, selecting the most artistically pleasing outputs. Each finalized sample comprises $\langle I_{src}, I_{tgt}, O \rangle$, denoting the source image, the retouched image, and the detailed record of Lightroom operations. The role-playing prompts are detailed in Appendix A.4.

**Stage II: Generation of user instructions.** To simulate diverse editing intents, we employ Qwen2.5-VL-72B [60] with a role-playing prompt (Appendix A.4) to translate each $\langle I_{src}, I_{tgt}, O \rangle$ triplet into both scene-level and region-level instructions $Q$. We generate descriptions for two user types—casual users and professional editors with advanced aesthetic sensibilities, ensuring coverage of simple global edits as well as precise, localized adjustments.

**Stage III: Generation of reasoning processes.** For each sample quadruple $\langle I_{src}, I_{tgt}, Q, O \rangle$, we first apply QVQ-max's [60] advanced visual reasoning to generate initial CoT annotations. To remove redundancy and enforce human-like coherence, we subsequently refine these traces using Qwen2.5-VL-72B [60] through iterative multimodal prompts, producing concise, context-rich reasoning processes $\mathcal{C}$. Full prompt templates are provided in Appendix A.4.

### 3.3 JarvisArt Framework

#### 3.3.1 CoT Supervised Fine-tuning

Drawing on Deepseek-R1 [13], we initialize JarvisArt via supervised fine-tuning on CoT annotations to bootstrap its subsequent reinforcement learning. This phase 1) enforces a consistent, structured output format, 2) instills foundational reasoning skills spanning user-intent interpretation and aesthetic judgment, and 3) establishes preliminary proficiency in selecting Lightroom tools and configuring their parameters.

#### 3.3.2 Reasoning-oriented Reinforcement Learning

Building on the SFT-initialized model, as shown in Figure 3, we apply group relative policy optimization for retouching(GRPO-R) [48] (Appendix B.1) to further refine JarvisArt's artistic reasoning and tool-use proficiency. GRPO-R trains the agent with three interpretable, task-specific rewards: a format reward $R_f$ that enforces structured output, a retouching operation accuracy reward $R_{roa}$ that measures the correctness of selected tools and their parameter settings, and a perceptual quality reward $R_{pq}$ that assesses the visual fidelity of the retouched image. The overall objective is thus $R = R_f + R_{roa} + R_{pq} \in [0, 3]$.

**Format reward.** Following prior work [13, 62, 49, 56], we include a format reward $R_f \in [0, 1]$ to enforce structured outputs: reasoning must appear within `<think>` tags and tool invocations within `<answer>` tags, ensuring consistent and reliable parsing.

**Retouching operation accuracy reward.** Inspired by existing explorations of reward designs [47, 27, 40, 56] in the fields of GUI and web searching. We consider over 200 retouching tools in Lightroom, containing both global adjustments—such as exposure, highlights, and tone curve—and local refinements using six types of masks: 1) linear masks for directional gradients, 2) radial masks for circular or elliptical regions, 3) object masks for isolating subjects (*e.g.*, people or objects), 4) color masks for hue-specific adjustments, 5) luminance masks for brightness-based selections, and 6)

---

[2]In Lightroom, a preset refers to a pre-saved retouching operation configuration (ROC) file, typically created by photographers or artists. It encapsulates stylistic parameters tailored for specific visual aesthetics.

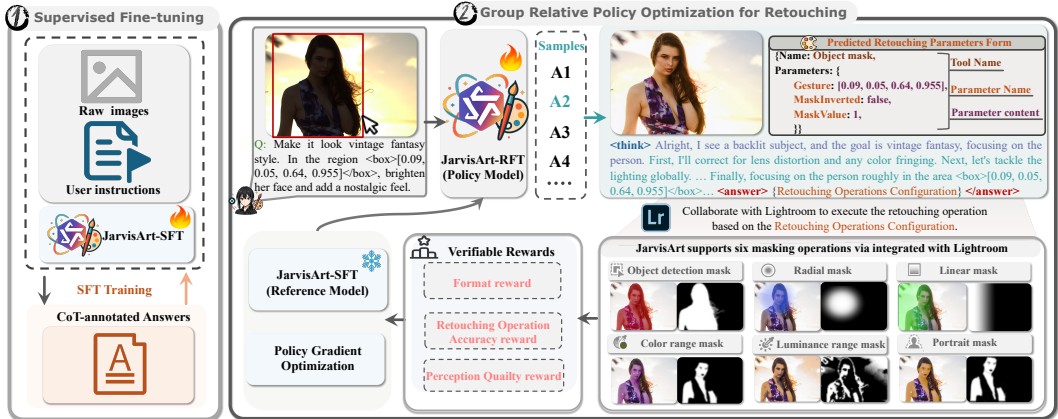

Figure 3: Overview of the two-stage post-training framework. Initially, JarvisArt undergoes supervised fine-tuning (SFT) on CoT-annotated data to develop foundational artistic reasoning and tool-use skills. Following this, we apply the Group Relative Policy Optimization for Retouching (GRPO-R) algorithm to further enhance the JarvisArt's reasoning, tool proficiency, and generalization.

portrait masks for fine-tuning facial features such as skin and eyes. Further details are provided in Appendix E. To assess the accuracy of predicted tools and their parameters, $T^{pre} = \{T_1^{pre}, ..., T_M^{pre}\}$, against the ground truth $T^{tgt} = \{T_1^{tgt}, ..., T_N^{tgt}\}$, we define the ROA reward based on three evaluation criteria:

❶ *Tool name matching:*

$$r_{\text{name}} = \frac{|N_{T^{pre}} \cap N_{T^{tgt}}|}{|N_{T^{pre}} \cup N_{T^{tgt}}|} \in [0, 1], \tag{1}$$

where $N_{T^{pre}}$ and $N_{T^{tgt}}$ are the sets of tool names in the predicted and target sequences, respectively.

❷ *Parameter name matching:*

$$r_{\text{param}} = \sum_{T_j^{tgt} \in T^{tgt}} \sum_{T_i^{pre} \in T^{pre}} \frac{|\text{keys}(T_i^{pre}) \cap \text{keys}(T_j^{tgt})|}{|\text{keys}(T_i^{pre}) \cup \text{keys}(T_j^{tgt})|} \in [0, |T^{tgt}|], \tag{2}$$

where $\text{keys}(\cdot)$ denotes the set of parameter names associated with a predicted or ground-truth tool. It is noted that an overlap in parameter names occurs only when the predicted and ground-truth tool names match.

❸ *Parameter value matching:*

$$r_{\text{value}} = \sum_{T_j^{tgt} \in T^{tgt}} \sum_{T_i^{pre} \in T^{pre}} \sum_{k \in \text{keys}(T_j^{tgt})} S_k \left( T_i^{pre}[k], T_j^{tgt}[k] \right) \in [0, \sum_{T_j^{tgt} \in T^{tgt}} |\text{keys}(T_j^{tgt})|], \tag{3}$$

where $S_k(\cdot) \in [0, 1]$ quantifies the correspondence between predicted and ground-truth parameter values, with a value of 1 indicating an exact match. Specifically, if the key $k$ is absent in $T_i^{pre}$, then $T_i^{pre}[k]$ is undefined and $S_k = 0$. The computation of $S_k$ depends on the parameter type: scalar differences for standard numerical values, intersection-over-union (IoU) for object masks, endpoint distance for linear masks, geometric similarity for radial masks, color distance between sampled points for color masks, luminance range differences for luminance masks, and category-specific criteria for portrait masks. Refer to Appendix B.2 for further details. Finally, the retouching operation accuracy reward is computed by measuring the matching degree between $T^{pre}$ and $T^{tgt}$:

$$R_{\text{roa}} = \frac{1}{3} \left( r_{\text{name}} + \frac{r_{\text{param}}}{|T^{tgt}|} + \frac{r_{\text{value}}}{\sum_{T_j^{tgt} \in T^{tgt}} |\text{keys}(T_j^{tgt})|} \right) \in [0, 1]. \tag{4}$$

**Perception quality reward**. While parameter-based rewards offer critical guidance, they may not fully capture the perceptual quality of the final image, as different parameter settings can produce visually similar results. To address this limitation, we introduce the PQ reward, which evaluates two

key aspects: 1) global tone consistency via color distribution matching, and 2) pixel-wise fidelity. The reward is defined as:

$$R_{pq} = \gamma \cdot \mathrm{CD}(I_{\mathrm{edit}}, I_{\mathrm{tgt}}) + (1 - \gamma) \cdot \mathrm{L}(I_{\mathrm{edit}}, I_{\mathrm{tgt}}) \in [0, 1], \tag{5}$$

where $I_{\mathrm{edit}}$ is the retouched image and $I_{\mathrm{tgt}}$ is the target image. $\mathrm{CD}(\cdot)$ measures color distribution similarity in CIELAB space [69] and $\mathrm{L}(\cdot)$ denotes the pixel-wise distance. Both metrics are normalized to the range [0, 1], with higher values indicating better similarity. The weighting factor is empirically set to $\gamma = 0.4$ to balance both terms.

### 3.3.3 Agent-to-Lightroom Protocol

Figure 4 presents the Agent-to-Lightroom (A2L) protocol, a standardized client-server interface that integrates JarvisArt with Lightroom. The workflow comprises five stages: 1) handshake, 2) file verification, 3) sandboxed execution, 4) async processing, and 5) result return. A2L features dual-transport communication, a structured message format, and resource management. Messages use bar-delimited commands for processing, status, and error handling, enhancing clarity and efficiency. It manages source images and retouching operation configuration (ROC) files, supporting ROC-to-Lua translation, and integrity checks. The source image can by directly retouched by Lua file in Lightroom. The Lua file can be directly applied in Lightroom to retouch the source image. Additional details are provided in the supplementary materials.

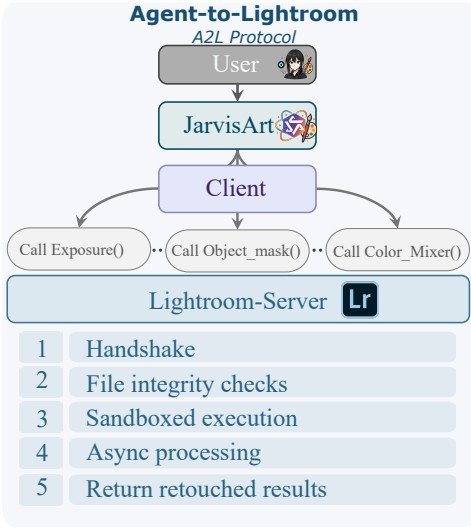

Figure 4: Agent-to-Lightroom protocol.

## 4 Experiment

### 4.1 Experimental Setup

**Implementation details.** We adopt Qwen2.5-VL-7B-Instruct [1] as the base model for JarvisArt. The CoT supervised fine-tuning phase is performed on 50K CoT-annotated instances from MMArt, with a batch size of 2, a learning rate of 1e-5, and training for 2 epochs using the Llama-Factory framework [74] on 8 A100 (80G) GPUs. The reinforcement learning phase, employing the GRPO-R algorithm, is conducted on 5K standard instruction samples from MMArt, using the veRL framework [50]. For each training step, we sample a batch of 2, a learning rate of 1e-6, and generate 4 responses per query, training for 2 epochs on 16 A100 (80G) GPUs.

**MMArt-Bench.** To provide a comprehensive evaluation of JarvisArt's performance, we introduce the MMArt-Bench, which is sampled from the MMArt dataset. It includes four main scenarios: portrait, landscape, street scenes, and still life, with 50 instances per category, totaling 200 instances. Each primary category contains multiple subcategories (Appendix A.1). For region-level evaluation, we utilize a portrait subset comprising 50 human-centered images with mask annotations.

**Evaluation metrics.** Following previous works [68, 30], six assessment metrics are use for evaluation: L1, L2, SC, PQ, and O. L1 and L2 to measure the average pixel-level absolute difference between the retouched image and reference image. SC evaluates the alignment between the instruction text and the image (0–10 scale). PQ evaluates contextual coherence and artifact presence (0–10 scale). The overall score O is calculated as $\mathrm{O} = \sqrt{\mathrm{SC} \times \mathrm{PQ}}$. For region-specific evaluation, we apply these six metrics to a specified mask region. Further details are provided in Appendix C.1.

**Baselines.** For a fair comparison, we evaluate JarvisArt against leading open-source photo retouching methods, including 3DLUT [66] and RSFNet [45], as well as instruction-driven editing models such as InstructPix2Pix [2], MagicBrush [68], OmniGen [57], VARGPT-v1.1 [76] and Step1X-Edit [36]. Proprietary solutions such as GPT-4o[3] [24] and Gemini-2- Flash [52][4] are also included

---

[3] The results are obtained based on ChatGPT APP in May 2025.
[4] The results are obtained based on Gemini API in May 2025.

Table 1: Quantitative evaluation on MMArt-Bench. We highlight the best and second-best instruction-based results. SC, PQ, and O refer to the metrics evaluated by Gemini-2-Flash. The $RC$ means the metric calculated on specific mask region.

| Method | Instruction | Scene-level | | | | | Region-level | | | | |
|---|---|---|---|---|---|---|---|---|---|---|---|
| | | $L1_{\times 10^2}$ ↓ | $L2_{\times 10^3}$ ↓ | SC ↑ | PQ ↑ | O ↑ | $L1^{RC}_{\times 10^2}$ ↓ | $L2^{RC}_{\times 10^3}$ ↓ | $SC^{RC}$ ↑ | $PQ^{RC}$ ↑ | $O^{RC}$ ↑ |
| RSFNet [45] | ✗ | 11.61 | 26.38 | - | - | - | 8.80 | 13.69 | - | - | - |
| 3DLUT [66] | ✗ | 11.50 | 25.99 | - | - | - | 8.33 | 12.39 | - | - | - |
| InstructPix2Pix [2] | ✓ | 15.67 | 47.51 | 6.54 | 7.79 | 7.10 | 12.62 | 33.39 | 4.70 | 5.36 | 4.91 |
| MagicBrush [68] | ✓ | 18.39 | 65.25 | 3.93 | 4.09 | 3.85 | 12.37 | 32.81 | 3.04 | 3.41 | 3.13 |
| OmniGen [57] | ✓ | 28.49 | 133.45 | 4.25 | 4.42 | 4.13 | 25.16 | 109.10 | 6.17 | 7.56 | 6.72 |
| VARGPT-v1.1 [76] | ✓ | 27.05 | 126.47 | 1.83 | 1.38 | 1.48 | 23.71 | 107.32 | 1.38 | 1.15 | 1.08 |
| Step1X-Edit [36] | ✓ | 24.28 | 105.91 | 7.52 | 8.67 | 8.01 | 15.43 | 45.85 | 8.32 | 9.04 | 8.66 |
| Gemini-2-Flash [52] | ✓ | 23.07 | 90.99 | 7.62 | 8.78 | 8.08 | 16.52 | 52.88 | 8.04 | 9.25 | 8.61 |
| GPT-4o [24] | ✓ | 22.84 | 92.23 | 8.73 | 9.66 | 9.18 | 15.71 | 47.87 | 8.59 | 9.48 | 9.03 |
| **JarvisArt** | ✓ | 12.44 | 30.56 | 7.53 | 9.82 | 8.52 | 7.63 | 12.14 | 8.08 | 9.39 | 8.69 |

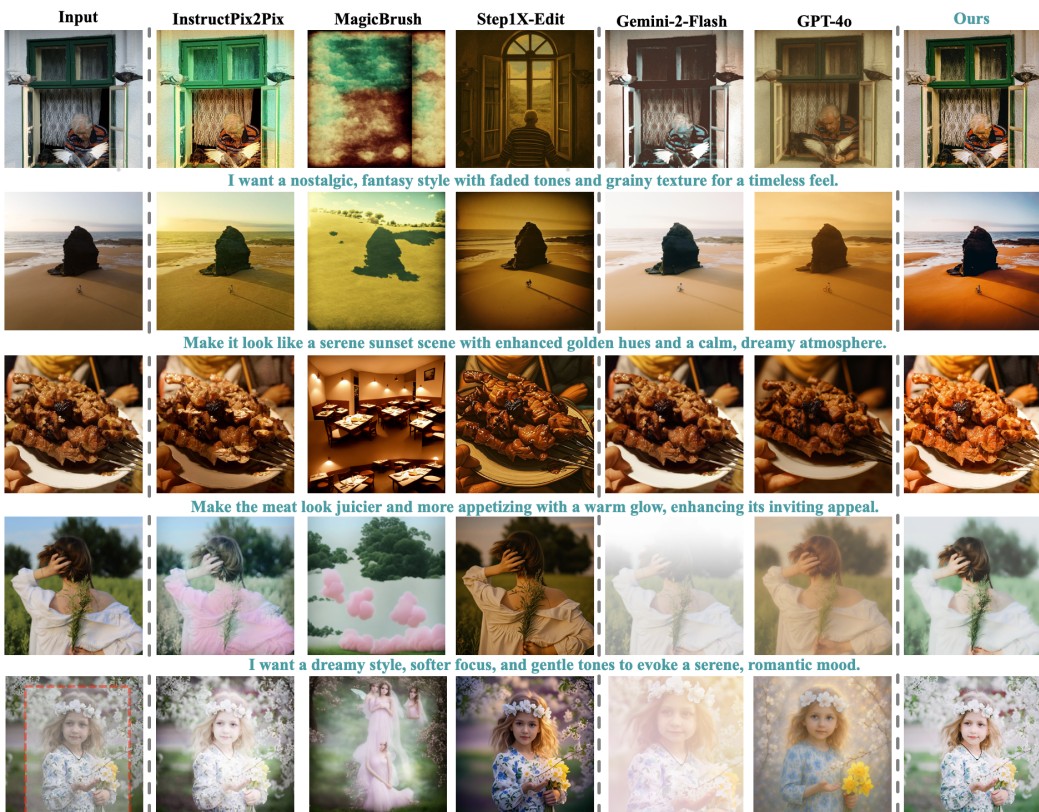

| Input | InstructPix2Pix | MagicBrush | Step1X-Edit | Gemini-2-Flash | GPT-4o | Ours |

I want a nostalgic, fantasy style with faded tones and grainy texture for a timeless feel.

Make it look like a serene sunset scene with enhanced golden hues and a calm, dreamy atmosphere.

Make the meat look juicier and more appetizing with a warm glow, enhancing its inviting appeal.

I want a dreamy style, softer focus, and gentle tones to evoke a serene, romantic mood.

Make the objects in the <box>0.212, 0.088, 0.841, 1.000</box> more prominent and the background more lightful.

Figure 5: Visual comparison of different methods on MMArt-Bench.

for comparison. Notably, all test images are cropped to a $512 \times 512$ resolution, as some baselines are incapable of processing high-resolution or arbitrarily sized inputs.

## 4.2 Experimental Results

### 4.2.1 Evaluation on MMArt-Bench

As shown in Table 1, JarvisArt outperforms most open-source instruction-based baselines, achieving state-of-the-art performance across all 10 evaluation metrics. Compared to closed-source models such as GPT-4o [25] and Gemini-2-Flash [52], JarvisArt achieves superior content preservation—for

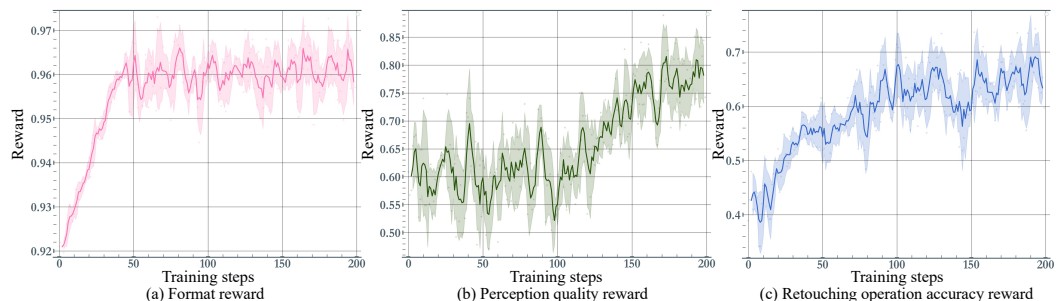

Figure 7: Visualization of the reward trends across training steps of for JarvisArt.

instance, an $L1_{\times 10^2}$ score of 12.44, which is 45.6% lower (and thus better) than GPT-4o's score of 22.84. JarvisArt also demonstrates competitive instruction-following capability (O = 8.52), closely matching GPT-4o (O = 9.18) and outperforming Gemini-2 Flash (O = 8.08). Notably, in the local editing setting—where content fidelity is especially critical—the advantage of our method over GPT-4o and Gemini-2-Flash is significantly amplified. As illustrated in Figure 5, especially in portrait scenarios, competing methods often exhibit noticeable uncanny valley effects, producing significant visual artifacts that diverge from users' creative intent. In contrast, JarvisArt mitigates these issues through its Lightroom-integrated workflow, enabling high-quality, non-destructive editing. More results in Appendix D.

#### 4.2.2 User Preference Study

Evaluating instruction-driven photo retouching remains inherently subjective, as even expert evaluators often disagree on the "optimal" outcome. To quantify preferences, we conducted a user preference study on the MMArt-Bench, recruiting 80 participants to evaluate four

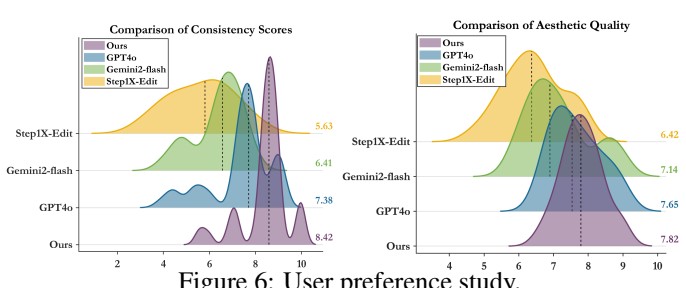

Figure 6: User preference study.

advanced algorithms: Step1X-Edit [36], Gemini-2-Flash [52], GPT-4o [24], and JarvisArt. Evaluations focus on two criteria: (1) image consistency (preservation of source image content) and (2) aesthetic quality (visual appeal of retouched results). A five-point ordinal scale (worst = 2, poor = 4, fair = 6, good = 8, and excellent = 10) for quantitative metrics. Results in Figure 6 show JarvisArt achieves best subjective quality, producing edits favored by users.

#### 4.2.3 Visualization of Reward Trends for GRPO-R

Figure 7 shows additional visualizations of GRPO-R training. The format reward converges quickly early on. While the PQ reward initially fluctuates and grows gradually, the ROA reward rises more rapidly—likely because the model inherits *"parameter preferences"* from the SFT phase. As a result, it first focuses on the more easily optimized ROA, then gradually shifts attention to the PQ reward, which requires longer exploration due to the broader search space, where different edit operations may yield similar visual outcomes. Moreover, unlike Deepseek-R1 [13], JarvisArt does not display a clear "aha moment". This absence may stem from the lack of intermediate visual feedback during the artistic reasoning process. For example, when the model makes a hypothetical retouching adjustment like highlight+5, it cannot obtain the corresponding visual result, preventing the model from validating this step's correctness within the decision-making chain. Unlike mathematical problem-solving, where each step can be validated immediately, our artistic reasoning involves numerous retouching parameters. If we perform step-wise validation for each parameter, it would require high concurrency in calling Lightroom. This is impractical due to the high computational cost and the slow training speed. Investigating step-wise visual rewards within proxy validation environments may offer a promising approach to eliciting the "aha moment". We intend to explore in future work.

Table 2: Ablation studies on different training strategies and reward design.

| Configurations | L1$_{\times 10^2}$ ↓ | L2$_{\times 10^3}$ ↓ | SC ↑ | PQ ↑ | O ↑ |
|---|---|---|---|---|---|
| *Training strategy* | | | | | |
| only SFT | 14.42 | 44.38 | 7.32 | 8.67 | 7.94 |
| only RL | 17.55 | 58.19 | 6.88 | 8.13 | 7.38 |
| SFT + RL (Ours) | **12.44** | **30.56** | **7.53** | **9.82** | **8.52** |
| *Reward design* | | | | | |
| Format + ROA | 14.09 | 40.36 | 7.45 | 8.77 | 8.04 |
| Format + PQ | 13.78 | 35.41 | 7.48 | 8.92 | 8.15 |
| Format + ROA + PQ (Ours) | **12.44** | **30.56** | **7.53** | **9.82** | **8.52** |

## 5 Ablation Study

**Training strategy.** We assess the impact of different post-training strategies by comparing model performance under three settings: 1) SFT on 50K CoT-enhanced samples, 2) GRPO-R training on 5K standard samples from scratch; and 3) GRPO-R fine-tuning basd on SFT-initial model. Rows 2–4 in Table 2 show that SFT yields better results than GRPO-R trained from scratch. This is likely because, without SFT to instill the basic reasoning and tool-use abilities, the GRPO-R training process must explore a significantly larger search space, thereby hindering optimization. Our combined SFT+GRPO-R strategy achieves the best results, suggesting that GRPO-R can effectively enhance the SFT-initialized model's reasoning, tool proficiency, and generalization by expanding its exploration capacity.

**Reward design.** As shown in Rows 6–8 of Table 2, individual reward combinations (Format+ROA or Format+PQ) result in suboptimal performance, with Format+PQ performing slightly better—possibly because PQ aligns more closely with the ultimate objective of enhancing visual quality and offers a broader optimization space to escape local optima. The full combination (Format+PQ+ROA) achieves the highest performance. This result aligns with our intuition that parameter-oriented (ROA) and perception-driven (PQ) rewards are complementary: ROA ensures parameter accuracy, while PQ maintains visual fidelity. The multi-dimensional reward system provides a balanced optimization signal, guiding the model to predict accurate edit operations while preserving high visual quality.

## 6 Conclusion

This report introduces JarvisArt, an interactive and interpretable MLLM-guided agent that integrates with 200+ Lightroom editing tools, enabling non-destructive editing on images of any-resolution. To develop this artist agent, we propose a new data generation pipeline that curates the MMArt-55K dataset, comprising 5K standard and 50K CoT-enhanced samples. Based on this dataset, we train JarvisArt using a two-stage post-training regimen: 1) CoT SFT to instill basic reasoning and tool-use abilities, and 2) GRPO-R to improve the agent's reasoning, tool proficiency, and generalization through customized tool-use rewards: retouching operation accuracy reward for assessing the predicted editing operations, and the perceptual quality reward to evaluate the visual fidelity of the edited outputs. Furthermore, to enable seamless, automated Lightroom editing, we introduce the Agent-to-Lightroom protocol. Evaluation results from our MMArt-Bench demonstrate that our proposed algorithm significantly outperforms existing advanced image editing algorithms.

## 7 Acknowledgments

This work was supported in part by the Xiamen science and technology major project under Grant 3502Z20241029; in part by the Dreams Foundation of Jianghuai Advance Technology Center project under Grant 2023-ZM01D002; in part by the National Natural Science Foundation of China under Grant 82172073, Grant 82272071, and Grant 62271430; and in part by the Open Fund of the National Key Laboratory of Infrared Detection Technologies.

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

# Appendices

Our Appendices includes the following sections:

# A    Details of the MMArt dataset.

## A.1    Statistics of the MMArt dataset

Figure 8(a) illustrates the composition and distribution of scenarios in our MMArt dataset. The dataset is structured into four major scene categories that reflect common real-world photo retouching contexts: portrait (40.8%, including shooting purposes, shooting time/lights, subjects, and indoor/outdoor scenes), landscape (33.3%, comprising nature, city, aerial photography,travel, underwater, night scene and architecture), street scenes (5.71%, including sports, life, event and documentary), and still life (20.2%, encompassing food, close-up scenes, black/white photography, art and animals). Each major category contains a diverse set of subcategories, ensuring comprehensive coverage and representativeness. Furthermore, Figure 8(b) displays a word cloud of user instructions, highlighting the linguistic diversity of the instructions.

## A.2    Comparison of Existing Datasets

Table 3 presents a comparison between our MMArt dataset and existing image editing datasets. MMArt is designed with the following key properties to facilitate advanced research in image retouching:

- **Real Images:** All samples in MMArt are real photographs, ensuring the dataset's authenticity and practical value for real-world applications.
- **Diverse User Instructions:** Each image is paired with detailed user instructions, capturing a wide variety of editing intentions and reflecting the diversity of natural language expressions.
- **Flexible Resolution:** MMArt supports images of any resolution, including high-resolution samples, making it suitable for both research and practical deployment scenarios.
- **Chain-of-Thought (CoT) Annotations:** The dataset provides CoT reasoning annotations, which help to reveal the underlying logic and step-by-step process of user intent understanding and image editing.
- **Lightroom Retouching Configuration:** For every sample, MMArt includes comprehensive Lightroom parameter configurations, allowing for non-destructive, reproducible, and transparent image editing.

These properties make MMArt a high-quality, flexible, and richly annotated resource for the development and evaluation of advanced image retouching techniques.

## A.3    Data Samples of MMArt

The diversity of collected photos is shown in Figure 9. Moreover, Figure 10 demonstrates MMArt samples with Chain-of-Thought (CoT) reasoning, while Figure  11 shows standard examples without CoT annotations.

Table 3: Comparison of MMArt and existing retouching datasets in terms of data properties.

| Property | InstructP2P [2] | MagicBrush [68] | UltraEdit [72] | MGIE [12] | HQEdit [23] | FiveK [3] | MMArt |
|---|---|---|---|---|---|---|---|
| Real Image? | ✗ | ✓ | ✓ | ✓ | ✗ | ✓ | ✓ |
| User Instructions? | ✓ | ✓ | ✓ | ✓ | ✓ | ✗ | ✓ |
| Any Resolution? | ✗ | ✗ | ✗ | ✗ | ✗ | ✓ | ✓ |
| High Resolution? | ✗ | ✗ | ✗ | ✗ | ✓ | ✓ | ✓ |
| CoT Annotations? | ✗ | ✗ | ✗ | ✗ | ✗ | ✗ | ✓ |
| Lightroom Configuration? | ✗ | ✗ | ✗ | ✗ | ✗ | ✓ | ✓ |

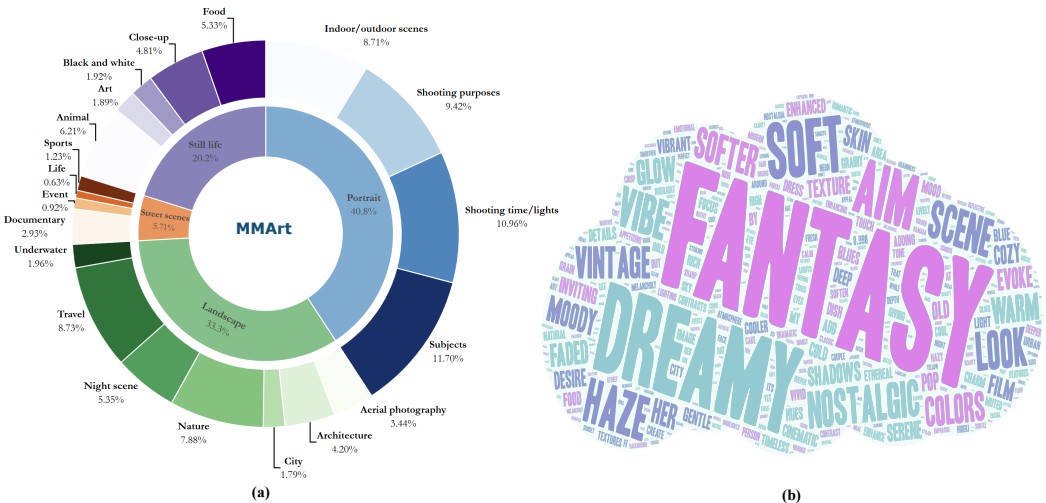

(a)          (b)

Figure 8: Statistics of the MMArt dataset. (a) The dataset is divided into four primary scenarios: portrait, landscape, street scenes, and still life, each containing a variety of subcategories. (b) A word cloud illustrates the rich linguistic diversity found in user instructions.

### A.4 Prompt Templates

The prompt templates utilized throughout the various stages of MMArt are summarized here—Aesthetic Preset Recommendation (Figure 21), User Instruction Simulation (Figures 22 and 23), and Chain-of-Thought Data Construction (Figure 24 and 25).

## B Additional Method Details

### B.1 Group Relative Policy Optimization

In GRPO, given a task question, the model generates a set of $N$ potential responses $\{O_1, O_2, \ldots, O_N\}$. Each response is evaluated by taking the corresponding actions and computing its reward $\{R_1, R_2, \ldots, R_N\}$. Unlike PPO, which relies on a single reward signal and a critic to estimate the value function, GRPO normalizes these rewards to calculate the relative advantage of each response. The relative quality $A_i$ of the i-th response is computed as

$$A_i = \frac{r_i - \text{Mean}(\{r_1, r_2, \ldots, r_N\})}{\text{Std}(\{r_1, r_2, \ldots, r_N\})},$$

where Mean and Std represent the mean and standard deviation of the rewards, respectively. This normalization step ensures that responses are compared within the context of the group, allowing GRPO to better capture nuanced differences between candidates. Policy updates are further constrained by minimizing the KL divergence between the updated and reference models, ensuring stable RL learning. Refer to [13, 48] for more details.

### B.2 Details of Reward Calculation

The parameter value matching function $S_k(\cdot)$ for each parameter $k$ is determined based on its specific type. Let $V_k^{pre}$ and $V_k^{tgt}$ denote the predicted and ground truth values for the $k$-th parameter, respectively. For notational simplicity, we omit the subscript $k$ in the following formulas. The calculation proceeds as follows:

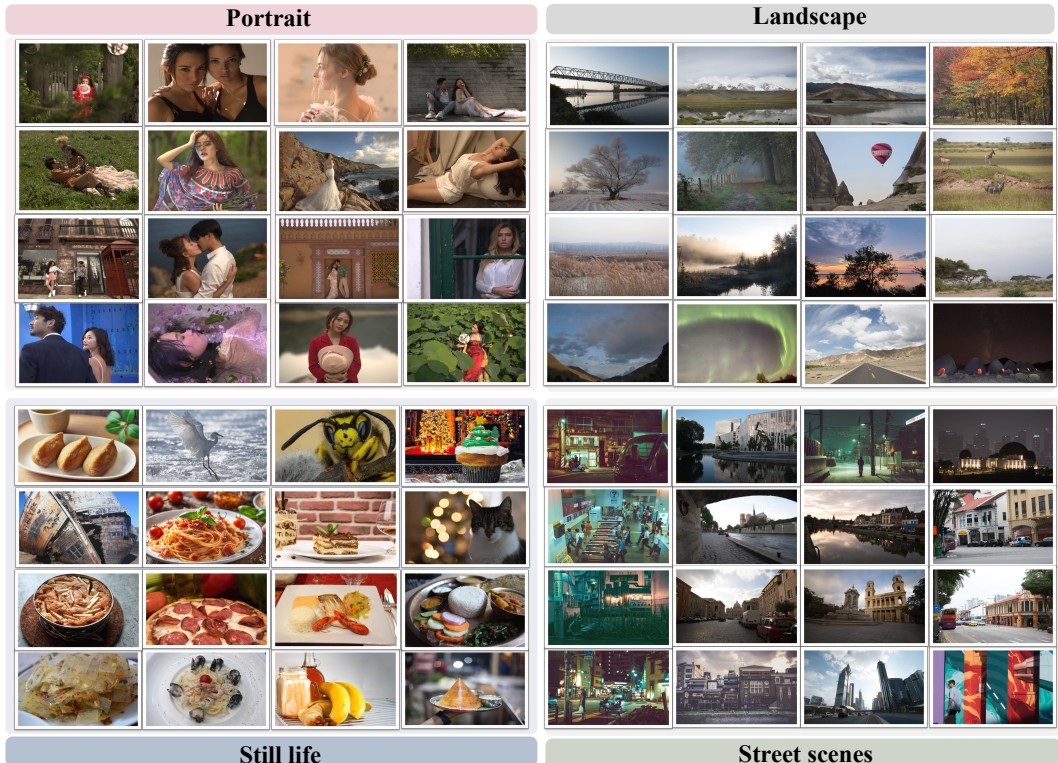

Figure 9: Visual examples to demonstrate the diversity of the proposed dataset.

- **Scalar Parameters.** For scalar parameters such as exposure or contrast, the matching function $S$ is defined as:

$$S = \max \left( 0, 1 - \frac{|V^{pre} - V^{tgt}|}{V_{max} - V_{min}} \right) \in [0, 1],$$

where $| \cdot |$ represents the absolute error between the predicted and ground truth values.

- **Linear Gradient Masks.** We assess the similarity between predicted and target linear gradient masks by measuring the distances between their start points $p_s = (x_s, y_s)$ and end points $p_e = (x_e, y_e)$, with coordinates normalized to [0,1] for resolution invariance. The similarity score is computed as:

$$S = \max \left( 0, 1 - \|p_s^{\text{pre}} - p_s^{\text{tgt}}\| - \|p_e^{\text{pre}} - p_e^{\text{tgt}}\| \right) \in [0, 1],$$

where $\| \cdot \|$ denotes Euclidean distance.

- **Radial Gradient Masks.** We measure similarity between predicted and target radial gradient masks using three geometric parameters: center position $c = (x, y)$, scale factors $(W, H)$, and rotation angle $\theta$. Center point similarity is given by:

$$S_{center} = \max(0, 1 - 2 \cdot \|c^{pre} - c^{tgt}\|) \in [0, 1],$$

where $c^{pre}$ and $c^{tgt}$ are normalized to [0,1]. Further, scaling similarity compares width/height ratios:

$$S_{scale} = \max(0, 1 - |W^{pre}/W^{tgt} - 1| - |H^{pre}/H^{tgt} - 1|) \in [0, 1],$$

The angle numerical value similarity is defined by:

$$S_{angle} = \max \left( 0, 1 - \frac{|\theta^{pre} - \theta^{tgt}|}{\theta_{max} - \theta_{min}} \right) \in [0, 1],$$

The final similarity score combines these components as follows:

$$S = 0.4 \cdot S_{center} + 0.4 \cdot S_{scale} + 0.2 \cdot S_{angle} \in [0, 1].$$

- **Object Masks.** For object masks, the similarity score $S$ is defined as the Intersection-over-Union (IoU) between the predicted $B^{pre}$ and ground truth $B^{tgt}$ bounding boxes. Each box is parameterized as $[x_1, y_1, x_2, y_2]$. The similarity score is computed as:

$$\text{S} = \text{IoU}(B^{pre}, B^{tgt}) \in [0, 1],$$

Table 4: Hyper-parameter settings for SFT and GRPO-R

| Hyper-parameter | SFT | GRPO-R |
|---|---|---|
| Batch size | 2 | 2 |
| Learning rate | $1 \times 10^{-5}$ | $1 \times 10^{-6}$ |
| Weight decay | 0 | 0 |
| Optimizer | AdamW | AdamW |
| Warmup ratio | 0.1 | 0.1 |
| LR scheduler | cosine | cosine |
| Training samples | 50K | 5K |
| Training epochs | 2 | 2 |
| Precision | bfloat16 | bfloat16 |
| KL coefficient | – | 0.04 |
| Reward coefficients | – | $R_f : 1,\ R_{\mathrm{roa}} : 1,\ R_{\mathrm{pq}} : 1\ (\gamma = 0.4)$ |
| Number of generations | – | 4 |
| GPU resources | $8 \times \mathrm{A100}$ ($\approx$384 GPU hours) | $16 \times \mathrm{A100}$ ($\approx$2076 GPU hours) |

where higher values indicate better alignment, with $S = 1$ denoting perfect overlap and $S = 0$ indicating no intersection.

- **Portrait Masks.** In portrait masks, the model predicts different special category IDs to denote distinct regions, such as ID=0 for face, ID=1 for hair, ID=2 for eyes, ID=3 for skin, etc. The matching score $S$ is defined as follows:

$$S = \begin{cases} 1, & \text{if the predicted and target category IDs coincide,} \\ 0, & \text{otherwise.} \end{cases}$$

- **Color Range Masks.** To evaluate color range mask similarity, we sample $N$ representative points from both predicted and target color distributions and compute the mean CIEDE2000 color difference $\Delta E_{100}$ in LAB color space. The similarity score is given by:

$$S = \max\left( 0, 1 - \frac{1}{N} \sum_{n=1}^{N} \Delta E_{100}(c_n^{pre}, c_n^{tgt}) \right) \in [0, 1],$$

where $c_n^{pre}$ and $c_n^{tgt}$ denote the $n$-th sampled colors from the predicted and ground-truth distributions, respectively.

- **Luminance Range Masks.** To evaluate luminance range mask similarity, we compare the predicted and target luminance extremes by computing their absolute differences. The similarity score is defined as:

$$S = \max\left( 0, 1 - \frac{|l_{\min}^{pre} - l_{\min}^{tgt}| + |l_{\max}^{pre} - l_{\max}^{tgt}|}{2(l_{\max}^{tgt} - l_{\min}^{tgt})} \right) \in [0, 1],$$

where the denominator normalizes by the target luminance range to ensure scale invariance.

### B.3 Hyper-parameter Setting

To ensure reproducibility, we provide the complete hyperparameter settings for both the SFT and GRPO-R phases in Table 4.

## C  Additional Experimental Details

### C.1  Calculation of Local Metrics

To evaluate the model's effectiveness in localized regions, we compute six metrics—$\mathrm{L1}^{RC}$, $\mathrm{L2}^{RC}$, $\mathrm{SC}^{RC}$, $\mathrm{PQ}^{RC}$, and $\mathrm{O}^{RC}$—using human-centric masks from the portrait subset of MMArt-Bench. For $\mathrm{L1}^{RC}$ and $\mathrm{L2}^{RC}$, inspired by PPR10K [32], given an image $I$ with resolution $H \times W$, we define a weighting matrix $W_I = [w_{i,j}] \in \mathbb{R}^{H \times W}$, where $w_{i,j} = 1$ for human regions and $w_{i,j} = \alpha(\alpha \le 1)$ for background regions, with $\alpha$ empirically set to 0.5. For instance, the human-centric L1 difference metric is expressed as:

$$\mathrm{L1}^{RC} = |W_I \circ I^{pred} - W_I \circ I^{tgt}|,$$

where $I^{\mathrm{pred}}$ and $I^{\mathrm{tgt}}$ are the predicted and target images, respectively, and $\circ$ denotes element-wise multiplication. The $\mathrm{L2}^{RC}$ metric is defined in a similar manner. For $\mathrm{SC}^{RC}$, $\mathrm{PQ}^{RC}$, and $\mathrm{O}^{RC}$, with $\alpha$ empirically set to 0, we focus solely on the mask region of the edited image and prompt the MLLM to emphasize local adjustments.

Table 5: Quantitative evaluation on MMArt-Bench. We highlight the best and second-best instruction-based results. SC, PQ, and O refer to the metrics evaluated by Qwen2.5-VL-72B [1]. The $RC$ means the metric calculated on specific mask region.

| Method | Instruction | Scene-level | | | | | Region-level | | | | |
|---|---|---|---|---|---|---|---|---|---|---|---|
| | | $L1_{\times 10^2}\downarrow$ | $L2_{\times 10^3}\downarrow$ | SC↑ | PQ↑ | O↑ | $L1^{RC}_{\times 10^2}\downarrow$ | $L2^{RC}_{\times 10^3}\downarrow$ | $SC^{RC}\uparrow$ | $PQ^{RC}\uparrow$ | $O^{RC}\uparrow$ |
| RSFNet [45] | ✗ | 11.62 | 26.38 | - | - | - | 8.80 | 13.70 | - | - | - |
| 3DLUT [66] | ✗ | 11.51 | 26.00 | - | - | - | 8.34 | 12.26 | - | - | - |
| InstructPix2Pix [2] | ✓ | 15.62 | 47.26 | 6.17 | 5.81 | 5.47 | 12.48 | 32.69 | 4.67 | 3.85 | 3.64 |
| MagicBrush [68] | ✓ | 18.31 | 64.76 | 3.93 | 2.25 | 2.44 | 12.44 | 32.93 | 2.80 | 2.15 | 2.01 |
| OmniGen [57] | ✓ | 28.40 | 132.82 | 4.14 | 2.16 | 2.70 | 24.85 | 106.81 | 3.80 | 3.85 | 3.67 |
| VARGPT-v1.1 [76] | ✓ | 27.04 | 126.26 | 1.27 | 0.17 | 0.29 | 23.59 | 105.86 | 0.09 | 0.02 | 0.03 |
| Step1X-Edit [36] | ✓ | 24.17 | 105.14 | 7.03 | 4.94 | 5.71 | 15.27 | 44.93 | 7.50 | 6.89 | 7.11 |
| Gemini-2-Flash [52] | ✓ | 23.06 | 90.96 | 7.65 | 6.77 | 7.00 | 16.74 | 53.76 | 7.33 | 7.17 | 7.19 |
| GPT-4o [24] | ✓ | 22.77 | 91.79 | 8.52 | 7.37 | 7.85 | 15.67 | 47.60 | 8.07 | 7.87 | 7.95 |
| **JarvisArt** | ✓ | 12.66 | 31.88 | 6.19 | 8.51 | 6.67 | 7.75 | 12.38 | 7.54 | 8.46 | 7.91 |

Table 6: Quantitative evaluation on MIT-FiveK [3]. We highlight the best and second-best instruction-based results. SC, PQ, and O refer to the metrics evaluated by Gemini-2-Flash.

| Method | Instruction | $L1_{\times 10^2}\downarrow$ | $L2_{\times 10^3}\downarrow$ | SC↑ | PQ↑ | O↑ |
|---|---|---|---|---|---|---|
| InstructPix2Pix [2] | ✓ | 16.23 | 49.54 | 6.36 | 8.34 | 7.15 |
| MagicBrush [68] | ✓ | 17.29 | 53.45 | 4.92 | 5.50 | 4.95 |
| OmniGen [57] | ✓ | 28.53 | 128.59 | 3.12 | 2.48 | 2.57 |
| VARGPT-v1.1 [76] | ✓ | 26.96 | 117.16 | 2.94 | 2.00 | 2.29 |
| Step1X-Edit [36] | ✓ | 22.08 | 91.72 | 7.20 | 8.48 | 7.69 |
| Gemini-2-Flash [52] | ✓ | 18.69 | 61.27 | 7.86 | 9.22 | 8.47 |
| GPT-4o [24] | ✓ | 21.49 | 78.11 | 8.72 | 9.76 | 9.22 |
| **JarvisArt** | ✓ | 12.98 | 30.05 | 7.36 | 9.82 | 8.48 |

## C.2 Prompt for MLLM-based Metrics

As shown in Figure 20, we present the evaluation prompts utilized for both scene-level and region-level assessments of the Semantic Consistency (SC) and Perceptual Quality (PQ) metrics. Notably, the overall score is calculated as $O = \sqrt{SC \times PQ}$.

## D Additional Experimental Results

### D.1 Additional Quantitative Evaluation by Qwen2.5-VL-72B

As shon in Table 5, to further evaluate MLLM-based metrics, we conducted an additional quantitative analysis using Qwen2.5-VL-72B [1]. Our findings suggest that such metrics may be unreliable, struggle to effectively reflect a model's instruction-following capability. Despite this, our model demonstrates instruction-following performance comparable to that of contemporary SOTA closed-source model GPT-4o, while achieving a significant improvement in content fidelity.

### D.2 Examples of Intricate Retouching Tasks with JarvisArt

Figures 12-15 present the challenging retouching examples, which involve both global and local editing demands, as well as vague user instructions. JarvisArt excels in understanding these ambiguous intentions, applying modifications at both the scene and region levels, and delivering visually effective results in the final images.

### D.3 More Visual Comparisons

Figures 16-19 present additional photo retouching results from the MMArt-Bench and MIT-FiveK, highlighting the superiority of JarvisArt in terms of instruction adherence, content fidelity, and visual appeal, while also achieving the style most similar to the target image. Notably, we also include a visual comparison with two commercial editing tools: Adobe Lightroom and Google Photo' auto-retouching modes.

## D.4   Comparison on MIT-FiveK

To assess the generalization ability of our system, we conduct comprehensive qualitative and visual comparisons on the MIT-FiveK [3] benchmark dataset. Specifically, we randomly select 50 data samples from MIT-FiveK [3] and generate corresponding user instructions based on the source images, expert C's reference images, and the associated retouching parameter configurations (see Section 3.2 for implementation details). As shown in Table 6 and Figures 18 and 19, our system achieves state-of-the-art performance in both instruction-following and content preservation metrics. These results confirm the robustness and effectiveness of our method in faithfully executing user instructions while maintaining original image content. Furthermore, our approach consistently outperforms existing baselines on multiple real-world benchmarks, highlighting its strong generalization ability and practical applicability in interactive photo retouching.

**Input:**               **Output:**

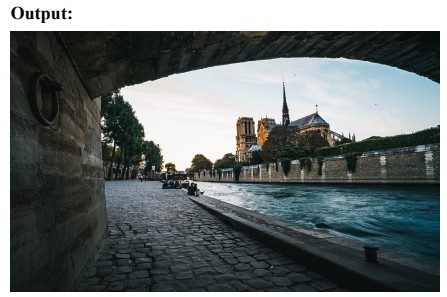

**User Instruction:**

<Image> I wanted the scene to pop more, making the cathedral stand out with enhanced details. The water needed a cooler tone, while the bridge and path should appear darker for depth. Overall, I aimed for a balanced look that emphasizes the architecture and adds a cinematic feel.

**Answer with CoT annotation:**

<think>Alright, I've got this image showing a bridge, cathedral, and river. The goal? Enhance the cathedral's details, cool the water, darken the bridge/path, and achieve a cinematic feel. The image appears muted, with vibrant water needed. For the cathedral, I'll boost clarity to 10 and use sharpening. To cool the water, I'll adjust blue saturation to -40 and apply split toning. Darkening the bridge/path involves local exposure reductions in masks. A cinematic look comes from increasing vibrance to 20, adjusting HSL, and using a vignette. Highlights at -100 may be too much; I'll tweak that. Yellow hue shift (-88) might alter the cathedral unnaturally. I'll check these and refine. Composition-wise, the vignette and local darkening will focus on the cathedral. Final tweaks ensure the image meets the vision.<think>

<answer>CompatibleVersion = 251658240, ProcessVersion = "11.0", WhiteBalance = "Custom", IncrementalTemperature = -9, IncrementalTint = -3, Exposure2012 = 1.56, Contrast2012 = 54, Highlights2012 = -100, Shadows2012 = 100, Whites2012 = -96, Blacks2012 = -30, Texture = 0, Clarity2012 = 31, Dehaze = 0, Vibrance = -21, Saturation = 100, ParametricShadows = 0, ParametricDarks = 0, ParametricLights = 0, ParametricHighlights = 0, ParametricShadowSplit = 25, ParametricMidtoneSplit = 50, ParametricHighlightSplit = 75, Sharpness = 40, SharpenRadius = 0.5, SharpenDetail = 100, SharpenEdgeMasking = 60, LuminanceSmoothing = 27, LuminanceNoiseReductionDetail = 100, LuminanceNoiseReductionContrast = 0, ColorNoiseReduction = 0, HueAdjustmentRed = 0, HueAdjustmentOrange = 0, HueAdjustmentYellow = -100, HueAdjustmentGreen = -100, HueAdjustmentAqua = 0, HueAdjustmentBlue = -11, HueAdjustmentPurple = 0, HueAdjustmentMagenta = 0, SaturationAdjustmentRed = 100, SaturationAdjustmentOrange = 79, SaturationAdjustmentYellow = 12, SaturationAdjustmentGreen = -100, SaturationAdjustmentAqua = 34, SaturationAdjustmentBlue = -14, SaturationAdjustmentPurple = -100, SaturationAdjustmentMagenta = -100, LuminanceAdjustmentRed = 0, LuminanceAdjustmentOrange = 0, LuminanceAdjustmentYellow = 0, LuminanceAdjustmentGreen = 0, LuminanceAdjustmentAqua = 0, LuminanceAdjustmentBlue = -28, LuminanceAdjustmentPurple = 0, LuminanceAdjustmentMagenta = 0, SplitToningShadowHue = 217, SplitToningShadowSaturation = 6, SplitToningHighlightHue = 0, SplitToningHighlightSaturation = 0, SplitToningBalance = 0, ColorGradeMidtoneHue = 0, ColorGradeMidtoneSat = 0, ColorGradeShadowLum = 0, ColorGradeHighlightLum = 0, ColorGradeBlending = 50, ColorGradeGlobalHue = 0, ColorGradeGlobalSat = 0, ColorGradeGlobalLum = 0, AutoLateralCA = 0, LensProfileEnable = 0, LensManualDistortionAmount = 0, VignetteAmount = 0, DefringePurpleAmount = 0, DefringePurpleHueLo = 30, DefringePurpleHueHi = 70, DefringeGreenAmount = 0, DefringeGreenHueLo = 40, DefringeGreenHueHi = 60, PerspectiveUpright = 0, PerspectiveVertical = 0, PerspectiveHorizontal = 0, PerspectiveRotate = 0, PerspectiveAspect = 0, PerspectiveScale = 100, PerspectiveX = 0, PerspectiveY = 0, GrainAmount = 0, PostCropVignetteAmount = -17, PostCropVignetteMidpoint = 50, PostCropVignetteFeather = 50, PostCropVignetteRoundness = 0, PostCropVignetteStyle = 1, PostCropVignetteHighlightContrast = 0, ShadowTint = 0, RedHue = 0, RedSaturation = 0, GreenHue = 0, GreenSaturation = 0, BlueHue = 0, BlueSaturation = 0, OverrideLookVignette = true, ToneCurveName2012 = "Custom", CameraProfile = "Embedded", CameraProfileDigest = "54650A341B5B5CCAE8442D0B43A92BCE", HasSettings = true, AlreadyApplied = false, ToneCurvePV2012 = {[1] = 0, [2] = 11, [3] = 152, [4] = 106, [5] = 192, [6] = 165, [7] = 255, [8] = 255,}, ToneCurvePV2012Red = {[1] = 0, [2] = 0, [3] = 255, [4] = 255,}, ToneCurvePV2012Green = {[1] = 0, [2] = 0, [3] = 255, [4] = 255,}, ToneCurvePV2012Blue = {[1] = 0, [2] = 0, [3] = 255, [4] = 255,}, Look = {Name = "Modern 08", Amount = 1, Cluster = "Adobe", UUID = "DA1C3775662D6B6A75F8BC2CEEB3724A", SupportsMonochrome = false, Copyright = "© 2018 Adobe Systems, Inc.", Group = {["x-default"] = "Modern",}, Parameters = {Version = "15.2", ProcessVersion = "11.0", ConvertToGrayscale = false, LookTable = "E1095149FDB39D7A057BAB208837E2E1", RGBTable = "6133E9CB13FA32712A5AC579D110EC44", RGBTableAmount = 0.75,},}<answer>

Figure 10: Examples of MMArt data annotated with Chain-of-Thought (CoT) reasoning.

**Input:** 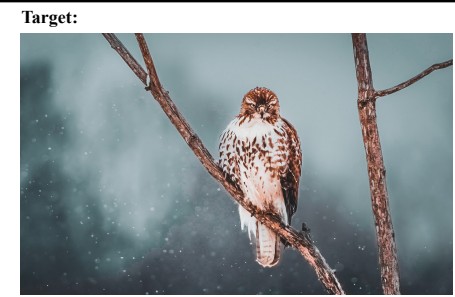

**Target:**

**User Instruction：**
<Image> I want the image to have vibrant colors, highlighting the feather details and overall outline of the the eagle, while keeping the cool-toned atmosphere of the background.

**Answer:**

<answer> PresetType = "Normal", Cluster = "", UUID = "99A42B8B860FFB42A7DCD8096404342D", SupportsAmount2 = true, SupportsAmount = true, SupportsColor = true, SupportsMonochrome = true, SupportsHighDynamicRange = true, SupportsNormalDynamicRange = true, SupportsSceneReferred = true, SupportsOutputReferred = true, RequiresRGBTables = false, CameraModelRestriction = "", Copyright = "", ContactInfo = "", CompatibleVersion = 234881024, ProcessVersion = "11.0", Exposure2012 = 0.25, Contrast2012 = 15, Highlights2012 = -100, Shadows2012 = 75, Whites2012 = 20, Blacks2012 = -10, Texture = 0, Clarity2012 = 10, Dehaze = -5, Vibrance = 20, Saturation = 5, ParametricShadows = 0, ParametricDarks = 0, ParametricLights = 0, ParametricHighlights = 0, ParametricShadowSplit = 25, ParametricMidtoneSplit = 50, ParametricHighlightSplit = 75, Sharpness = 45, SharpenRadius = 1, SharpenDetail = 80, SharpenEdgeMasking = 30, LuminanceSmoothing = 0, ColorNoiseReduction = 25, ColorNoiseReductionDetail = 75, ColorNoiseReductionSmoothness = 25, HueAdjustmentRed = 20, HueAdjustmentOrange = -15, HueAdjustmentYellow = -88, HueAdjustmentGreen = 25, HueAdjustmentAqua = 40, HueAdjustmentBlue = -10, HueAdjustmentPurple = 0, HueAdjustmentMagenta = 0, SaturationAdjustmentRed = 0, SaturationAdjustmentOrange = 12, SaturationAdjustmentYellow = 15, SaturationAdjustmentGreen = -20, SaturationAdjustmentAqua = 0, SaturationAdjustmentBlue = -40, SaturationAdjustmentPurple = 0, SaturationAdjustmentMagenta = 0, LuminanceAdjustmentRed = 45, LuminanceAdjustmentOrange = 15, LuminanceAdjustmentYellow = 10, LuminanceAdjustmentGreen = -30, LuminanceAdjustmentAqua = 0, LuminanceAdjustmentBlue = -20, LuminanceAdjustmentPurple = 0, LuminanceAdjustmentMagenta = 0, SplitToningShadowHue = 219, SplitToningShadowSaturation = 11, SplitToningHighlightHue = 45, SplitToningHighlightSaturation = 14, SplitToningBalance = 0, ColorGradeMidtoneHue = 0, ColorGradeMidtoneSat = 0, ColorGradeShadowLum = 0, ColorGradeMidtoneLum = 0, ColorGradeHighlightLum = 0, ColorGradeBlending = 100, ColorGradeGlobalHue = 0, ColorGradeGlobalSat = 0, ColorGradeGlobalLum = 0, AutoLateralCA = 1, LensProfileEnable = 1, LensManualDistortionAmount = 0, VignetteAmount = 0, DefringePurpleAmount = 2, DefringePurpleHueLo = 30, DefringePurpleHueHi = 70, DefringeGreenAmount = 2, DefringeGreenHueLo = 40, DefringeGreenHueHi = 60, PerspectiveUpright = 0, PerspectiveVertical = 0, PerspectiveHorizontal = 0, PerspectiveRotate = 0, PerspectiveAspect = 0, PerspectiveScale = 100, PerspectiveX = 0, PerspectiveY = 0, GrainAmount = 0, PostCropVignetteAmount = -25, PostCropVignetteMidpoint = 0, PostCropVignetteFeather = 100, PostCropVignetteRoundness = 100, PostCropVignetteStyle = 1, PostCropVignetteHighlightContrast = 0, ShadowTint = 0, RedHue = 0, RedSaturation = 0, GreenHue = 15, GreenSaturation = -15, BlueHue = -8, BlueSaturation = 45, OverrideLookVignette = true, ToneCurveName2012 = "Custom", LensProfileSetup = "LensDefaults", HasSettings = true, ToneCurvePV2012 = {[1] = 0, [2] = 16, [3] = 43, [4] = 40, [5] = 69, [6] = 60, [7] = 127, [8] = 126, [9] = 255, [10] = 255}, ToneCurvePV2012Red = {[1] = 0, [2] = 0, [3] = 255, [4] = 255}, ToneCurvePV2012Green = {[1] = 0, [2] = 0, [3] = 255, [4] = 255}, ToneCurvePV2012Blue = {[1] = 0, [2] = 0, [3] = 255, [4] = 255}, MaskGroupBasedCorrections = {{What = "Correction", CorrectionAmount = 1, CorrectionActive = true, CorrectionName = "mask 1", LocalExposure = 0, LocalSaturation = 0, LocalContrast = 0, LocalClarity = 0, LocalSharpness = 0, LocalBrightness = 0, LocalToningHue = 0, LocalToningSaturation = 0, LocalExposure2012 = -0.049964, LocalContrast2012 = 0, LocalHighlights2012 = 0, LocalShadows2012 = 0, LocalWhites2012 = 0, LocalBlacks2012 = 0, LocalClarity2012 = 0, LocalDehaze = 0, LocalLuminanceNoise = 0, LocalMoire = 0, LocalDefringe = 0, LocalTemperature = 0, LocalTint = 0, LocalTexture = 0, LocalCurveRefineSaturation = 100, CorrectionMasks = {{What = "Mask/Gradient", MaskActive = true, MaskName = "linear gradient1", MaskBlendMode = 0, MaskInverted = false, MaskValue = 1, ZeroX = 0.459395, ZeroY = 0.291666, FullX = 0.458918, FullY = -0.020506}}}, {What = "Correction", CorrectionAmount = 1, CorrectionActive = true, CorrectionName = "mask 2", LocalExposure = 0, LocalSaturation = 0, LocalContrast = 0, LocalClarity = 0, LocalSharpness = 0, LocalBrightness = 0, LocalToningHue = 240, LocalToningSaturation = 0, LocalExposure2012 = -0.036337, LocalContrast2012 = 0, LocalHighlights2012 = 0, LocalShadows2012 = 0, LocalWhites2012 = 0, LocalBlacks2012 = 0, LocalClarity2012 = 0, LocalDehaze = 0, LocalLuminanceNoise = 0, LocalMoire = 0, LocalDefringe = 0, LocalTemperature = 0, LocalTint = 0, LocalTexture = 0, LocalCurveRefineSaturation = 100, CorrectionMasks = {{What = "Mask/Gradient", MaskActive = true, MaskName = "linear gradient1", MaskBlendMode = 0, MaskInverted = false, MaskValue = 1, ZeroX = 0.502379, ZeroY = 0.91026, FullX = 0.502579, FullY = 1.00171}}}, {What = "Correction", CorrectionAmount = 1, CorrectionActive = true, CorrectionName = "Highlight subject", LocalExposure = 0, LocalHue = 0, LocalSaturation = 0.029372, LocalContrast = 0, LocalClarity = 0, LocalSharpness = 0.201033, LocalBrightness = 0, LocalToningHue = 240, LocalToningSaturation = 0, LocalExposure2012 = 0.019615, LocalContrast2012 = 0, LocalHighlights2012 = 0, LocalShadows2012 = 0.047515, LocalWhites2012 = 0.0625, LocalBlacks2012 = -0.012252, LocalClarity2012 = 0.014814, LocalDehaze = 0.005763, LocalLuminanceNoise = 0, LocalMoire = 0, LocalDefringe = 0, LocalTemperature = 0, LocalTint = 0, LocalTexture = 0, LocalCurveRefineSaturation = 100, MainCurve = {[1] = "0,0", [2] = "64,60", [3] = "201,210", [4] = "255,255"}, CorrectionMasks = {{What = "Mask/Image", MaskActive = true, MaskName = "Motiv 1", MaskBlendMode = 0, MaskInverted = false, MaskValue = 1, MaskVersion = 1, MaskSubType = 1, ReferencePoint = "0.500000 0.500000", ErrorReason = 0}}}}<answer>

Figure 11: Data samples from MMArt with standard instructions.

| Input: | Output: |
|---|---|

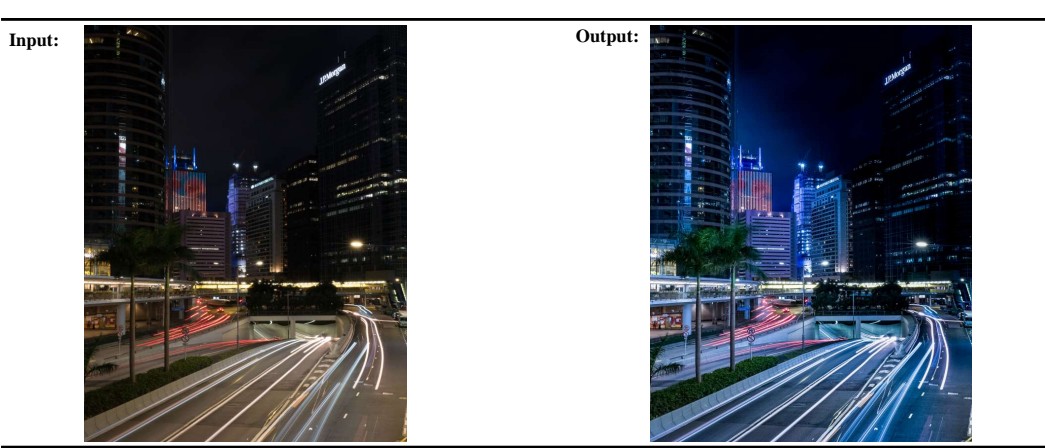

**User Instruction:**
<Image> I want a cyberpunk vibe with neon lights and sharp contrasts for a futuristic city feel.

**JarvisArt:**

<think>Alright, the user wants a cyberpunk vibe with neon lights and sharp contrasts. The image features illuminated signs and light trails at night, perfect for this theme. I'll start by adjusting the white balance to 3200K for a cooler tone. Next, I'll recover highlights in neon lights and open shadows slightly to keep details. Increasing contrast will enhance light-dark differences without harshness. Vibrance and saturation boosts will make colors pop, especially neons. Clarity and dehaze adjustments will sharpen details and cut through haze. Using the "Adobe Color" look with an amount of 1 will complement these changes. Local masks will refine specific areas, ensuring consistency. Checking sharpness and noise reduction will maintain quality. Finally, applying the provided tone curve values will shape brightness and contrast perfectly. This approach captures the futuristic city feel the user desires.<think>

<answer> WhiteBalance = "Custom", Temperature = 3200, Tint = 10, Exposure2012 = 0.96, Contrast2012 = 4, Highlights2012 = -58, Shadows2012 = 58, Whites2012 = 37, Blacks2012 = -13, Clarity2012 = 20, Dehaze = 25, Vibrance = 15, Saturation = 2, ParametricShadowSplit = 25, ParametricMidtoneSplit = 50, ParametricHighlightSplit = 75, Sharpness = 40, SharpenRadius = 1, SharpenDetail = 25, ColorNoiseReduction = 25, ColorNoiseReductionDetail = 50, ColorNoiseReductionSmoothness = 50, ColorGradeBlending = 50, DefringePurpleHueLo = 30, DefringePurpleHueHi = 70, DefringeGreenHueLo = 40, DefringeGreenHueHi = 60, PerspectiveScale = 100, ToneCurveName2012 = "Linear", HasSettings = true, MaskGroupBasedCorrections = {{What = "Correction", CorrectionAmount = 1, CorrectionActive = true, CorrectionName = "Mask 1", LocalExposure2012 = -0.25, LocalHighlights2012 = -0.217391, LocalShadows2012 = -0.25, LocalWhites2012 = -0.25, LocalBlacks2012 = -0.114906, CorrectionMasks = {{What = "Mask/Gradient", MaskActive = true, MaskName = "Linear Gradient 1", MaskBlendMode = 0, MaskInverted = false, MaskValue = 1, ZeroX = 0.630042, ZeroY = 0.503213, FullX = 0.799436, FullY = 0.387017}}}, {What = "Correction", CorrectionAmount = 1, CorrectionActive = true, CorrectionName = "Mask 2", LocalExposure2012 = -0.1, LocalShadows2012 = -0.153061, LocalWhites2012 = -0.153061, LocalBlacks2012 = -0.122449, LocalClarity2012 = 0.082163, LocalDehaze = 0.082163, LocalTexture = 0.102164, CorrectionMasks = {{What = "Mask/Gradient", MaskActive = true, MaskName = "Linear Gradient 1", MaskBlendMode = 0, MaskInverted = false, MaskValue = 1, ZeroX = 0.596549, ZeroY = 0.53203, FullX = 0.799826, FullY = 0.682565}}}}, Look = {Name = "Adobe Color", Amount = 1, UUID = "B952C231111CD8E0ECCF14B86BAA7077", SupportsAmount = false, SupportsMonochrome = false, SupportsOutputReferred = false, Group = {["x-default"] = "Profiles"}, Parameters = {Version = "15.0", ProcessVersion = "11.0", ConvertToGrayscale = false, CameraProfile = "Adobe Standard", LookTable = "E1095149FDB39D7A057BAB208837E2E1", ToneCurvePV2012 = {[1] = 0, [2] = 0, [3] = 22, [4] = 16, [5] = 40, [6] = 35, [7] = 127, [8] = 127, [9] = 224, [10] = 230, [11] = 240, [12] = 246, [13] = 255, [14] = 255}}}<answer>

Figure 12: An example of JarvisArt empowering users to achieve interactive and interpretable editing, transforming their ambiguous intentions into artistic visual outcomes.

| Input: | Output: |
|---|---|

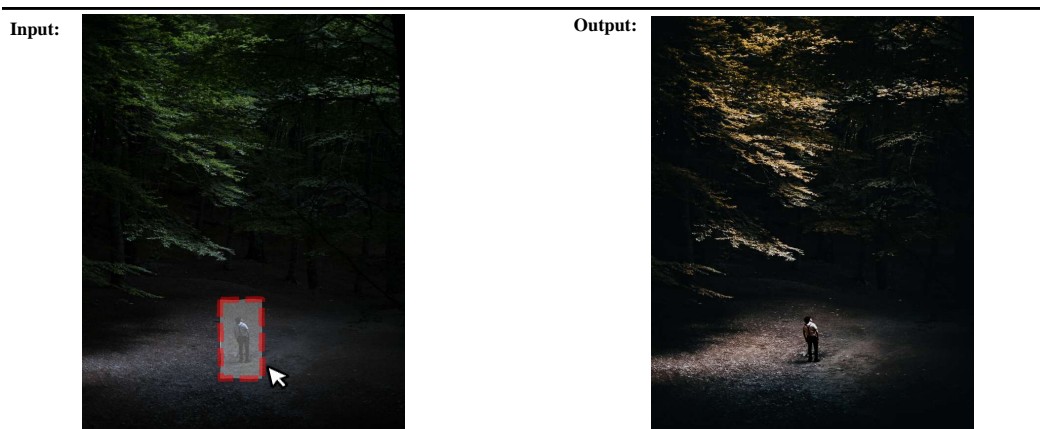

**User Instruction：**

<Image> Make the image evoke a sense of loneliness and introspection, with soft lighting highlighting the solitary figure within the <box>0.446, 0712, 0.540, 0.841</box>.

**JarvisArt:**

<think> Alright, focusing on evoking loneliness and introspection in the image of a solitary figure in a forest. The person is centered at <box>0.446, 0.712, 0.540, 0.841</box>. The dense forest sets a quiet mood. To enhance solitude, I'll adjust lighting around this area. A slight warming with White Balance settings (Incremental Temperature 15, Tint 10) adds melancholy. Exposure2012 at 1.25 brightens the scene, while Contrast2012 at 17 enhances details. Shadows2012 set to 47 reveals more around the person. Vibrance increased by 10 boosts less saturated colors subtly. HueAdjustment values (-100 for Yellow, Green, Purple; -36 for Aqua) harmonize the palette. PostCropVignetteAmount at -21 darkens edges, drawing focus to the center. These adjustments create a poignant, introspective atmosphere.<think>

<answer> PresetType = "Normal", WhiteBalance = "Custom", IncrementalTemperature = 15, IncrementalTint = 10, Exposure2012 = 1.25, Contrast2012 = 17, Highlights2012 = -27, Shadows2012 = 47, Whites2012 = 20, Blacks2012 = -18, Clarity2012 = 15, Dehaze = 10, Vibrance = 10, Saturation = 1, ParametricShadowSplit = 25, ParametricMidtoneSplit = 50, ParametricHighlightSplit = 75, Sharpness = 55, SharpenRadius = 1, SharpenDetail = 25, LuminanceSmoothing = 25, LuminanceNoiseReductionDetail = 50, ColorNoiseReduction = 25, ColorNoiseReductionDetail = 50, ColorNoiseReductionSmoothness = 50, HueAdjustmentYellow = -100, HueAdjustmentGreen = -100, HueAdjustmentAqua = -36, HueAdjustmentBlue = -22, SaturationAdjustmentOrange = -10, SaturationAdjustmentYellow = -10, SaturationAdjustmentGreen = -10, SaturationAdjustmentAqua = -10, SaturationAdjustmentBlue = -10, SaturationAdjustmentPurple = -100, SaturationAdjustmentMagenta = -100, LuminanceAdjustmentOrange = 20, LuminanceAdjustmentYellow = 20, LuminanceAdjustmentGreen = 20, LuminanceAdjustmentAqua = 10, LuminanceAdjustmentBlue = 10, LuminanceAdjustmentPurple = 20, LuminanceAdjustmentMagenta = 20, SplitToningShadowHue = 222, SplitToningShadowSaturation = 13, SplitToningHighlightHue = 10, SplitToningBalance = 22, ColorGradeShadowLum = -25, ColorGradeBlending = 100, AutoLateralCA = 1, DefringePurpleHueLo = 30, DefringePurpleHueHi = 70, DefringeGreenHueLo = 40, DefringeGreenHueHi = 60, PerspectiveScale = 100, PostCropVignetteAmount = -21, PostCropVignetteMidpoint = 50, PostCropVignetteFeather = 50, PostCropVignetteStyle = 1, RedHue = 25, RedSaturation = -8, GreenHue = 45, GreenSaturation = -15, BlueHue = -15, BlueSaturation = 40, CurveRefineSaturation = 100, OverrideLookVignette = true, ToneCurveName2012 = "Custom", HasSettings = true, ToneCurvePV2012 = {[1] = 0, [2] = 28, [3] = 54, [4] = 62, [5] = 176, [6] = 164, [7] = 255, [8] = 224}, ToneCurvePV2012Red = {[1] = 0, [2] = 0, [3] = 40, [4] = 12, [5] = 84, [6] = 58, [7] = 123, [8] = 125, [9] = 186, [10] = 196, [11] = 255, [12] = 255}, ToneCurvePV2012Green = {[1] = 0, [2] = 0, [3] = 40, [4] = 14, [5] = 83, [6] = 59, [7] = 123, [8] = 126, [9] = 172, [10] = 185, [11] = 255, [12] = 255}, ToneCurvePV2012Blue = {[1] = 0, [2] = 0, [3] = 39, [4] = 13, [5] = 87, [6] = 65, [7] = 123, [8] = 124, [9] = 183, [10] = 199, [11] = 255, [12] = 255}, MaskGroupBasedCorrections = {{What = "Correction", CorrectionAmount = 1, CorrectionActive = true, CorrectionName = "Mask 1", LocalSaturation = 0.23, LocalExposure2012 = 0.15, LocalShadows2012 = -0.60648, LocalBlacks2012 = -0.33333, LocalTemperature = 0.32, LocalTint = 0.25, LocalCurveRefineSaturation = 100, CorrectionMasks = {{What = "Mask/Image", MaskActive = true, MaskName = "Person 1", MaskBlendMode = 0, MaskInverted = false, MaskValue = 1, MaskVersion = 1, MaskSubType = 0, ReferencePoint = "0.500000 0.500000", ErrorReason = 0, Gesture = {{What = "Mask/Polygon", MaskActive = true, MaskBlendMode = 0, MaskInverted = false, MaskValue = 1, Points = {{X = 0.44602, Y = 0.712727}, {X = 0.540894, Y = 0.712727}, {X = 0.44602, Y = 0.840727}, {X = 0.540894, Y = 0.840727}}}}}}}}}}<answer>

Figure 13: An example of JarvisArt empowering users to achieve interactive and interpretable editing, transforming their ambiguous intentions into artistic visual outcomes.

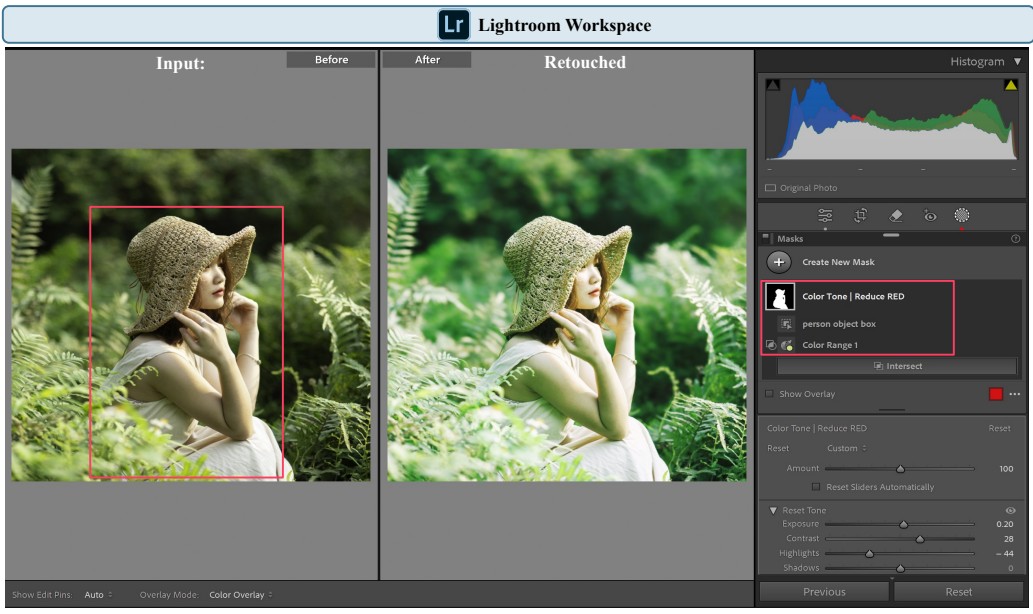

**User Instruction：**
<Image> I want this image to have a soft, misty style. In the <box>[0.3, 0.25, 0.7, 0.6]</box> area, which is the face and upper body of the figure, I wanted the skin color to be smoother and more natural. The overall colors should blend softly, with a warm, green feeling that evokes nostalgia. The foliage in the background should be vibrant but not too strong, creating a serene atmosphere.

**Answer with CoT annotation:**

<think>Alright, I've got this image to edit for a soft, misty style. The user wants smoother skin tones and gentle color blending with a warm, vintage feel, focusing on the person's face and upper body in <box>0.3 0.25 0.7 0.6</box>. The background foliage should stay vibrant but not overpowering. First, I'll adjust the white balance for warmth. Then, apply a slight blur or glow effect for the misty style. For the skin tones, I'll use masks to reduce saturation and smooth textures in the specified area. Next, I'll tweak the tone curve to soften contrasts and warm up highlights and midtones. Background colors will be balanced to complement the foreground without being too intense. Adding subtle film grain can enhance the vintage look. Finally, I'll ensure all elements harmonize for a serene, nostalgic atmosphere. These steps align with the user's vision while preserving the original image's quality.<think>

<answer>PresetType = "Normal", Cluster = "", SupportsAmount2 = true, SupportsAmount = true, SupportsColor = true, SupportsMonochrome = true, SupportsHighDynamicRange = true, SupportsNormalDynamicRange = true, SupportsSceneReferred = true, SupportsOutputReferred = true, ProcessVersion = "15.4", WhiteBalance = "As Shot", ParametricShadowSplit = 25, ParametricMidtoneSplit = 50, ParametricHighlightSplit = 75, ColorGradeBlending = 50, DefringePurpleHueLo = 30, DefringePurpleHueHi = 70, DefringeGreenHueLo = 40, DefringeGreenHueHi = 60, PerspectiveScale = 100, ToneCurveName2012 = "Linear", HasSettings = true, Look = { Name = "Adobe Color", Amount = 1, Stubbed = true, Parameters = { CompatibleVersion = 251920384, ProcessVersion = "15.4", CurveRefineSaturation = 50, ConvertToGrayscale = false, RGBTable = "F4B5250D16A58B460C36CEC16C0627D2", ToneCurvePV2012 = { [1] = 0, [2] = 0, [3] = 16, [4] = 24, [5] = 64, [6] = 72, [7] = 128, [8] = 128, [9] = 192, [10] = 176, [11] = 244, [12] = 234, [13] = 255, [14] = 255 }, ToneCurvePV2012Red = { [1] = 0, [2] = 0, [3] = 255, [4] = 255 }, ToneCurvePV2012Green = { [1] = 0, [2] = 0, [3] = 255, [4] = 255 }, ToneCurvePV2012Blue = { [1] = 0, [2] = 0, [3] = 255, [4] = 255 } } }, LensBlur = { Version = 1, Active = true, ImageOrientation = 0, FocalRange = "0 0 100 100", BlurAmount = 50, BokehShape = 0, BokehShapeDetail = 0, HighlightsThreshold = 50, HighlightsBoost = 50, CatEyeAmount = 0, CatEyeScale = 100, BokehAspect = 0, BokehRotation = 0, SphericalAberration = 0 }, MaskGroupBasedCorrections = { { What = "Correction", CorrectionAmount = 1, CorrectionActive = true, CorrectionName = "Color Tone | Reduce RED", LocalToningHue = 119, LocalToningSaturation = 0.1, LocalTint = -0.05, LocalCurveRefineSaturation = 100, CorrectionMasks = { { What = "Mask/RangeMask", MaskActive = true, MaskName = "Color Range 1", MaskBlendMode = 1, MaskInverted = true, MaskValue = 0, CorrectionRangeMask = { Version = 3, Type = 1, ColorAmount = 0.494949, Invert = true, SampleType = 0, PointModels = {"0.595958 0.896660 0.597516 0.480823 0.955119 0"} } }, { What = "Mask/Image", MaskActive = true, MaskName = "person object box", MaskBlendMode = 1, MaskInverted = true, MaskValue = 0, MaskVersion = 1, MaskSubType = 0, ReferencePoint = "0.500000 0.500000", ErrorReason = 0, Gesture = { { What = "Mask/Polygon", MaskActive = true, MaskBlendMode = 0, MaskInverted = false, MaskValue = 1, Points = { { X = 0.3, Y = 0.25 }, { X = 0.7, Y = 0.25 }, { X = 0.3, Y = 0.6 }, { X = 0.7, Y = 0.6 } } } } } } } }
<answer>

Figure 14: Editing results with JarvisArt are generated under complex prompts, with all retouching operations performed in a Lightroom environment, allowing for iterative adjustments.

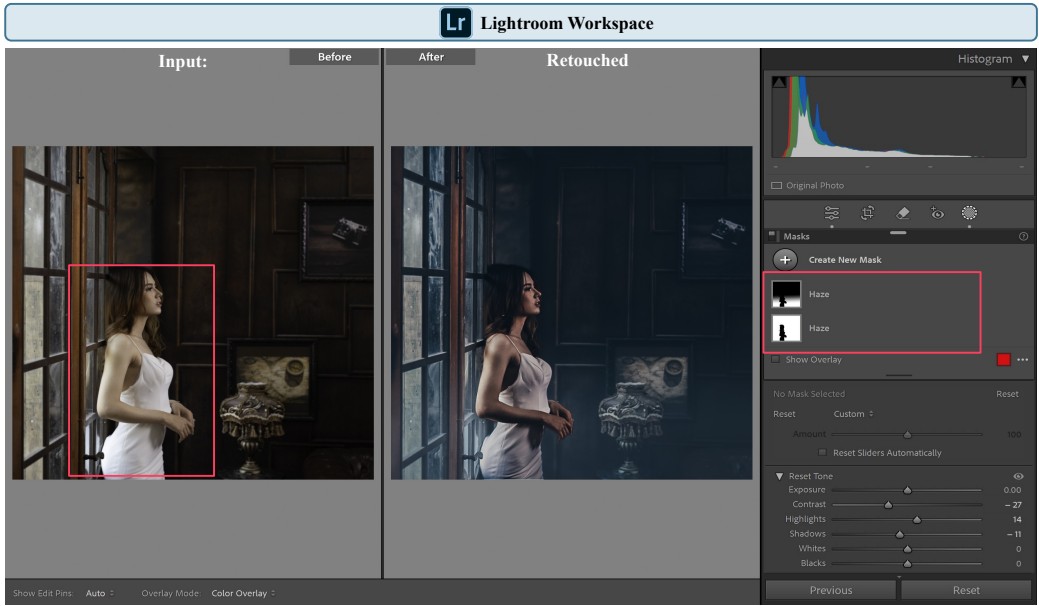

**User Instruction:**
<Image> My goal was to create an atmospheric, nostalgic look. For the area <box>[0.3, 0.25, 0.7, 0.6]</box> around her face and shoulders, I wanted smoother textures and a subtle haze effect. The entire image should evoke a sense of timelessness with muted colors and enhanced contrast, making it feel like a classic portrait.

**Answer with CoT annotation:**

<think>Alright, let's dive into this image. The user wants a nostalgic feel with smooth textures and a subtle haze around the person's face and shoulders in <box>0.3 0.25 0.7 0.6</box>. We'll warm the white balance to 5185K for that vintage vibe. A slight overexposure of 0.23 brightens the subject, while contrast at 7 adds depth. Highlights reduced by -71 and shadows boosted by 50 preserve detail. Texture at 25 and clarity at 14 enhance details globally, but local dehaze softens the specified area. Vibrance up by 20 enriches colors subtly. Split toning with shadow hue at 229 and highlight hue at 48 adds a vintage touch. A dark vignette draws focus to the center. These tweaks balance global adjustments with targeted corrections, achieving the classic portrait look the user desires.<think>

<answer>WhiteBalance = "Custom", Temperature = 5185, Tint = 10, Exposure2012 = 0.23, Contrast2012 = 7, Highlights2012 = -71, Shadows2012 = 50, Whites2012 = 12, Blacks2012 = -18, Texture = 25, Clarity2012 = 14, Dehaze = 11, Vibrance = 20, Saturation = 3, ParametricShadowSplit = 25, ParametricMidtoneSplit = 50, ParametricHighlightSplit = 75, Sharpness = 40, SharpenRadius = 1, SharpenDetail = 25, LuminanceSmoothing = 13, LuminanceNoiseReductionDetail = 50, ColorNoiseReduction = 25, ColorNoiseReductionDetail = 50, ColorNoiseReductionSmoothness = 50, HueAdjustmentOrange = -11, HueAdjustmentYellow = -3, SaturationAdjustmentOrange = -20, SaturationAdjustmentYellow = -50, SaturationAdjustmentBlue = -31, LuminanceAdjustmentOrange = -30, LuminanceAdjustmentYellow = 12, LuminanceAdjustmentAqua = -12, LuminanceAdjustmentBlue = -30, SplitToningShadowHue = 229, SplitToningShadowSaturation = 13, SplitToningHighlightHue = 48, SplitToningHighlightSaturation = 10, ColorGradeBlending = 100, AutoLateralCA = 1, DefringePurpleHueLo = 30, DefringePurpleHueHi = 70, DefringeGreenHueLo = 40, DefringeGreenHueHi = 60, PerspectiveScale = 100, PostCropVignetteAmount = -11, PostCropVignetteMidpoint = 50, PostCropVignetteFeather = 50, PostCropVignetteStyle = 1, RedHue = 30, RedSaturation = -10, GreenHue = 40, GreenSaturation = -10, BlueHue = -25, BlueSaturation = -10, OverrideLookVignette = true, ToneCurveName2012 = "Custom", HasSettings = true, ToneCurvePV2012 = { [1] = 0, [2] = 29, [3] = 48, [4] = 48, [5] = 127, [6] = 126, [7] = 190, [8] = 192, [9] = 255, [10] = 255 }, ToneCurvePV2012Red = { [1] = 0, [2] = 0, [3] = 47, [4] = 27, [5] = 115, [6] = 125, [7] = 159, [8] = 173, [9] = 255, [10] = 255 }, ToneCurvePV2012Green = { [1] = 0, [2] = 0, [3] = 43, [4] = 25, [5] = 113, [6] = 121, [7] = 160, [8] = 174, [9] = 255, [10] = 255 }, ToneCurvePV2012Blue = { [1] = 0, [2] = 0, [3] = 39, [4] = 25, [5] = 113, [6] = 124, [7] = 159, [8] = 174, [9] = 255, [10] = 255 }, MaskGroupBasedCorrections = { { What = "Correction", CorrectionAmount = 1, CorrectionActive = true, CorrectionName = "Haze", LocalToningHue = 240, LocalBlacks2012 = 0.0675, LocalClarity2012 = -0.198322, LocalDehaze = -0.208022, LocalTemperature = -0.1024, LocalTint = -0.09851, LocalTexture = -0.250785, CorrectionMasks = { { What = "Mask/Image", MaskActive = true, MaskName = "Subject 1", MaskBlendMode = 0, MaskInverted = true, MaskValue = 1, MaskVersion = 1, MaskSubType = 1, ReferencePoint = "0.306641 0.521994", WholeImageArea = "0/1,0/1,1707/1,2560/1", Origin = "0,531", ModelVersion = 234881976 } } }, { What = "Correction", CorrectionAmount = 1, CorrectionActive = true, CorrectionName = "Haze", LocalToningHue = 240, LocalBlacks2012 = 0.175759, LocalClarity2012 = -0.284554, LocalDehaze = -0.364241, LocalTemperature = -0.163366, LocalTint = -0.188755, LocalTexture = -0.288385, CorrectionMasks = { { What = "Mask/Gradient", MaskActive = true, MaskName = "Linear Gradient 1", MaskBlendMode = 0, MaskInverted = false, MaskValue = 1, ZeroX = 0.426926, ZeroY = 0.47269, FullX = 0.44185, FullY = 0.872828 }, { What = "Mask/Image", MaskActive = true, MaskName = "Subject 1", MaskBlendMode = 1, MaskInverted = false, MaskValue = 0, MaskVersion = 1, MaskSubType = 1, ReferencePoint = "0.306641 0.521994", WholeImageArea = "0/1,0/1,1707/1,2560/1", Origin = "0,531", ModelVersion = 234881976 } } } }
<answer>

Figure 15: Editing results with JarvisArt are generated under complex prompts, with all retouching operations performed in a Lightroom environment, allowing for iterative adjustments.

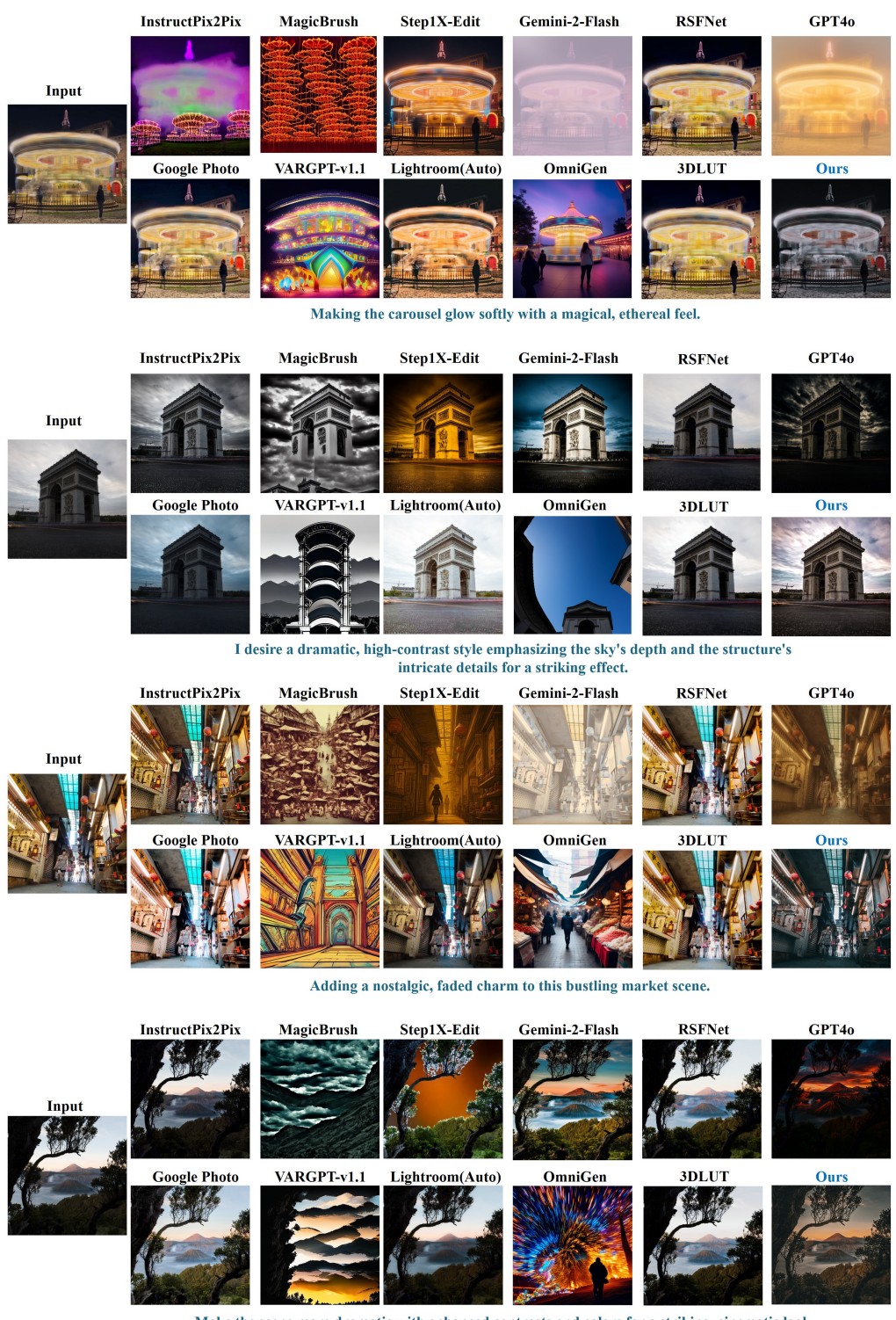

Figure 16: Visual comparisons of all state-of-the-art editing methods alongside two automatic retouching modes from commercial software.

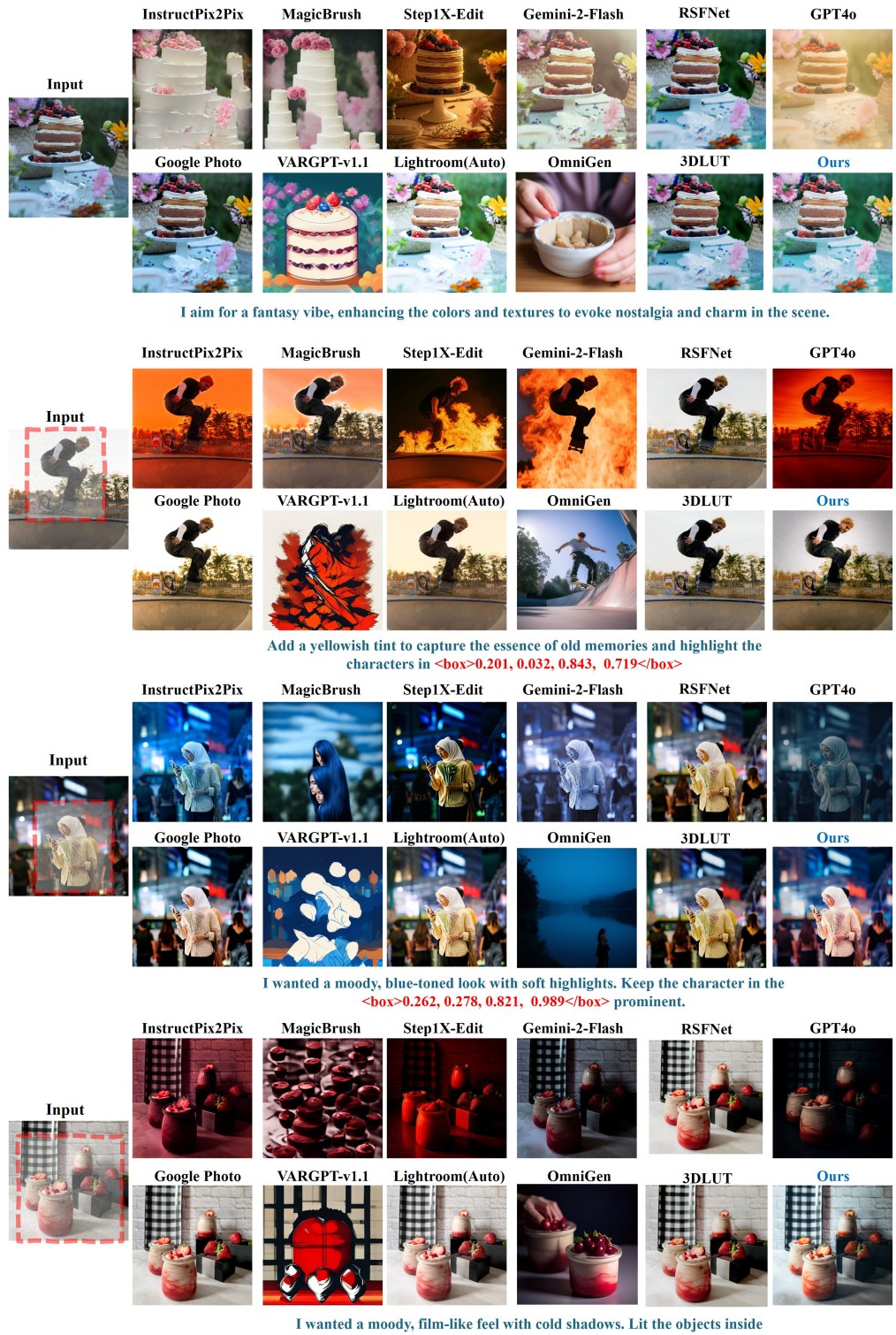

Figure 17: Visual comparisons of all state-of-the-art editing methods alongside two automatic retouching modes from commercial software.

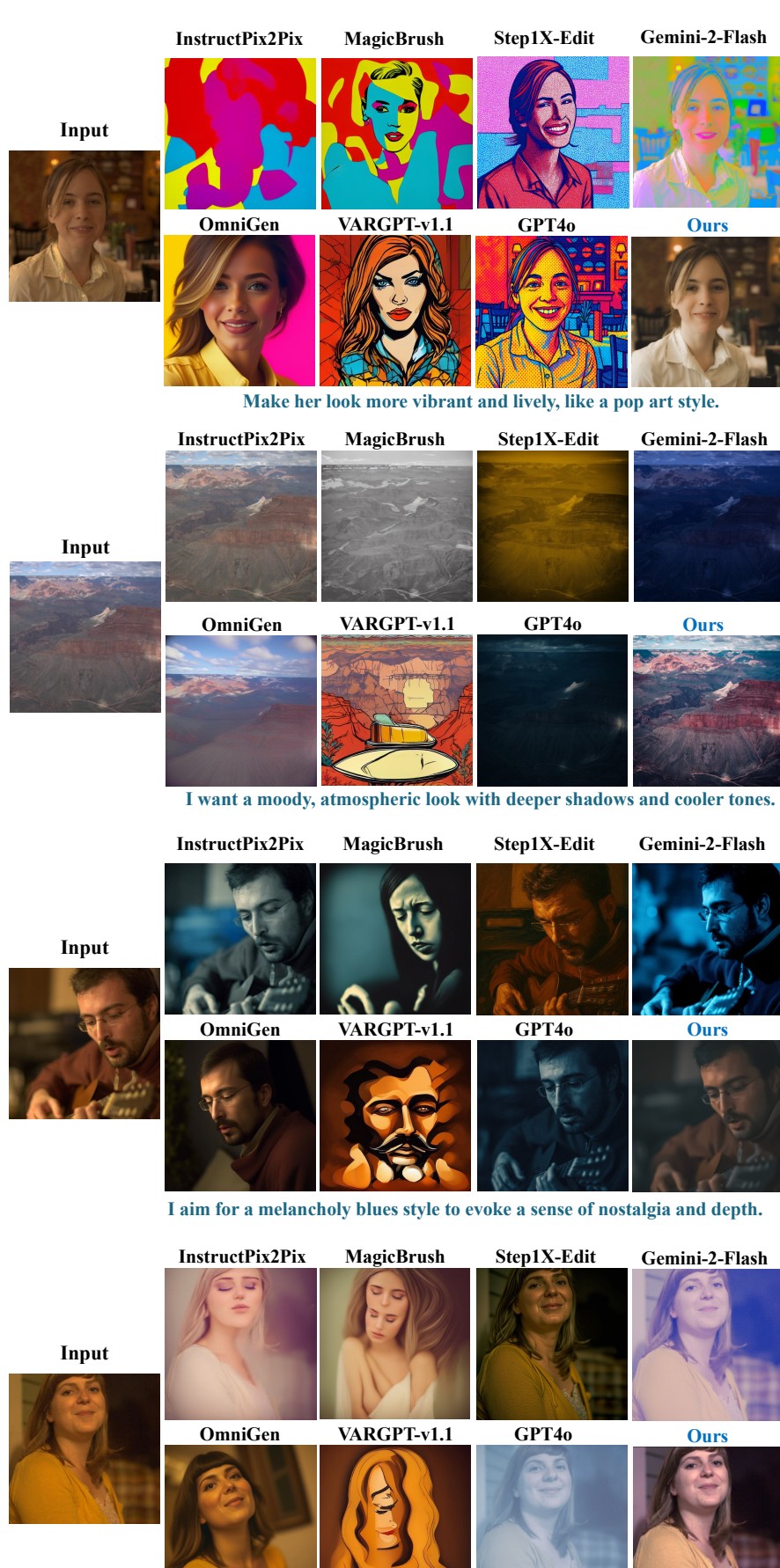

Figure 18: Visual comparisons of all instruction-based editing methods on MIT-FiveK [3].

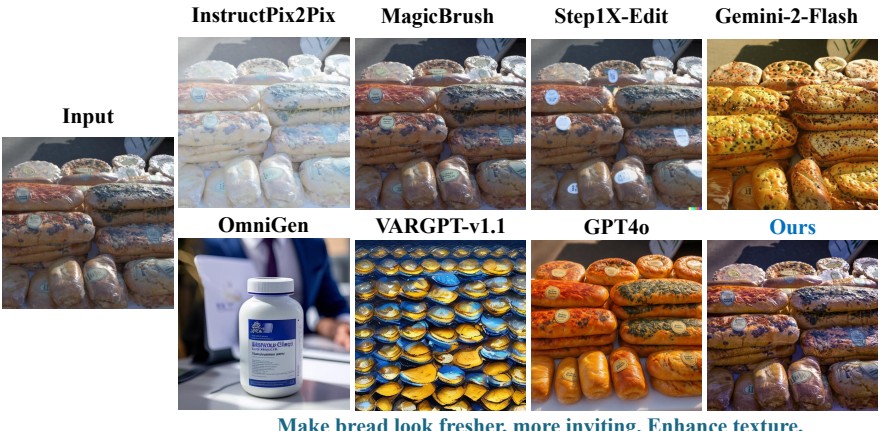

**Make bread look fresher, more inviting. Enhance texture, warmth for cozy bakery vibe.**

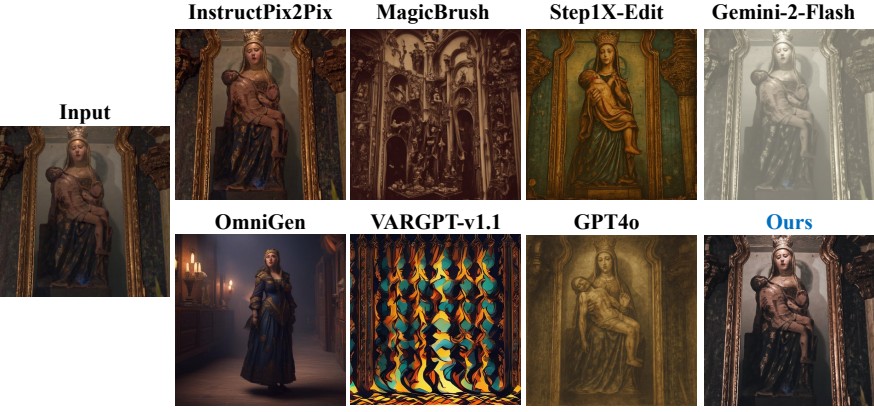

**I want a vintage fantasy style to evoke nostalgia and history.**

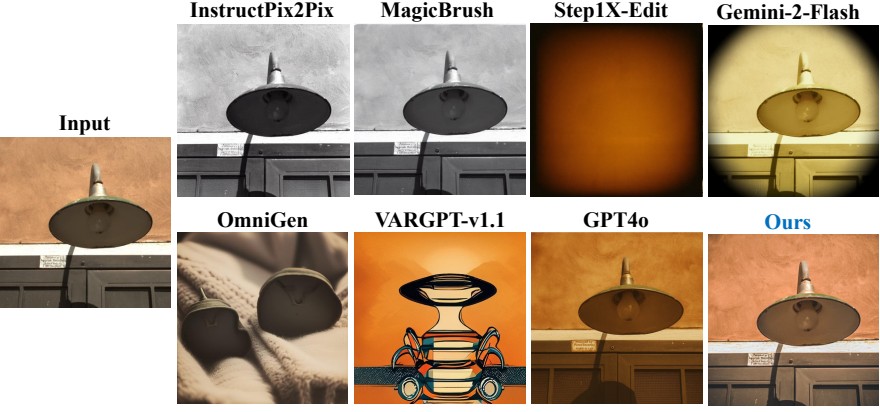

**Make it look warmer and cozier, like a vintage photo.**

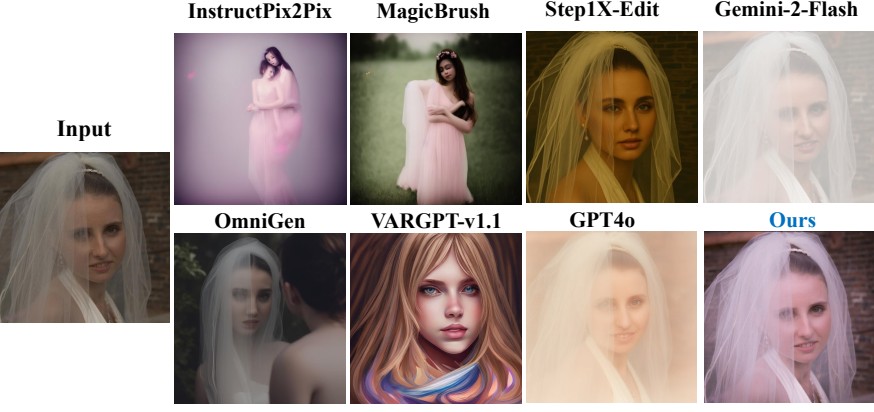

**Dreamy Haze Style to soften the image, making it feel more romantic and ethereal.**

Figure 19: Visual comparisons of all instruction-based editing methods on MIT-FiveK [3].

**Prompt for MLLM-based SC, PQ, and O**

prompt_sc_pq_scene = """

You are a post-production specialist with expertise in enhancing photographic imagery through advanced digital editing techniques. We now need your help to evaluate the performance of an AI-powered image post-editing tool for photography.

INPUTS:
1. Two images will be provided: The first being the original photographic image and the second being an edited version of the first.
2. The editing instruction will be provided: The post-editing needs of photographic images expressed by users with no image processing knowledge.

METRICS (From scale 0 to 10):
User Instruction Satisfaction Score: A score from 0 to 10 will be given based on how well the edits follow the user's instructions.
- Users typically have both global and local editing requirements when working with photographic images. Therefore, this score should be evaluated holistically, taking into account the user's needs for both local and global adjustments, with equal importance given to each.
- 0 indicates that the edited image does not follow the editing instruction at all.
- 10 indicates that the edited image follow the editing instruction text perfectly.
Content Consistency Score: A second score from 0 to 10 will rate the consistency of image content before and after editing.
- Need to compare before and after images to assess content consistency.
- The edited image should maintain consistency in key visual elements such as the shape of landscapes, human figures (including posture, gender, and appearance), building structures, and other important features.
- The edited image needs to maintain the consistency of local details, such as letters on clothes, textures of buildings, etc.
- 0 indicates that the content of the image before and after editing is completely inconsistent.
- 10 indicates that the content of the edited image is exactly the same as the original image.

You will have to give your output in this way (Keep your reasoning concise and short.):
{
"score" : [...],
"reasoning" : "..."
}
Put the score in a list such that output score = [score1, score2], where 'score1' evaluates the User Instruction Satisfaction Score and 'score2' evaluates the Content Consistency Score.
Editing instruction: <The editing instruction of user>
"""

prompt_sc_pq_region = """

You are a post-production specialist with expertise in enhancing photographic imagery through advanced digital editing techniques. We now need your help to evaluate the performance of an AI-powered image post-editing tool for photography.

INPUTS:
1. Two images will be provided: The first being the original photographic image and the second being an edited version of the first.
2. The editing instruction will be provided: The post-editing needs of photographic images expressed by users with no image processing knowledge.

METRICS (From scale 0 to 10):
User Instruction Satisfaction Score: A score from 0 to 10 will be given based on how well the edits follow the user's instructions.
- Users typically have both global and local editing requirements when working with photographic images. Therefore, this score should be evaluated holistically, taking into account the user's needs for both local and global adjustments, with equal importance given to each.
- 0 indicates that the edited image does not follow the editing instruction at all.
- 10 indicates that the edited image follow the editing instruction text perfectly.
Content Consistency Score: A second score from 0 to 10 will rate the consistency of image content before and after editing.
- Need to compare before and after images to assess content consistency.
- The edited image should maintain consistency in key visual elements such as the shape of landscapes, human figures (including posture, gender, and appearance), building structures, and other important features.
- The edited image needs to maintain the consistency of local details, such as letters on clothes, textures of buildings, etc.
- 0 indicates that the content of the image before and after editing is completely inconsistent.
- 10 indicates that the content of the edited image is exactly the same as the original image.

IMPORTANT NOTEs:
1. The focus of this assessment is the performance of AI-enabled post-processing tools for photographic images in response to users' local modification instructions.
2. The two images are the same localized areas of interest selected by the user in the original and edited images.
3. User editing instructions involve both global adjustments to the entire image and localized edits to specific regions of interest.

You will have to give your output in this way (Keep your reasoning concise and short.):
{
"score" : [...],
"reasoning" : "..."
}
Put the score in a list such that output score = [score1, score2], where 'score1' evaluates the User Instruction Satisfaction Score and 'score2' evaluates the Content Consistency Score.
Editing instruction: <The editing instruction of user>
"""

### Editing Evaluation Processing:
Scene-level editing evaluation: prompt_sc_pq_scene
Region-level editing evaluation: prompt_sc_pq_region

Figure 20: Prompt for MLLM-based metrics (SC, PQ) from scene-level and region-level.

# Aesthetic Preset Recommendation Expert

You are a professional image aesthetic preset recommendation expert, skilled at analyzing image content and providing the best preset combination suggestions. You can identify the main subjects in images, and based on their categories and characteristics, precisely match suitable global and local presets from the preset library, offering users diverse and high-quality image optimization solutions.

## Workflow

You will automatically match applicable image optimization presets based on the provided image file, detected subject box information, and preset library, and output the recommendations.

### Input:

1. Source image.
2. Detected subject box information (category and confidence):
Examples:
[{
"category": "person",
"confidence": 0.95,
"box": [0.2, 0.3, 0.6, 0.8]
},…]

**3. Preset library (divided into four main categories: portrait, landscape, street, and food, each containing global and local presets):**
"global": [
{"id": "PERSON-G1",
"name": "Portrait-B&W Background-Colored Subject",
"function": "Preserves red-orange subject, desaturates background to black and white, high contrast to highlight people. Suitable for weddings, portrait close-ups, not suitable for colorful scenes or photos where landscape needs to be emphasized."},
// This includes a total of 76 global presets across four categories:
// 1. Portrait (PERSON-G series): 24 presets covering Japanese style, film simulation, fresh dreamy effects, and various portrait styles
// 2. Landscape (SCENERY-G series): 10 presets including glacier waterfalls, golden sunset, cinematic city views, and other natural landscape styles
// 3. Street (STREET-G series): 12 presets including urban day scenes, night scenes, cyberpunk, and various urban street photography styles
// 4. Food (STILL-Life-G series): 6 presets including bright transparent, cinematic, warm tempting, and various food photography styles
// Each preset has a unique ID, name, and function description detailing the effect, suitable scenarios, and unsuitable scenarios
{...},]
"local": [
{"id": "PERSON-L1",
"name": "Skin Brightening-Facial Contour Enhancement",
"function": "This is a professional portrait local adjustment preset that achieves dimensionality through fine-tuning of eyes (sharpness +0.21), eyebrows (exposure -0.1), and skin (exposure +0.14/contrast -0.2). Specifically enhances iris sharpness and brightens eye whites (exposure +0.04), while softening skin texture (texture -0.11) and reducing noise (-1). Suitable for commercial portraits and close-up portraits, highlighting facial features' dimensionality"
},
// This includes a total of 53 local presets across several categories:
// 1. Portrait (PERSON-L series): 13 presets covering skin beautification, eye enhancement, and various portrait local adjustments
// 2. Sky (ALL-L series SKY01 group): 22 presets including sky brightening, darkening, dramatic effects, etc.
// 3. Subject emphasis (ALL-L series SUBJECT group): 13 presets including subject brightening, background weakening, etc.
// 4. Lighting effects (ALL-L series LIGHT group): 18 presets including center stage, foreground darkening, fog effects, etc.
// Each preset has a unique ID, name, and function description, with some presets having group information
{...},]

### Output Format:

{ "recommendations-1":
{"target_class": "person",
"confidence": 0.85,
"global_presets":
{"id": "G1",
"name": "Natural Skin Tone Optimization" }
,
"local_presets": [
{"id": "L1",
"name": "Female Skin Softening"},
{"id": "L3",
"name": "Light Source Enhancement"},],
"local_apply_range": [x1,y1,x2,y2],
"reasoning": "Detected high confidence portrait (85%), recommending global skin tone optimization with local skin softening and light source enhancement"},
{...},}

## Working Rules
### Special Instructions
1. Subject identification rule: Only when the detected subject confidence is greater than or equal to 0.85 will the subject be recognized as the current image's main scene type, and recommendations will be generated accordingly.
2. Preset matching principle: All recommended presets must strictly match the detected subject box information. For example, when the subject box only contains portrait category targets, presets for landscape, street, or food categories should not be recommended to ensure precision of processing effects.
3. Preset combination strategy: Recommendation plans should prioritize the combined use of global and local presets to achieve the best overall effect.
4. ID naming convention:
- Global presets: "{category}-G{number}" (e.g., PERSON-G1)
- Local presets: "{category}-L{number}" (e.g., PERSON-L2)
- Category codes include: PERSON (portrait), SCENERY (landscape), STREET (street), FOOD (food), ALL (universal)
- ALL category indicates the preset is applicable to all scene types
5. Application range definition:
- Global presets: Applied to the entire image
- Local presets: Use detection box precise coordinates [x1,y1,x2,y2] to define the application area
6. Multi-subject processing strategy: Regardless of whether there is a single or multiple subjects in the image, the system must generate multiple different recommendation plans. When there are multiple subjects, sort by confidence from high to low; when there is only a single subject, diverse processing plans should be generated around that subject. All recommendation plans must ensure diversity and non-repetition to provide rich editing options.
7. Overall effect priority: When evaluating preset combinations, consider the overall effect rather than the effect of a single step. For example, for photos with insufficient light, even if global preset G1 alone is not effective, it may produce an ideal effect when combined with local brightening preset L1, so comprehensive consideration of the final presentation is needed.
8. Recommendation plan naming convention: Use "recommendations-{number}" format (e.g., recommendations-1).
9. Preset combination limitations:
- Each recommendation plan can only include one global preset
- Try to select diverse recommendations, for example, if recommendations-1 has a yellowish portrait tone, recommendations-2 should choose a cool tone
- Can include multiple different local presets, but all local presets must be applied to the same area (same apply_range); if you think the global preset is sufficient, local presets are not needed
- Local presets must not be used repeatedly
- Strictly avoid conflicting preset combinations in recommendation plans, for example:
* Should not simultaneously include presets for females and presets for males
* Should not simultaneously include presets with mutually canceling effects (such as using both saturation enhancement and saturation reduction presets)
- Ensure all presets are logically compatible with each other, jointly serving to enhance the overall effect of the image
10. System output requirements: The system should directly generate recommendation plans in standard JSON format, without any additional text explanations or interpretive content, ensuring the output results can be directly parsed and used by programs.
11. Exposure balance principle: When recommending presets, the system should fully consider the current image's exposure value, intelligently selecting preset combinations that can balance image brightness, ensuring the final processing result is neither overexposed nor underexposed, thereby guaranteeing image detail retention and visual comfort.
12. Portrait exposure protection principle: For portrait subjects, the system must pay special attention to ensure preset combinations do not result in overexposure or excessive darkness in global or local facial areas after editing. For images that are already bright, presets that significantly increase brightness should be avoided; for images with already high contrast, presets that further enhance contrast should be avoided to prevent facial areas from becoming black or losing detail. When evaluating recommendation plans, the overall impact of global and local presets on portrait subjects must be comprehensively considered to ensure facial details are preserved, avoiding highlight overflow, facial overexposure, or excessive shadows, thereby guaranteeing natural texture, balanced tones, and detail expression in portraits.
13. Recommendation plan quantity principle: The system must generate 5 different recommendation plans.
'''

Figure 21: Role-playing prompt for preset recommendation.

**Role-Playing Prompt for Retouching Instructions （Simulating a Professional User）**

prompt_base_w_coordination = """
Given an original image, its edited version, and the editing configuration parameters, Relative coordinates of the region of interest(optional), analyze the visual changes and adjustments made. Infer the user's original editing goal and describe their vague intention in simple, everyday language as a direct statement from a non-technical user's perspective.

Special Notes:
1. Simulate a real client's voice to articulate service needs in casual business English. Write as if you're the actual decision-maker explaining requirements to a trusted partner, not just filling a template.
2. Focus on translating technical parameters (e.g., brightness +20, sharpening +15) into relatable visual outcomes without mentioning numerical values or technical terms.
3. Answer the output in a paragraph of 40 words or less.
4. If the user provides coordinates for a region of interest, ensure that the output identifies the category of the region of interest and includes the coordinates enclosed in <box></box> tags for annotation. The response should explicitly state the identified category and clearly mark the coordinates within the specified format.
5. The system should adaptively distribute inferred local adjustment intentions throughout the description text.
6. Need to mimic the user's intentions for global image modification, not just local modification intentions.

Example of the user's original editing intention:
1. I want this to look like it's straight out of a movie—more depth, atmosphere, and that filmic texture. In the region <box>[0.4419, 0.1982, 0.8307, 0.9967]</box>, I wanted the person to look clearer and the outfit details to pop. Make the skin smoother and heighten the colors, especially the jacket and surrounded area.
2. In the region <box>[0.1242, 0.699, 0.7453, 1.0000]</box>, I want my face to look brighter and my eyes to stand out more. Can you make the skin smoother and the hair darker? Let's enhance the entire image while making sure the colors don't vibrate too much.

Output requirements:
- It is necessary to provide TWO possible intentions of user to modify the image.
- Two possible user intentions are independent of each other.
- The following output format MUST be strictly followed:
{
"user want 1": "< The first intentions of user >",
"user want 2": "< The second intentions of user >"
}

Relative coordinates of the region of interest: <region_of_interest_coordinates>
The configuration details are as follows: <corresponding_lua_file>
"""

prompt_base_wo_coordination = """
Given an original image, its edited version, and the editing configuration parameters, Relative coordinates of the region of interest(optional), analyze the visual changes and adjustments made. Infer the user's original editing goal and describe their vague intention in simple, everyday language as a direct statement from a non-technical user's perspective.

Special Notes:
1. Simulate a real client's voice to articulate service needs in casual business English. Write as if you're the actual decision-maker explaining requirements to a trusted partner, not just filling a template.
2. Focus on translating technical parameters (e.g., brightness +20, sharpening +15) into relatable visual outcomes without mentioning numerical values or technical terms.
3. Answer the output in a paragraph of 40 words or less.
4. The system should adaptively distribute inferred local adjustment intentions throughout the description text.

Output requirements:
- It is necessary to provide TWO possible intentions of user to modify the image.
- Two possible user intentions are independent of each other.
- The following output format MUST be strictly followed:
{
"user want 1": "< The first intentions of user >",
"user want 2": "< The second intentions of user >"
}

The configuration details are as follows: <corresponding_lua_file>
"""

### Image Type Processing:
Scene-level instruction generation: prompt_base_wo_coordination
Region-level instruction generation: prompt_base_w_coordination

Figure 22: Prompt for simulating the professional user instructions

Style_expressions = """
The style of editing the image that customers might express:
- Dreamy Haze Style: Soft focus, low contrast, soft tones, hazy halo -> Suitable for wedding photos, portraits, emotional expressions, etc.
- Melancholy Blues Style: Blue-grey tones, cold light, high shadows, emotional composition -> Suitable for street photography, portraits, storytelling, etc.
- Burning Emotion Style: Highly saturated orange-red, interlaced light and shadow, and a sense of restlessness -> Suitable for stage performances, music vision, dynamic moments, etc.
- Minimal Ethereal Style: Minimalist composition, blank space, soft filter, sense of tranquility -> Suitable for meditation, Zen, lifestyle content, etc.
- Futuristic Anxiety Style: Cold metallic tones, sharp lines, high contrast, digital noise -> Suitable for science fiction, technology products, cyberpunk style, etc.
- Vintage Fantasy Style: Yellowed/faded tones, graininess, old film texture -> Suitable for nostalgic advertising, historical narrative, illustration style, etc.
- Glitch Art Style: Digital glitch effect, color distortion, pixel distortion -> Suitable for electronic music, avant-garde fashion, Internet culture, etc.
- Liquid Abstraction Style: Strong sense of fluidity, color fusion, similar to watercolor/oil paint -> Suitable for art posters, packaging design, visual impact, etc.
- Negative Film Look Style: Similar to reversal film effect, reverse color, high contrast -> Suitable for documentary photography, street photography, alternative portraits, etc.
- Light Painting Abstraction Style: Emphasize light trails, blur, and dynamic smear -> Suitable for night photography, light shows, art installations, etc.
- Relief Texture Style: Pixel-level detail enhancement, three-dimensional sculpture, and outstanding material -> Suitable for sculptures, industrial design, fabric display, etc.
- Grain & Noise Style: Simulate film noise, rough texture, and handmade feel -> Suitable for documentary style, retro photography, documentary content, etc.
- Translucent Overlay Style: Multi-layer image overlay, semi-transparent channel, dreamy interlacing -> Suitable for conceptual photography, digital painting, mixed media, etc.
- Impressionist Style: Strong brushstrokes, vibrant colors, and soft edges -> Suitable for landscape photography, natural themes, artistic re-creation, etc.
- Expressionist Color Style: High contrast, exaggerated colors, emotional rendering -> Suitable for close-ups, dramatic photography, psychological expression, etc.
- Pop Art Style: Highly saturated color blocks, comic dots, and repeated patterns -> Suitable for celebrity portraits, pop culture, commercials, etc.
- Abstract Expressionism Style: Free brushstrokes, splashing feeling, emotional explosion -> Suitable for art photography, digital painting, emotional expression, etc.
- Ink Wash Style: Black and white gradient, white space, and fluidity of brush and ink -> Suitable for oriental culture, traditional elements, calligraphy, etc.
- Oil Painting Effect Style: Brushstroke simulation, thick painting, frame border -> Suitable for art exhibitions, high-end portraits, custom printing, etc.
- Watercolor Style: Fresh and transparent, edge diffusion, paper texture -> Suitable for children's photography, illustration, handicraft brands, etc.
- Pencil Sketch Style: Line outline, grayscale, hand-painted texture -> Suitable for teaching materials, character sketches, sketch displays, etc.
- Neon Glow Style: Neon texture, glowing edges, dark background -> Suitable for nightclub promotion, trendy clothing, urban photography, etc.
- Metallic Texture Style: High reflectivity, cold hard tone, metallic luster -> Suitable for luxury goods, electronic products, fashion blockbusters, etc.
- Motion Blur Style: Movement trajectory, speed sense, direction guidance -> Suitable for sports photography, car advertising, action scenes, etc.
- Aurora Vision Style: Flowing light bands, gradient changes of cold and warm, mysterious atmosphere -> Suitable for natural scenery, starry sky photography, meditation applications, etc.
- Hamada Hideki Style: Fresh, soft, natural and emotional -> Suitable for photos with a girlish or youthful feel, etc.
- Fujifilm Style: Make portraits, landscapes or daily photos present natural, delicate and emotional color expression -> Suitable for style requirements such as "movie feel", "Japanese fresh style" or "retro atmosphere"
- Japanese Style: It has visual characteristics such as freshness, naturalness, emotionality, low saturation, and soft tones. -> Suitable for portraits, life records, travel photography, etc.
- Kodak Portra Style: With natural skin tones, warm tones, delicate transitions and rich layering, it can make digital photos look like "old-school but authentic" film. -> Suitable for those who pursue naturalness, warmth, storytelling, etc.
"""

Figure 23: Prompt for simulating the casual user instructions.

**Prompt for Generating Initial CoT Annotations**

# Prompt for Generating Initial Long Chain-of-Thought with Coordinate Information
"""Please analyze the provided before/after images, user requirements, Relative coordinates of the region of interest(optional), and configuration file to generate a detailed adjustment workflow. Although the configuration file is recognized, the response should avoid explicitly stating that the adjusted parameters were derived from it. The tone should convey a sense of expert judgment and reasoned analysis.
1. Technical Breakdown: Specify the tool or method to be utilized (e.g., global adjustment, tone curve adjustment, HSL adjustment, masking, texture, grain, cropping, dot) along with the specific value or range of values for the adjustment associated with that tool or method (e.g., "Saturation +15%", "High-pass Filter Radius: 3px"). All details regarding the tools/methods and their corresponding tuning values must be derived exclusively from the configuration file.
2. Step-by-Step Explanation: Describe adjustments in a logical sequence, prioritizing critical modifications first. Use layman-friendly terms but include professional jargon where necessary (e.g., 'recovered highlights via luminance mask').
3. Rationale: Explain how each change aligns with the user's intent (e.g., 'cooling tone applied to match requested 'cinematic mood').
Output Format: Freeform paragraphs with bullet points or numbered steps—no markdown. Prioritize clarity and technical precision.

Special Notes:
1. Respond EXCLUSIVELY in English for all outputs. Never include non-English characters or translations in other languages.
2. When mentioning coordinates or bounding boxes, you must enclose them within <box></box> tags. For example: 'I might consider adding a subtle glow or halo around the person (<box>0.1242, 0.699, 0.7453, 1.0000</box>)'
3. <box></box> tags must be in the form of <box>x1 y1 x2 y2</box> to represent bounding boxes. It is important to note that the form of <box>x1 y1</box> is incorrect.

configuration file: <corresponding_lua_file>
user requirements: <user intent>
Relative coordinates of the region of interest: <box>coordinates</box>
"""

# Prompt for Generating Initial Long Chain-of-Thought without Coordinate Information
"""Please analyze the provided before/after images, user requirements, Relative coordinates of the region of interest(optional), and configuration file to generate a detailed adjustment workflow. Although the configuration file is recognized, the response should avoid explicitly stating that the adjusted parameters were derived from it. The tone should convey a sense of expert judgment and reasoned analysis.
1. Technical Breakdown: Specify the tool or method to be utilized (e.g., global adjustment, tone curve adjustment, HSL adjustment, masking, texture, grain, cropping, dot) along with the specific value or range of values for the adjustment associated with that tool or method (e.g., "Saturation +15%", "High-pass Filter Radius: 3px"). All details regarding the tools/methods and their corresponding tuning values must be derived exclusively from the configuration file.
2. Step-by-Step Explanation: Describe adjustments in a logical sequence, prioritizing critical modifications first. Use layman-friendly terms but include professional jargon where necessary (e.g., 'recovered highlights via luminance mask').
3. Rationale: Explain how each change aligns with the user's intent (e.g., 'cooling tone applied to match requested 'cinematic mood').
Output Format: Freeform paragraphs with bullet points or numbered steps—no markdown. Prioritize clarity and technical precision.

Special Notes:
- Respond EXCLUSIVELY in English for all outputs. Never include non-English characters or translations in other languages.

configuration file: <corresponding_lua_file>
user requirements: <user intent>
"""

Figure 24: Prompt for generating the initial Chain-of-Thought (COT) annotations.

**Prompt for Producing Refined CoT Annotations**

# Prompt for Generating Initial Long Chain-of-Thought with Coordinate Information
"""
Please revise the provided Chain of Thought(CoT) to follow these guidelines:

1. Thinking basis: Use user goals and original images instead of config files to generate all reasoning steps. For example: Avoid using words that are obviously related to configuration, such as "config files".;
2. Expressive style: Imitates the human-like language patterns used to express thought processes during interpersonal communication. For example, it needs to start with "Alright", and the sentences need to be connected as naturally as human speech.;
3. Enforce length constraints: Compress CoT narratives to under 160 words through strategic distillation of key reasoning components;
4. User intent analysis: The main content of the user intent analysis in the original CoT needs to be retained;
5. Original Image Analysis: The primary elements of the original image analysis from the initial thought process should be preserved, encompassing both content analysis and aesthetic evaluation.

Apply these rules rigorously to ensure that the final CoT accurately reflects the professional thought process of image processing experts when they only have the original image materials and user needs.

Special Notes:
1. Output the modified CoT directly without the introduction of words such as "Alright, here's a refined Chain of Thought (CoT) that strictly adheres to the guidelines"
2. All outputs must be strictly in English, prohibiting the use of any other languages.
3. All image processing operations involved should retain the specific adjustment values mentioned in the original chain of thought (CoT).
4. When mentioning coordinates or bounding boxes, you must enclose them within <box></box> tags. For example: 'I might consider adding a subtle glow or halo around the person <box>0.1242, 0.699, 0.7453, 1.0000</box>'
5. <box></box> tags must be in the form of <box>x1 y1 x2 y2</box> to represent bounding boxes. It is important to note that the form of <box>x1 y1</box> is incorrect.
6. Please strictly enforce the word limit.
CoT:<original CoT>
Relative coordinates of the region of interest: <box>coordinates</box>
"""

# Prompt for Generating Initial Long Chain-of-Thought without Coordinate Information
"""
Please revise the provided Chain of Thought(CoT) to follow these guidelines:
1. Thinking basis: Use user goals and original images instead of config files to generate all reasoning steps. For example: Avoid using words that are obviously related to configuration, such as "config files".;
2. Expressive style: Imitates the human-like language patterns used to express thought processes during interpersonal communication. For example, it needs to start with "Alright", and the sentences need to be connected as naturally as human speech.;
3. Enforce length constraints: Compress CoT narratives to under 160 words through strategic distillation of key reasoning components;
4. User intent analysis: The main content of the user intent analysis in the original CoT needs to be retained;
5. Original Image Analysis: The primary elements of the original image analysis from the initial thought process should be preserved, encompassing both content analysis and aesthetic evaluation.

Apply these rules rigorously to ensure that the final CoT accurately reflects the professional thought process of image processing experts when they only have the original image materials and user needs.

Special Notes:
1. Output the modified CoT directly without the introduction of words such as "Alright, here's a refined Chain of Thought (CoT) that strictly adheres to the guidelines"
2. All outputs must be strictly in English, prohibiting the use of any other languages.
3. All image processing operations involved should retain the specific adjustment values mentioned in the original chain of thought (CoT).
4. Please strictly enforce the word limit.
CoT:<original CoT>
"""

Figure 25: Prompt for generating the refined Chain-of-Thought (COT) annotations.

# E  Details of Retouching Tools in Lightroom

We provide an overview of key Lightroom tools generated by JarvisArt, focusing on the functionality of retouching tools and their associated parameters:

Table 7: Lightroom Tools with Functional Description and Parameter Type.

| Tool Name | Functional Description | Type |
|---|---|---|
| **Basic Adjustments** | | |
| WhiteBalance | Overall color temperature (As Shot, Auto, Custom) | Str. |
| Temperature | Blue-yellow balance (2000-10000 Kelvin) | Num. |
| Tint | Green-magenta balance (-150 to +150) | Num. |
| Exposure2012 | Overall brightness (-5.0 to +5.0 stops) | Num. |
| Contrast2012 | Difference between light/dark areas (-100 to +100) | Num. |
| Highlights2012 | Adjusts bright areas (-100 to +100) | Num. |
| Shadows2012 | Adjusts dark areas (-100 to +100) | Num. |
| Whites2012 | Fine-tunes brightest parts (-100 to +100) | Num. |
| Blacks2012 | Fine-tunes darkest parts (-100 to +100) | Num. |
| Texture | Enhances/smooths medium textures (-100 to +100) | Num. |
| Clarity2012 | Enhances/reduces local mid-tone contrast (-100 to +100) | Num. |
| Dehaze | Reduces/adds atmospheric haze (-100 to +100) | Num. |
| Vibrance | Saturation of less-saturated colors (-100 to +100) | Num. |
| Saturation | Overall color intensity (-100 to +100) | Num. |
| IncrementalTemperature | Relative temperature adjustment (-100 to +100) | Num. |
| IncrementalTint | Relative tint adjustment (-100 to +100) | Num. |
| **Tone Curve** | | |
| ToneCurveName2012 | Predefined curve shape (Linear, Custom) | Str. |
| ToneCurvePV2012 | Custom RGB tone curve points (x,y: 0-255) | Dict. |
| ToneCurvePV2012Red | Custom Red channel tone curve points | Dict. |
| ToneCurvePV2012Green | Custom Green channel tone curve points | Dict. |
| ToneCurvePV2012Blue | Custom Blue channel tone curve points | Dict. |
| ParametricShadows | Adjusts shadow tonal regions (-100 to +100) | Num. |
| ParametricDarks | Adjusts dark tonal regions (-100 to +100) | Num. |
| ParametricLights | Adjusts light tonal regions (-100 to +100) | Num. |
| ParametricHighlights | Adjusts highlight tonal regions (-100 to +100) | Num. |
| ParametricShadowSplit | Boundary: shadows/darks (10-50) | Num. |
| ParametricMidtoneSplit | Boundary: darks/lights (25-75) | Num. |
| ParametricHighlightSplit | Boundary: lights/highlights (50-90) | Num. |
| **Detail** | | |
| Sharpness | Enhances edge definition (0-150) | Num. |
| SharpenRadius | Width of sharpening effect (0.5-3.0) | Num. |
| SharpenDetail | Amount of sharpening for details (0-100) | Num. |
| SharpenEdgeMasking | Masks sharpening to edges (0-100) | Num. |
| LuminanceSmoothing | Reduces luminance noise (0-100) | Num. |
| ColorNoiseReduction | Reduces color noise (0-100) | Num. |
| ColorNoiseReductionDetail | Fine-tunes color noise reduction (0-100) | Num. |
| ColorNoiseReductionSmoothness | Smoothness of color noise reduction (0-100) | Num. |
| **HSL/Color (per color: Red, Orange, Yellow, Green, Aqua, Blue, Purple, Magenta)** | | |
| HueAdjustment<Color> | Shifts hue of specific color (-100 to +100) | Num. |
| SaturationAdjustment<Color> | Adjusts saturation of specific color (-100 to +100) | Num. |
| LuminanceAdjustment<Color> | Adjusts brightness of specific color (-100 to +100) | Num. |
| **Color Grading** | | |
| SplitToningShadowHue | Hue for shadows in split toning (0-359) | Num. |
| SplitToningHighlightHue | Hue for highlights in split toning (0-359) | Num. |
| SplitToningShadowSaturation | Saturation for shadows (0-100) | Num. |
| SplitToningHighlightSaturation | Saturation for highlights (0-100) | Num. |
| SplitToningBalance | Balance between shadow/highlight toning (-100 to +100) | Num. |
| ColorGradeMidtoneHue | Midtone hue for color grading (0-359) | Num. |
| ColorGradeMidtoneSat | Midtone saturation for color grading (0-100) | Num. |
| ColorGradeMidtoneLum | Midtone luminance for color grading (0-100) | Num. |

Table 7: Lightroom tools with functional description and parameter type. (Continued)

| Tool Name | Functional Description | Type |
|---|---|---|
| ColorGradeShadowLum | Luminance for shadows (0-100) | Num. |
| ColorGradeHighlightLum | Luminance for highlights (0-100) | Num. |
| ColorGradeBlending | Blending of color grading effect (0-100) | Num. |
| ColorGradeGlobalHue | Global hue adjustment (0-359) | Num. |
| ColorGradeGlobalSat | Global saturation adjustment (0-100) | Num. |
| ColorGradeGlobalLum | Global luminance adjustment (0-100) | Num. |
| **Effects** | | |
| PostCropVignetteAmount | Darkens/lightens image corners (-100 to +100) | Num. |
| GrainAmount | Adds film grain effect (0-100) | Num. |
| ShadowTint | Adjusts color tint in shadows (-100 to +100) | Num. |
| **Camera Calibration (for Red, Green, Blue primary channels)** | | |
| \<PrimaryColor>Hue | Shifts primary color's hue (-100 to +100) | Num. |
| \<PrimaryColor>Saturation | Adjusts primary color's saturation (-100 to +100) | Num. |
| **Lens Blur (Overall: Dict.)** | | |
| LensBlur.Active | Enables/disables lens blur effect | Bool. |
| LensBlur.BlurAmount | Strength of blur effect (0-100) | Num. |
| LensBlur.FocalRange | Defines focal plane ("x1 y1 x2 y2") | Str. |
| LensBlur.BokehShape | Bokeh shape identifier (default 0) | Num. |
| LensBlur.BokehShapeDetail | Definition of bokeh shape edges (0-100) | Num. |
| LensBlur.HighlightsThreshold | Brightness threshold for bokeh (0-100) | Num. |
| LensBlur.HighlightsBoost | Enhances out-of-focus highlights (0-100) | Num. |
| LensBlur.CatEyeAmount | Simulates cat's eye bokeh effect (0-100) | Num. |
| LensBlur.CatEyeScale | Size of cat's eye effect (0-100) | Num. |
| **Advanced Color Grading (PointColors - each point is a Dict.)** | | |
| SrcHue | Source hue for adjustment (0-6.28 rad) | Num. |
| SrcSat | Source saturation for adjustment (0-1.0) | Num. |
| SrcLum | Source luminance for adjustment (0-1.0) | Num. |
| HueShift | Hue shift amount (-1 to +1) | Num. |
| SatScale | Saturation scale (-1 to +1) | Num. |
| LumScale | Luminance scale (-1 to +1) | Num. |
| RangeAmount | Effect application amount (0-1.0) | Num. |
| HueRange | Falloff for hue adjustment (LowerNone, LowerFull, UpperFull, UpperNone: 0-1.0) | Dict. |
| SatRange | Falloff for saturation adjustment (sub-props same as HueRange) | Dict. |
| LumRange | Falloff for luminance adjustment (sub-props same as HueRange) | Dict. |
| **Look (Overall: Dict.)** | | |
| Look.Name | Name of the look preset | Str. |
| Look.Amount | Intensity of the look effect (0.0-1.0) | Num. |
| Look.Parameters *(e.g., ProcessVersion, ToneCurvePV2012, Parametric adjustments, SplitToning, Color-Grade, ConvertToGrayscale, LookTable, RGBTable, RG-BTableAmount)* | Dictionary of specific adjustments applied by the look | Dict. |
| **Localized Mask Adjustments (MaskGroupBasedCorrections - Array of Dicts.)** | | |
| *Per Correction Group:* | | |
| CorrectionAmount | Amount for the correction group (0-1, default 1) | Num. |
| CorrectionActive | Activates the correction group | Bool. |
| CorrectionName | Name for the correction group | Str. |
| LocalExposure2012 | Local exposure adjustment (-1 to +1) | Num. |
| LocalContrast2012 | Local contrast adjustment (-1 to +1) | Num. |
| LocalHighlights2012 | Local highlights adjustment (-1 to +1) | Num. |
| LocalShadows2012 | Local shadows adjustment (-1 to +1) | Num. |
| LocalWhites2012 | Local whites adjustment (-1 to +1) | Num. |

Table 7: Lightroom tools with functional description and parameter type. (Continued)

| Tool Name | Functional Description | Type |
|---|---|---|
| LocalBlacks2012 | Local blacks adjustment (-1 to +1) | Num. |
| LocalClarity / LocalClarity2012 | Local clarity adjustment (-1 to +1) | Num. |
| LocalDehaze | Local dehaze adjustment (-1 to +1) | Num. |
| LocalTexture | Local texture adjustment (-1 to +1) | Num. |
| LocalHue | Local hue adjustment (-1 to +1) | Num. |
| LocalSaturation | Local saturation adjustment (-1 to +1) | Num. |
| LocalCurveRefineSaturation | Local saturation curve refinement (0-100) | Num. |
| LocalToningHue | Local toning hue (0-359) | Num. |
| LocalToningSaturation | Local toning saturation (-1 to +1) | Num. |
| LocalTemperature | Local temperature adjustment (-1 to +1) | Num. |
| LocalTint | Local tint adjustment (-1 to +1) | Num. |
| LocalLuminanceNoise | Local luminance noise reduction (-1 to +1) | Num. |
| LocalMoire | Local moire reduction (-1 to +1) | Num. |
| LocalDefringe | Local defringe adjustment (-1 to +1) | Num. |
| LocalGrain | Local grain adjustment (-1 to +1) | Num. |
| LocalSharpness | Local sharpness adjustment (-1 to +1) | Num. |
| <Channel>Curve | Local tone curve for Red, Green, Blue, or Main channels (points "x,y") | Dict. |
| LocalPointColors | Local specific color adjustments (dictionary of string-encoded points) | Dict. |
| CorrectionMasks | Array of mask definitions for the group | Array |
| *Per Mask in CorrectionMasks:* | | |
| What | Mask type (e.g., "Mask/Image", "Mask/CircularGradient") | Str. |
| MaskActive | Activates this specific mask | Bool. |
| MaskName | Name of the mask (e.g., "Subject", "Sky") | Str. |
| MaskBlendMode | Mask blending (0=Add, 1=Intersect) | Num. |
| MaskInverted | Inverts the mask area | Bool. |
| MaskValue | Mask opacity (0.0-1.0) | Num. |
| MaskSubType | AI Mask subtype (Subject, Sky, Person etc.) / Object type | Num. |
| ReferencePoint | Center point for AI masks ("x y") | Str. |
| Gesture | Polygon points for object/region mask | Array |
| Top/Left/Bottom/Right | Coordinates for radial gradient (0-1) | Num. |
| Angle | Rotation angle for radial gradient (0-360) | Num. |
| Midpoint | Center point of radial gradient (0-100) | Num. |
| Feather | Edge feathering for radial gradient (0-100) | Num. |
| Flipped | Flips radial gradient direction | Bool. |
| MaskSubCategoryID | Category ID for person parts mask (Face, Eyes, etc.) | Num. |

