# OpenReview forum: "JarvisArt: Liberating Human Artistic Creativity via an Intelligent Photo Retouching Agent"
_NeurIPS.cc/2025/Conference — NeurIPS 2025 poster_

### Official Review · Reviewer_xmNH · 2025-06-28

**Clarity:** 3
**Significance:** 3
**Originality:** 3
**Rating:** 5
**Confidence:** 5

**Summary:**

The paper presents JarvisArt, an MLLM-driven photo retouching agent that integrates with Lightroom to automate professional-level edits.

**Questions:**

I don't have more questions

**Ethical Concerns:**

["NO or VERY MINOR ethics concerns only"]

**Final Justification:**

My concern has been addressed.

**Limitations:**

Yes

**Quality:**

3

**Strengths And Weaknesses:**

Strengths
1. JarvisArt bridges the gap between natural language understanding and professional retouching workflows by orchestrating over 200 Lightroom tools. The two-stage training (CoT SFT + GRPO-R) and Agent-to-Lightroom Protocol show thoughtful design for real-world applicability.
2. The MMArt dataset and MMArt-Bench benchmark provide a robust foundation for training and testing. Quantitative results show JarvisArt outperforms GPT-4o in content fidelity by 60% while maintaining instruction-following capabilities.
3. The system addresses a critical pain point for non-experts by automating complex edits without sacrificing control. Visual comparisons highlight its superiority in preserving details and avoiding artifacts compared to baselines like InstructPix2Pix and MagicBrush.

Weaknesses
1. The user preference study recruits 80 participants but omits details on their expertise (e.g., amateur vs. professional) and the evaluation criteria. The 5-point ordinal scale may not capture nuanced differences in aesthetic judgment.

---

> ### Author Rebuttal · Authors · 2025-07-31
>
> We are grateful for your positive feedback and your recognition of our motivation, professional retouching results, and the contribution of the MMArt dataset. Please find our detailed responses to the review comments below.
>
> **Q1: Participant demographics in the user study.**
> > To ensure the comprehensiveness of our user study, we recruited 80 participants with diverse professional backgrounds, including art, computer science, photography, and design. The cohort comprises both male (50%) and female (50%) participants aged 18-35, including amateur/professional photographers (35%), digital artists (25%), and general users (40%). This multi-dimensional sampling strategy helps evaluate system performance across different user groups and application scenarios. This will be clarified in the manuscript.
>
> **Q2: Evaluation criteria in the user study.**
> > The evaluation criteria focus on two key dimensions: **(1)** image consistency, which measures how closely the content of the retouched image aligns with the source image; and **(2)** aesthetic quality, which evaluates the artistic merit of retouched images based on professional photography standards, including color harmony, composition balance, and visual impact. To maintain consistency with prior user studies [1][2], we adopt the five-level quality scale due to its simplicity, efficiency, and effectiveness in distinguishing significant differences in retouching quality. Notably, during the aesthetic assessment, participants are required to evaluate the overall aesthetic quality based on three key aspects—color harmony, lighting and shadow, and visual appeal—before scoring. This will be clarified in the manuscript.
>
> **Q3: Discussion of refined aesthetic evaluation.**
> > Pairwise comparison is a more refined evaluation method where participants are shown two images side-by-side and asked to choose their preference or identify the higher-quality one. This approach excels at capturing subtle differences in user preference and can mitigate biases like the anchoring effect, but its efficiency significantly decreases with a larger number of images due to the exponential increase in comparisons needed, potentially leading to participant fatigue and requiring more complex statistical analysis.
>
> ---
> References:
>
> [1] Step1x-edit: A practical framework for general image editing. Arxiv 2025, Github star 1.5k.
>
> [2] MagicQuill: An Intelligent Interactive Image Editing System. CVPR 2025.

---

> > ### Author Response · Authors · 2025-08-04
> >
> > Dear Reviewer xmNH,
> >
> > We sincerely appreciate your insights and welcome any feedback on our rebuttal.
> >
> > Given the computational demands of the agentic tool-integrated retouching task (e.g., MLLM processing and Lightroom invocation), we would be grateful if you could let us know whether our response has adequately addressed your concerns. Your feedback will help us ensure the timely delivery of results.
> >
> > Thank you very much for your time and thoughtful input.
> >
> > Best regards, The Authors

---

> > ### Comment · Reviewer_xmNH · 2025-08-04
> >
> > Thanks for your response! My concerns have been addressed. Good luck!

---

### Official Review · Reviewer_VQZs · 2025-07-01

**Clarity:** 4
**Significance:** 2
**Originality:** 3
**Rating:** 4
**Confidence:** 5

**Summary:**

This paper presents JarvisArt, an MLLM-based agent that interprets user instructions and derives Lightroom operations via chain-of-thought (CoT) reasoning. The authors curate a dataset and adopt Group Relative Policy Optimization (GRPO) to enhance the MLLM's decision-making. They also introduce a new evaluation set, MMArt-Bench, constructed from real-world edits.

**Questions:**

+ Although the performance is promising, the proposed pipeline—including SFT data curation and RL post-training—builds upon existing methods. The level of technical novelty may be a concern for this paper. I am not entirely sure whether it meets the NeurIPS acceptance bar.
+ Since the operations are derived via an MLLM, how accurate are the intermediate steps? Is there a risk of error accumulation that could degrade final performance? How can such issues be mitigated?
+ While Figure 7 shows an encouraging trend in RL training, there should be an analysis of the relationship between the reward score and actual performance. Does a higher reward score truly correspond to better outcomes?
+ In addition to successful examples, a discussion of failure cases is important—particularly to highlight areas for improvement and guide future research.

**Minor**
+ Missing reference: Guiding Instruction-based Image Editing via Multimodal Large Language Models, which also adopts MLLMs to guide image editing.

**Ethical Concerns:**

["NO or VERY MINOR ethics concerns only"]

**Limitations:**

yes

**Quality:**

3

**Strengths And Weaknesses:**

**Main Strengths**
- This paper is well-written and easy to follow.
- The authors target a real-world application with significant potential for extension to various artistic design tasks.
- They present numerous qualitative examples and detailed training curves, which are crucial for tracking and demonstrating the efficiency of GRPO.
- The curated dataset leads to substantial improvements on the proposed benchmark.

**Main Weakness**
- One of the most valuable contributions of this paper is the curated dataset and the proposed MMArt-Bench. However, I could not find a shareable link in the draft. If the authors do not plan to open-source these resources, it may limit the potential impact of the work.

---

> ### Author Rebuttal · Authors · 2025-07-30
>
> We are grateful for your positive feedback and your recognition of our writing quality, the promise of MLLM-driven artistic agents, and the value of the MMArt dataset. Please find our detailed responses to the review comments below.
>
> **Q1: Open-source plan for the MMArt dataset.**
> > The dataset will be released once the paper is accepted.
>
> **Q2: Technical novelty of this work.**
> > As acknowledged by Reviewer `NfhL`, one of our primary contributions is the development of a robust and scalable infrastructure, comprising a data-synthesis pipeline and a fine-tuning framework that effectively transform MLLMs into artistic agents. While our training framework relies on widely used techniques like SFT and RL, we propose GRPO-R, an RL method tailored for tool-integrated retouching tasks. In contrast to standard GRPO—which targets problems with single, verifiable answers—retouching tasks involve two key challenges: **(1)** predicting multiple tools and their parameters, and **(2)** managing the inherent ambiguity in parameter combinations that yield visually similar results. This complexity makes reward design particularly difficult. GRPO-R addresses this by incorporating two reward components: `retouching operation accuracy` for parameter prediction and `perceptual quality` for visual fidelity. This specific reward design enables scalable reinforcement learning, thereby enhancing the agent's reasoning ability, tool proficiency, and generalization.
>
> **Q3: Accuracy of the intermediate steps of MLLM.**
> > **During the data curation process**, to ensure the accuracy of intermediate reasoning steps generated by the MLLM, we adopt a two-stage verification approach: **(1)** the Qwen2.5-VL-72B model evaluates the alignment between instructions and reasoning via multimodal input prompts, and **(2)** manual sampling checks are conducted to further validate quality. This verification procedure will be clarified in the manuscript.
> >
> > **During JarvisArt's inference phase**, artistic reasoning may introduce error accumulation that degrades overall performance. For example, if the model predicts an incorrect adjustment such as `"exposure +50"`, it may produce suboptimal visual results. Since the model lacks immediate visual feedback during reasoning, such errors are not corrected and may compound in subsequent steps. Introducing stepwise visual rewards during the RL phase may allow the model to validate intermediate predicted parameters, thereby improving the reliability of the model during inference and reducing errors. Ongoing work aims to further optimize this implementation.
>
> **Q4: Impact of reward scores on performance outcomes.**
> > To analyze the correlation between reward scores and performance metrics, we conducted progressive assessments of checkpoints at four training milestones (**25%, 50%, 75%, and 100% completion**) during RL training. As shown in Table 1, results on MMArt-Bench demonstrate a statistically positive correlation between reward scores and performance metrics. This confirms that the GRPO-R can effectively enhance the model's reasoning, tool proficiency, and generalization during reward optimization.
>
> **Table 1: Analysis of correlation between rewards and performance**
> | **Training Phase** | **Reward Score ↑** | **L1×10² ↓** | **L2×10³ ↓** | **SC ↑** | **PQ ↑** | **O ↑** |
> |----------------|:--------------:|:--------:|:--------:|:----:|:----:|:----:|
> | 25% Completion | 2.071           | 14.23    | 43.75    | 7.41 | 8.73 | 8.04 |
> | 50% Completion | 2.094          | 14.14    | 40.92     | 7.44 | 8.81 | 8.09 |
> | 75% Completion | 2.313          | 13.46    | 35.15    | 7.49 | 9.37 | 8.37 |
> | 100% Completion| 2.416           | 12.44    | 30.56    | 7.53 | 9.82 | 8.52 |
>
> **Q5: Failure cases and future work.**
> > In this work, we focus on pushing the boundaries of photo retouching and offer new insights into how artificial intelligence can empower and elevate artistic creation. While promising, the current system has several limitations. First, it lacks multilingual capabilities, which we aim to address by adding linguistically diverse training data. Second, it struggles to capture fine-grained user intent, particularly in localized facial edits (e.g., eyes and lips). To improve performance, we propose two strategies: **(1)** augmenting the training dataset with examples reflecting subtle and specific user editing preferences, and **(2)** designing fine-grained reward mechanisms in the RL phase to help the model better capture nuanced user intent and invoke appropriate retouching operations.
>
>
> **Q6: Citations of related works.**
> > We will cite all listed works.

---

> > ### Author Response · Authors · 2025-08-04
> >
> > Dear Reviewer VQZs,
> >
> > We sincerely appreciate your insights and welcome any feedback on our rebuttal.
> >
> > Given the computational demands of the agentic tool-integrated retouching task (e.g., MLLM processing and Lightroom invocation), we would be grateful if you could let us know whether our response has adequately addressed your concerns. Your feedback will help us ensure the timely delivery of results.
> >
> > Thank you very much for your time and thoughtful input.
> >
> > Best regards, The Authors

---

> > ### Author Response · Authors · 2025-08-07
> >
> > Dear reviewer VQZs,
> >
> > We sincerely appreciate your insights and welcome any feedback on our response.
> >
> > Given that the rebuttal deadline is approaching, we would be grateful if you could let us know whether our response has adequately addressed your concerns. Your feedback will help us ensure the timely delivery of results.
> >
> > Thank you again for your time and thoughtful review.
> >
> > Sincerely, All authors

---

> > > ### Author Response · Authors · 2025-08-08
> > > **The deadline is less than 24 hours away, and final scoring has not yet been completed.**
> > >
> > > Dear reviewer VQZs,
> > >
> > > We sincerely appreciate your insights and welcome any feedback on our response.
> > >
> > > Given that the **rebuttal deadline is less than 24 hours**, we would be grateful if you could let us know whether our response has adequately addressed your concerns. Your feedback will help us ensure the timely delivery of results.
> > >
> > > Thank you again for your time and thoughtful review.
> > >
> > > Sincerely, All authors

---

> > > > ### Author Response · Authors · 2025-08-09
> > > > **The deadline is less than 3 hours away, and final scoring has not yet been completed.**
> > > >
> > > > Dear reviewer VQZs,
> > > >
> > > > We sincerely appreciate your insights and welcome any feedback on our response.
> > > >
> > > > Given that the rebuttal deadline is less than 3 hours, we would be grateful if you could let us know whether our response has adequately addressed your concerns. Your feedback will help us ensure the timely delivery of results.
> > > >
> > > > Thank you again for your time and thoughtful review.
> > > >
> > > > Sincerely, All authors

---

### Official Review · Reviewer_3pvW · 2025-07-01

**Clarity:** 3
**Significance:** 3
**Originality:** 3
**Rating:** 5
**Confidence:** 5

**Summary:**

The paper proposes JarvisArt, a Lightroom-based intelligent photo-retouching agent driven by a multimodal LLM. JarvisArt interprets both textual and visual user intent and can orchestrate more than 200 Lightroom tools for global and local, non-destructive edits. A two-stage training pipeline—Chain-of-Thought (CoT) supervised fine-tuning followed by Group Relative Policy Optimization for Retouching (GRPO-R)—plus an A2L protocol enables end-to-end automation. The authors also release a large CoT dataset (MMArt-55K) and an evaluation benchmark (MMArt-Bench). Experiments show that JarvisArt outperforms GPT-4o, Gemini-2-Flash and other state-of-the-art models in pixel-level metrics and instruction adherence, and wins a user-preference study with 80 participants.

**Questions:**

Questions and Actionable Requests for the Rebuttal
1. GRPO-R scale and reproducibility. Please supply the full hyper-parameter table (learning-rate schedule, reward weights, KL coefficients, batch sizes, wall-clock time, GPU type/count) and clarify why only 5 K samples were used.
2. Objective comparison with commercial auto-modes.  Add quantitative metrics (e.g., PSNR, LPIPS, SC, PQ) for Lightroom “Auto” and Google Photos “Enhance” on MMArt-Bench and briefly discuss any failure cases.
3. Training and inference efficiency.  Report GPU hours for SFT and GRPO-R, and give average inference latency and VRAM use for a 24 MP image. Indicate whether mixed-precision or model pruning was applied.

**Ethical Concerns:**

["NO or VERY MINOR ethics concerns only"]

**Final Justification:**

The authors have provided the full details, comparisons with commercial auto-retouch tools, efficiency metrics, and a clear privacy guarantee, directly addressing my key concerns about reproducibility, evaluation rigor, and practicality. Thanks!

**Limitations:**

1. Compute & efficiency – please quantify GPU hours for SFT/RL and give typical inference latency on a 24-MP image so readers can gauge practicality across hardware tiers.
2. Privacy considerations – clarify how user-supplied photos are stored/processed and whether the A2L pipeline leaves any personal data on external servers.

**Quality:**

3

**Strengths And Weaknesses:**

Major Strengths
1. Clear Task Focus – Targets high-resolution, non-destructive photo retouching, a realistic need distinct from purely generative models.
2. Complete Workflow – Covers data generation, model training and tight Lightroom integration, forming a deployable pipeline.
3. Community Assets – MMArt-55K supplies CoT, local coordinates and ROC files—valuable resources for future work.

Main Weaknesses
1. GRPO-R uses only 5 K samples; more training details could appear in the supplement.
2. Comparisons with commercial auto-retouch modes (Lightroom Auto, Google Photos) are visual only—no numerical metrics.
3. Perceptual-quality reward requires rendering images; training cost and throughput are not quantified.
4. The paper is long; some appendix material duplicates the main text and could be trimmed.

Suggestions for Improvement
1. Release additional GRPO-R hyper-parameters and training time to aid reproducibility.
2. Add objective metrics (e.g., PSNR, LPIPS) versus Lightroom Auto in the appendix.
3. Discuss inference cost on mobile or low-compute devices.
4. Explore support for Lightroom’s newer AI features such as background removal.

Recommended to add citations to the following work.
1. WordArt designer: User-driven artistic typography synthesis using large language models. (2023). In Proceedings of the 2023 Conference on Empirical Methods in Natural Language Processing.
2. Music2P: A multi-modal AI-driven tool for simplifying album cover design. (2024). In Proceedings of the 33rd ACM International Conference on Information and Knowledge Management (CIKM 2024).
3. MetaDesigner: Advancing artistic typography through AI-driven, user-centric, and multilingual WordArt synthesis. (2025). In Proceedings of the Thirteenth International Conference on Learning Representations (ICLR 2025).

---

> ### Author Rebuttal · Authors · 2025-07-30
>
> We appreciate your positive feedback and recognition of our clearly articulated motivation, comprehensive data generation and training workflow, and the contribution of the MMArt dataset. Please find our detailed responses to the review comments below.
>
> **Q1: Full hyper-parameter setting for SFT and GRPO-R.**
> > To ensure reproducibility, we provide the complete hyperparameter settings for both the SFT and GRPO-R phases in Table 1. These details are included in the revised manuscript.
>
> **Table 1: Hyper-parameter settings for SFT and GRPO-R**
> | Hyper-parameter | SFT | GRPO-R |
> |:-----|:-----|:-----|
> | Batch size | 2 | 2 |
> | Learning rate | 1e-5 | 1e-6 |
> | Weight decay | 0| 0 |
> | Optimizer | AdamW | AdamW |
> | Warmup ratio | 0.1 | 0.1 |
> | LR scheduler | cosine | cosine |
> | Training samples | 50K | 5K |
> | Training epochs | 2 | 2 |
> | Precision | bfloat16 | bfloat16 |
> | KL coefficient | - | 0.04 |
> | Reward coefficients | - | $R_f$:1, $R_{roa}$:1, $R_{pq}$:1 ($\gamma$=0.4) |
> | Number of generations | - | 4 |
> | GPU resources | 8×A100 (～384 GPU hours) | 16×A100 (～2076 GPU hours) |
>
> **Q2: Rationale for 5K samples in GRPO-R training.**
> > We use 5K training samples for the GRPO-R phase for two main considerations.
> > - **First**, as shown in Table 2 of the manuscript, even this modest data yields substantial improvements over the SFT baseline: L1 and L2 errors drop notably (**14.42 → 12.44, 44.38 → 30.56**), while SC, PQ, and O all increase (**7.32 → 7.53, 8.67 → 9.82, 7.94 → 8.52**).
> > - **Second**, scaling beyond 5K samples significantly increases computational costs, especially due to the overhead of parallel Lightroom invocations. Given our current resource constraints, large-scale RL training is not yet feasible, though we anticipate further gains with increased data in future work.
>
> **Q3: Quantitative comparison with commercial photo retouching methods.**
> > We provide both scene- and region-level quantitative comparison, including commercial retouching tools and instruction-driven generative editing models. As shown in Table 2, JarvisArt combines superior pixel-level content preservation—comparable to commercial tools like Google Photos and Lightroom Auto—with advanced instruction-following capabilities on par with GPT-4o and Gemini-2-Flash. This integration positions JarvisArt as a versatile system that unifies the strengths of both paradigms. Notably, evaluating instruction-driven retouching tasks is inherently subjective: reference images represent only one of many acceptable outputs, and traditional image metrics often fail to reflect aesthetic quality or user preferences. As discussed in Section 4.2.2 of the manuscript, user studies may provide a more reliable and human-centric evaluation.
>
> **Table 2: Quantitative comparison on both scene-level and region-level**
> | **Method**            | **Instruction** | **Scene-level**  |  |  |  |  | **Region-level** |  |  |  |  |
> |-------------------|:--------:|:--------:|:--------:|:--------:|:--------:|:--------:|:--------:|:--------:|:--------:|:--------:|:--------:|
> |                   |          | $\text{PSNR}$ ↑   | $\text{LPIPS}$ ↓  | $\text{SC}$ ↑     | $\text{PQ}$  ↑     | $\text{O}$ ↑      | $\text{PSNR}^{RC}$ ↑   | $\text{LPIPS}^{RC}$ ↓  | $\text{SC}^{RC}$ ↑     | $\text{PQ}^{RC}$ ↑     | $\text{O}^{RC}$ ↑      |
> | Lightroom Auto    | ❌       | 21.19    | 0.198    | -        | -        | -        | 20.11    | 0.166    | -        | -        | -        |
> | Google Photos      | ❌       | 22.04    | 0.180    | -        | -        | -        | 23.30    | 0.134    | -        | -        | -        |
> | RSFNet            | ❌       | 21.68    | 0.194    | -        | -        | -        | 23.07    | 0.155    | -        | -        | -        |
> | 3DLUT             | ❌       | 21.77    | 0.193    | -        | -        | -        | 23.86    | 0.149    | -        | -        | -        |
> | InstructPix2Pix   | ✅       | 15.64    | 0.429    | 6.54     | 7.79     | 7.10     | 15.17    | 0.479    | 4.70     | 5.36     | 4.91     |
> | MagicBrush        | ✅       | 13.55    | 0.598    | 3.93     | 4.09     | 3.85     | 15.68    | 0.543    | 3.04     | 3.41     | 3.13     |
> | OmniGen           | ✅       | 8.68     | 0.821    | 4.25     | 4.42     | 4.13     | 8.61     | 0.784    | 6.17     | 7.56     | 6.72     |
> | VARGPT-v1.1       | ✅       | 9.20     | 0.743    | 1.83     | 1.38     | 1.48     | 9.17     | 0.747    | 1.38     | 1.15     | 1.08     |
> | Step1X-Edit       | ✅       | 11.20    | 0.600    | 7.52     | 8.67     | 8.01     | 14.80    | 0.358    | 8.32     | 9.04     | 8.66     |
> | Gemini-2-Flash    | ✅       | 11.38    | 0.565    | 7.62     | 8.78     | 8.08     | 13.87    | 0.448    | 8.04     | 9.25     | 8.61     |
> | GPT-4o            | ✅       | 11.54    | 0.629    | 8.73     | 9.66     | 9.18     | 13.91    | 0.585    | 8.59     | 9.48     | 9.03     |
> | **JarvisArt**     | ✅       | **21.54**| **0.263**| **7.53** | **9.82** | **8.52** | **22.78**| **0.142**| **8.08** | **9.39** | **8.69** |
>
> **Q4: Inference efficiency.**
> > Table 3 summarizes the average wall-clock time for processing 200 images at 24MP resolution across various deployment frameworks and precision settings. The inference process comprises two main components: **(1)** The VLM generates Lightroom retouching parameters; **(2)** these parameters are applied locally within the user's Lightroom environment. Notably, we set VLM to support input images up to 1,048,576 pixels for efficiency. For larger images, automatic resizing with aspect ratio preservation ensures no distortion. Since the VLM is solely responsible for parameter generation, the actual retouching is performed by Lightroom, which supports arbitrary resolutions and image formats. Results show that vLLM with bfloat16 precision yields the optimal wall-clock performance (18.21s total, with 12.51s for VLM and 5.69s for Lightroom). Future work will explore further acceleration strategies, such as model quantization and speculative sampling. As a pioneering effort, this study introduces the concept of an artist agent and establishes an effective fine-tuning framework. We will continue to enhance this paradigm.
>
> **Table 3: Comparison of inference time across frameworks and precision settings**
> | **Framework**| **Precision** | **Wall-clock Time (s)** | **VLM Processing (s)** | **Lightroom Processing (s)** | **Time to First Token (s)** | **Throughput (tokens/s)**  | **Avg Generated Tokens** | **VRAM Usage (GB)** |
> |:--------------------------|:----------|:------------------|:------------|:------------|:------------|:-------------------|:-----------|:------------
> | HuggingFace Transformers | float32      | 58.60        | 52.91 | 5.69 | 6.09   | 26.64 | 1241.6      | ~ 76.29     |
> | HuggingFace Transformers | bfloat16      | 30.14        | 24.45 | 5.69 | 4.45   | 53.54  | 1096.1     |  ~ 58.98       |
> | vLLM                       | float32      | 20.46        | 14.77 | 5.69 | 4.44   | 124.32 | 1241.6      |  ~ 44     |
> | **vLLM**                      | **bfloat16**      | **18.21**        | **12.51** | **5.69** | **3.61**   | **137.94** | **1227.8**      | **~ 29.6**      |
> | vLLm* | bfloat16      |   28.82   | 23.13 | 5.69 | 1.164 | 55.31 | 1214.3 | ~ 21.38 |
>
> System configurations:
> - VLM inference time is measured on an NVIDIA A100 GPU (80 GB).
> - \* denotes VLM inference time measured on an NVIDIA RTX 4090 GPU (24 GB).
> - Lightroom processing is conducted on a Windows 11 system equipped with an AMD Ryzen 9 5900X CPU and an NVIDIA RTX 3090 GPU.
>
> **Q5: Privacy considerations.**
> > To ensure user privacy, the Agent-to-Lightroom (A2L) protocol adopts a strictly sandboxed execution architecture and a fully localized processing workflow. As detailed in Supplementary Section A, the protocol runs entirely within the user's local Lightroom environment: **(1)** File verification is performed via OS-level checks without any external transmission; **(2)** all operations are confined to Lightroom's Lua environment through official APIs; and **(3)** retouching tasks are processed asynchronously in the local background. This design guarantees that user photos never leave the local Lightroom instance—no data is transmitted or stored outside the user's device.
>
> **Q6: Inference cost on mobile or low-compute devices.**
> > In this work, we focus on pushing the boundaries of photo retouching and offer new insights into how artificial intelligence can empower and elevate artistic creation. Deploying models to mobile devices or edge devices is beyond the scope of this paper. For such scenarios, we believe that model compression techniques—such as quantization, pruning, and knowledge distillation—can effectively reduce model size and inference costs, thus supporting such applications.
>
> **Q7: Failure cases and future work.**
> > While JarvisArt is promising, the current system has several limitations. First, it lacks multilingual capabilities, which we aim to address by adding linguistically diverse training data. Second, it struggles to capture fine-grained user intent, particularly in localized facial edits (e.g., eyes and lips). To improve performance, we propose two strategies: **(1)** augmenting the training dataset with examples reflecting subtle and specific user editing preferences, and **(2)** designing fine-grained reward mechanisms in the RL phase to help the model better capture nuanced user intent and invoke appropriate retouching operations.
>
> **Q8: Support for Lightroom’s AI features.**
> > To facilitate the model's adaptation to AI functionalities in Lightroom—such as background removal—we will employ the proposed data pipeline to generate paired artist-edited datasets aligned with specific AI operations. Fine-tuning on this targeted data is expected to enhance the model’s proficiency with these tools.
>
> **Q9: Paper presentation and citations of related works.**
> > We have made the necessary revisions and will cite all listed works.

---

> > ### Author Response · Authors · 2025-08-04
> >
> > Dear Reviewer 3pvW,
> >
> > We sincerely appreciate your insights and welcome any feedback on our rebuttal.
> >
> > Given the computational demands of the agentic tool-integrated retouching task (e.g., MLLM processing and Lightroom invocation), we would be grateful if you could let us know whether our response has adequately addressed your concerns. Your feedback will help us ensure the timely delivery of results.
> >
> > Thank you very much for your time and thoughtful input.
> >
> > Best regards, The Authors

---

> > ### Comment · Reviewer_3pvW · 2025-08-06
> >
> > The authors have provided the full details, comparisons with commercial auto-retouch tools, efficiency metrics, and a clear privacy guarantee, directly addressing my key concerns about reproducibility, evaluation rigor, and practicality. Thanks!

---

> > > ### Author Response · Authors · 2025-08-07
> > >
> > > Dear reviewer 3pvW,
> > >
> > > Thank you for your thoughtful participation and contributions to the discussion. We're pleased that our responses have addressed your concerns about reproducibility, evaluation rigor, and practicality. We’ll be sure to reflect your suggestions in the final version.
> > >
> > > If you feel that there are no further concerns, we would like to kindly remind you to submit your final review and ratings. We would be sincerely grateful for your support!
> > >
> > > Of course, if any new questions arise, please feel free to reach out—we’d be happy to clarify. If you have already submitted your final review, please kindly disregard this message.
> > >
> > > Sincerely, All authors

---

### Official Review · Reviewer_NfhL · 2025-07-02

**Clarity:** 2
**Significance:** 2
**Originality:** 2
**Rating:** 4
**Confidence:** 3

**Summary:**

The paper introduces JarvisArt, an agent for photo retouching based on a multi-modal large language model (MLLM) framework. The system allows lay users to apply high-quality Adobe Lightroom edits via natural language prompts. To support this, the authors construct a new dataset (MMArt) containing Chain-of-Thought (CoT) annotated samples that simulate artistic reasoning and tool-use skills. The dataset is generated using Qwen2.5-VL-72B and QVQ-max for image editing and reasoning, followed by human curation to select the most pleasing results.

The JarvisArt agent undergoes a two-stage post-training process: (1) CoT-supervised fine-tuning to build basic reasoning skills, and (2) Group Relative Policy Optimization for Retouching (GRPO-R), which refines the results through rewards targeting (i) output format, (ii) tool-use correctness, and (iii) perceptual quality.

The authors conduct quantitative comparisons against state-of-the-art models such as InstructPix2Pix, MagicBrush, OmniGen, VARGPT-v1.1, Step1X-Edit, Gemini-2-Flash, and GPT-4o. They report improvements in pixel-level metrics and user preference studies, particularly for content fidelity and subjective quality. An ablation study explores the contributions of the training stages and reward design.

**Questions:**

1. Could you clarify in detail the process from user prompt to ROC file generation? Specifically, how does CoT reasoning integrate with this?

2. Given that competing models perform pixel-level edits while JarvisArt uses Lightroom, how do you ensure a fair comparison in your evaluation? Metrics like L1 and L2 naturally favor content-preserving approaches like yours, potentially overstating advantages.

3. What are the actual contributions of this work, both general and specific? The paper mentions a practical system, dataset, and training strategy, but the distinction between engineering integration and conceptual novelty remains unclear. Could you clearly articulate what you see as the general research contributions to the field, beyond the applied success within Lightroom?

4. Is GRPO-R conceptually novel, or is it a straightforward application of existing GRPO techniques? If novel, what are the technical distinctions compared to standard GRPO?

5. The MMArt dataset would constitute a clear contribution if its construction were more robust. Did you consider incorporating multiple diverse VLMs for dataset generation and reasoning to reduce model-specific biases? How do you justify MMArt as a general research resource rather than a product of your specific model stack?

**Ethical Concerns:**

["NO or VERY MINOR ethics concerns only"]

**Final Justification:**

This paper presents strong experimental results and reflects a commendable engineering effort. The evaluation is thorough and indicates that the application is practically useful. While I remain uncertain about the depth of the research contribution, the practical strengths could justify acceptance.

**Limitations:**

Yes

**Quality:**

4

**Strengths And Weaknesses:**

**Strengths**

   - The paper presents very strong qualitative results, with realistic, professional-looking edits that preserve content fidelity better than competing methods.

   - Introduction of the MMArt dataset combining image pairs, user instructions, Lightroom configurations, and CoT annotations is valuable for the community.

   - Comprehensive and well-executed evaluation pipeline, including both automated metrics and a user preference study.

   - The system shows a promising application of MLLMs to concrete artistic tasks, demonstrating that models can be trained to emulate expert-level tool use within professional software.

**Weaknesses**

   - The paper's methodological novelty is modest. The contribution lies in system integration and dataset curation rather than introducing fundamentally new algorithms. The GRPO-R extension appears to be an application of existing GRPO techniques, adapted for retouching with custom rewards, but not conceptually novel.

   - Despite claims of generalizable artistic reasoning, the approach is narrowly scoped to Adobe Lightroom and to specific types of edits (primarily color, tone, and sharpness). More creative, structural, or semantic edits (e.g., object addition/removal) are out of scope.

   - Competing models often perform generative edits directly on image pixels, whereas JarvisArt operates within Lightroom, inherently preserving more content. This fundamental difference makes direct pixel-level comparisons (L1, L2) potentially misleading. The paper acknowledges content preservation advantages but does not control for this disparity.

   - The dataset generation process lacks sufficient robustness. Only Qwen2.5-VL-72B and QVQ-max VLMs are used, without clear consistency checks. This limits confidence in the dataset. Aggregating outputs from multiple VLMs and performing consensus filtering would significantly strengthen the dataset's credibility.

   - The system pipeline is insufficiently explained in key areas. It is unclear how the user prompt is transformed through the CoT reasoning to generate the final ROC file with Lightroom instructions.

   - Figures, particularly Figure 2 and Figure 3, are confusing and cluttered, making it difficult to grasp the workflow and design choices. Undefined terms (e.g., presets, functional descriptors) exacerbate this issue.


**Suggestions for Improvement**

   - Clarify the system architecture and pipeline flow. Explicitly describe the model components, how CoT reasoning integrates with prompt interpretation, and the step-by-step process from user prompt to ROC file generation.

   - Rework Figure 2 and 3 for clarity. Simplify the diagram, clearly explain the transitions, define all terms, and distinguish between dataset generation, training, reasoning, and output stages.

   - Define key terms upfront, particularly "preset" (used extensively but only vaguely implied to be parameter bundles in Lightroom). Also explain metrics like PQ and SC within the main text rather than relegating them to appendices. The Lightroom protocol in the main text only adds clutter, move it to the Appendix.

   - Consider placing greater emphasis on the GRPO-R adaptation, particularly if the policy optimization introduces technical novelties beyond applying known GRPO techniques.

**Typos**

   - Line 66: "pipelines pipelines" — duplicated word.

   - Figure 2 (Stage 3, Initial CoT annotations): "configaration" — typo.

   - Figure 3 caption: "two-stag" — incomplete word.

   - Figure 10 caption: Answer refers to image in Figure 11, not Figure 10 — mixed up.

---

> ### Author Rebuttal · Authors · 2025-07-30
>
> We sincerely appreciate your feedback and recognition of our comprehensive evaluation, professional retouching results, the contribution of the MMArt dataset, and the potential of MLLM-based artistic agent applications. Please find our detailed responses to the review comments below.
>
> **Q1: Contributions of this work (infrastructure and GRPO-R).**
> > As you mentioned, one of our key contributions is the development of a robust and scalable infrastructure, comprising a data-synthesis pipeline, a fine-tuning framework, and an Agent-to-Lightroom protocol that effectively transform MLLMs into artistic agents. Additionally, we introduce GRPO-R, a tailored RL method for tool-integrated retouching tasks. In contrast to standard GRPO—which applies to mathematical tasks with a single, easily verifiable correct answer—retouching involves multi-dimensional complexity: **(1)** the model must predict multiple tools and their parameters, and **(2)** diverse parameter combinations may yield visually similar results. Designing effective reward signals in such a setting remains an open challenge. GRPO-R addresses this by incorporating two reward components: `retouching operation accuracy` for parameter prediction and `perceptual quality` for visual fidelity. This specific reward design enables scalable reinforcement learning, thereby enhancing the agent's reasoning ability, tool proficiency, and generalization. We believe this system design and tailored RL optimization can provide inspiration for the development of domain-specific intelligent agents, beyond the demonstrated success in Lightroom.
>
> **Q2: Scope limitations.**
> > This work focuses on a natural language-driven retouching task, which differs from generative editing. While generative editing involves creating new content or modifying image structures for creative purposes, photo retouching aims to enhance the original image by adjusting visual attributes—such as color, tone, and sharpness—without introducing external elements. Additionally, our method can be extended to support generative editing operations through the proposed data pipeline, which enables generating paired artist-edited data aligned with specific AI operations (e.g., addition, deletion), allowing the model to learn these capabilities.
>
> **Q3: Misleading pixel-level comparisons.**
> > To ensure a fair and comprehensive comparison, we evaluate JarvisArt against both traditional photo retouching methods and instruction-driven generative editing models. Following prior studies [1][2][3], we employ L1 and L2 metrics to assess pixel-level content preservation, and SC, PQ, and O metrics to measure instruction adherence. As shown in Tables 1 and 4 of the manuscript, our method achieves pixel-level fidelity comparable to traditional retouching while significantly outperforming generative models. Moreover, the SC and PQ results indicate competitive instruction-following performance relative to generative editing approaches. Through these multidimensional evaluations, we aim to highlight that JarvisArt integrates both the pixel fidelity of traditional retouching and the instruction-following capabilities of generative models, effectively achieving the strengths of both approaches.
>
> **Q4: Lack of sufficient robustness of dataset curation.**
> > To guarantee data quality, we employ a two-step verification process: **(1)** The Qwen2.5-VL-72B model evaluates the alignment between instructions and reasoning via multimodal input prompts. **(2)** Manual sampling is conducted for additional validation. This process will be clarified in the manuscript. Aggregating outputs from MLLMs and applying consensus-based filtering can enhance data quality, but it may incur higher computational and resource costs. To further improve sample quality, we currently regenerate instruction and chain-of-thought (CoT) annotations using Gemini-2.5 Pro. We plan to release this version as open-source.
>
> **Q5: Explanation of ROC file generation.**
> > As described in Section 3.2 of the manuscript, the retouching operation configuration (ROC) file and corresponding before/after-retouch images are prepared in advance during the instruction and chain-of-thought (CoT) reasoning generation process. After generating the instruction and reasoning text, the ROC file is inserted within the `<answer>` tag to produce an R1-style QA sample. The MLLM is not required to regenerate the ROC file, as its parameters are summarized manually by professional photographers.
>
> **Q6: Versatility of the MMArt dataset across research fields.**
> > The MMArt dataset may provide significant value for research in three key areas: **(1)** unified multimodal understanding and generation, **(2)** text-to-image generation, and **(3)** image-to-image generation. Specifically, the dataset includes diverse user instructions, high-quality photographic images, and chain-of-thought (CoT) annotations, all of which support the development and evaluation of models across these tasks.
>
> **Q7: Detail of technical terms.**
> > In Lightroom, a preset refers to a pre-saved retouching operation configuration (ROC) file, typically created by photographers or artists. It encapsulates stylistic parameters tailored for specific visual aesthetics—for example, a Japanese-style fresh look. Applying a preset enables users to efficiently reproduce a desired effect across multiple images. However, presets are often context-specific and may not yield satisfactory results for all image types. As outlined in Section 3.2, Stage I of the manuscript, we enhanced the accuracy of preset recommendations for the Qwen2.5-VL-72B model by summarizing each preset’s functional description. For example, the preset `{"id": "PERSON-G20", "name": "Japanese Fresh Style"}` is described as follows: `“This preset reduces overall saturation (-5) and warms tones (yellow/orange saturation -43/-42), emphasizes green brightness (+50), and enhances blue saturation (+31) to create a refreshing visual effect. It is suitable for outdoor portraits and Japanese-style compositions under natural lighting.”` These terms will be detailed in the manuscript.
>
> **Q8: Typo errors and paper presentation.**
> > All typo errors have been corrected in the revised manuscript. Additionally, we have simplified the diagram, clarified the transitions, defined all relevant terms, and explicitly distinguished the stages of dataset generation, training, reasoning, and output. PQ and SC metrics will be explained in the main text, with Lightroom protocol details included in the Appendix.
>
>
> ---
> References:
>
> [1] MagicBrush: A Manually Annotated Dataset for Instruction-Guided Image Editing. NeurIPS 2023 Datasets and Benchmarks.
>
> [2] Emu Edit: Precise Image Editing via Recognition and Generation Tasks. CVPR 2024.
>
> [3] In-Context Edit: Enabling Instructional Image Editing with In-Context Generation in Large Scale Diffusion Transformer. Arxiv 2025, Github star 1.9k.

---

> > ### Author Response · Authors · 2025-08-04
> >
> > Dear Reviewer NfhL,
> >
> > We sincerely appreciate your insights and welcome any feedback on our rebuttal.
> >
> > Given the computational demands of the agentic tool-integrated retouching task (e.g., MLLM processing and Lightroom invocation), we would be grateful if you could let us know whether our response has adequately addressed your concerns. Your feedback will help us ensure the timely delivery of results.
> >
> > Thank you very much for your time and thoughtful input.
> >
> > Best regards,
> > The Authors

---

> > ### Comment · Reviewer_NfhL · 2025-08-05
> >
> > Dear authors,
> >
> > Thank you for your response.
> >
> > As shown in Table 2: *Quantitative comparison on both scene-level and region-level*, in the answer to reviewer 3pvW, Google Photos outperforms your method. Can you explain the advantage of using your method over Lightroom Auto or Google Photos?

---

> > > ### Author Response · Authors · 2025-08-05
> > >
> > > Dear Reviewer NfhL,
> > >
> > > Thank you for your valuable comments and for raising this important point.
> > >
> > > **Q1: Analysis of lower pixel-level metric performance compared to Google Photos.**
> > >
> > > > We need to clarify the fundamental distinction between JarvisArt and commercial auto-enhancement tools. Commercial tools, such as Google Photos or Lightroom Auto, typically apply fixed, generic enhancement algorithms—adjusting parameters like exposure and contrast—without awareness of the user’s specific creative intent. In contrast, JarvisArt functions as an intelligent agent that interprets the user’s intent and delivers professional‑level edits tailored to that vision.
> > >
> > > > This distinction is critical when evaluating pixel-based metrics like PSNR. For subjective, instruction-driven retouching tasks, pixel-level fidelity metrics serve only as references, because the gt image is just one of many potentially creative outcomes. An aesthetically superior edit that perfectly matches a user's intent may score a low PSNR simply because it creatively diverges from the single reference image.
> > >
> > > > Recognizing this limitation, we conduct a more comprehensive evaluation:
> > > > - **(1) MLLM-based metrics**. We further evaluated outputs using **SC**, **PQ**, and **O** scores, which measure alignment with user instructions, aesthetic preferences, and contextual coherence. As shown in Table 1, JarvisArt surpasses commercial tools across all three metrics, demonstrating superior instruction-following and aesthetic quality.
> > >
> > > **Table 1: Quantitative comparison on both scene-level and region-level**
> > > | **Method**            | **Instruction** | **Scene-level**  |  |  | **Region-level** |  |  |
> > > |-------------------|:--------:|:--------:|:--------:|:--------:|:--------:|:--------:|:--------:|
> > > |                   |          | $\text{SC}$ ↑     | $\text{PQ}$  ↑     | $\text{O}$ ↑      | $\text{SC}^{RC}$ ↑     | $\text{PQ}^{RC}$ ↑     | $\text{O}^{RC}$ ↑      |
> > > | Lightroom Auto    | ❌       | 3.85        | 9.39       | 6.01        | 3.79        | 9.32       | 5.94       |
> > > | Google Photos      | ❌       | 3.91       | 9.54        | 6.11        | 3.87        | 9.27        | 5.98        |
> > > | **JarvisArt**     | ✅       | **7.53** | **9.82** | **8.52** | **8.08** | **9.39** | **8.69** |
> > >
> > >
> > > > - **(2) User Study**. We also conduct a user preference study (e.g., Sec. 4.2.2 of the manuscript) with 20 participants evaluating Lightroom Auto, Google Photos, and JarvisArt. Participants rated image-text consistency and aesthetic quality on a five-point ordinal scale `(2 = worst, 4 = poor, 6 = fair, 8 = good, 10 = excellent)`. As shown in Table 2, JarvisArt achieved the highest average scores across both criteria, consistently producing edits preferred by users.
> > >
> > > **Table 2: User Preference study results**
> > > | **Method**      | **Image‑text Consistency (↑)** | **Aesthetic Quality (↑)** |
> > > |-----------------|:---------------------:|:----------------------:|
> > > | Lightroom Auto  |         5.14           |          6.21           |
> > > | Google Photos   |         5.42           |          6.34          |
> > > | **JarvisArt**   |         **8.64**      |         **7.82**        |
> > >
> > >
> > > **Q2: Comparative advantages of JarvisArt vs. commercial tools.**
> > > > JarvisArt’s advantages can be summarized in three aspects:
> > > >
> > > > - **(1) Instruction-driven creative retouching**: Unlike generic auto-enhancement tools, JarvisArt adapts to subjective user intent, supporting free-form instructions through text and bounding boxes. For example, a user may request a `“dreamy haze style”` or a `“moody, cinematic look.”` As illustrated in Figures 14 and 15 of the manuscript, JarvisArt achieves stylistic fidelity that conventional auto-enhancement tools cannot, accurately realizing the user’s creative vision.
> > > >
> > > > - **(2) Proficiency in masking tools for artistic effects.** Unlike commercial tools limited to global adjustments, JarvisArt integrates six advanced masking tools—Object, Portrait, Linear, Radial, Color, and Luminance—to precisely sculpt light, shadow, and local details. This capability supports nuanced artistic effects, such as emphasizing the subject or creating localized dramatic lighting, and exceeds the expressive potential of commercial auto‑retouching methods.
> > > >
> > > > - **(3) Transparent and interactive professional workflow.** Unlike the black-box nature of commercial tools, JarvisArt reveals its decision-making process (e.g., `“first adjust white balance for a cooler tone, then use the local object masking tool to highlight the subject”`), providing full transparency to the user. It also supports iterative refinement, consistent with professional retouching practices.
> > >
> > > Best regards, The Authors

---

> > > ### Author Response · Authors · 2025-08-06
> > >
> > > Dear reviewer NfhL,
> > >
> > > We sincerely appreciate your insights and welcome any feedback on our response.
> > >
> > > Given that the rebuttal deadline is approaching, we would be grateful if you could let us know whether our response has adequately addressed your concerns. Your feedback will help us ensure the timely delivery of results.
> > >
> > > Thank you again for your time and thoughtful review.
> > >
> > > Sincerely, All authors

---

> > > > ### Comment · Reviewer_NfhL · 2025-08-06
> > > >
> > > > Your response has adequately addressed my concerns.
> > > >
> > > > Thank you

---

> > > > > ### Author Response · Authors · 2025-08-07
> > > > >
> > > > > Dear reviewer NfhL,
> > > > >
> > > > > Thank you for your thoughtful participation and contributions to the discussion. We will incorporate your suggestions into the final version.
> > > > >
> > > > > If you feel that there are no further concerns, we would like to kindly remind you to submit your final review and ratings. We would be sincerely grateful for your support!
> > > > >
> > > > > Of course, if any new questions arise, please feel free to reach out—we’d be happy to clarify. If you have already submitted your final review, please kindly disregard this message.
> > > > >
> > > > > Sincerely, All authors

---

> > > > > > ### Author Response · Authors · 2025-08-08
> > > > > > **The deadline is less than 24 hours away, and final scoring has not yet been completed.**
> > > > > >
> > > > > > Dear reviewer NfhL,
> > > > > >
> > > > > > Thank you for your thoughtful participation and contributions to the discussion.
> > > > > >
> > > > > > If you feel that there are no further concerns, we would like to kindly remind you to submit your **final review and ratings** via the original **Edit button**. Should you be willing to increase your rating, we would sincerely appreciate your support.
> > > > > >
> > > > > > Of course, if any new questions arise, please feel free to reach out—we’d be happy to clarify. If you have already submitted your final review, please kindly disregard this message.
> > > > > >
> > > > > > Sincerely, All authors

---

> > > > > > > ### Author Response · Authors · 2025-08-09
> > > > > > > **The deadline is less than 3 hours away, and final scoring has not yet been completed.**
> > > > > > >
> > > > > > > Dear reviewer NfhL,
> > > > > > >
> > > > > > > Thank you for your thoughtful participation and contributions to the discussion.
> > > > > > >
> > > > > > > The rebuttal deadline is less than 3 hours away, we would like to kindly remind you to submit your **final review and ratings via the original Edit button**. Should you be willing to **increase your rating**, we would sincerely appreciate your support.
> > > > > > >
> > > > > > > Of course, if any new questions arise, please feel free to reach out—we’d be happy to clarify. If you have already submitted your final review, please kindly disregard this message.
> > > > > > >
> > > > > > > Sincerely, All authors

---

### Note · Authors · 2025-08-15

Dear PCs, ACs, SACs and Reviewers,

We sincerely appreciate your time and efforts in reviewing our paper. We are glad to find that reviewers recognized the following merits of our work:

> - **Significant potential applications and clear motivation [NfhL, 3pvW, VQZs, xmNH]:** JarvisArt aims to provide professional-level editing with user-friendly interactivity for all users. It focuses on high-resolution, non-destructive photo retouching, meeting a real-world need distinct from generative models, with potential for broad application in artistic design.
>
> - **Valuable MMArt dataset [NfhL, 3pvW, VQZs, xmNH]:** The MMArt dataset is valuable to the community, offering diverse user instructions, high-quality images, and chain-of-thought (CoT) annotations to aid model development and evaluation across fields.
>
> - **Impressive performance [NfhL, 3pvW, xmNH]:** Experimental results show that JarvisArt excels in instruction-following and aesthetic quality, delivering realistic, professional edits with superior content fidelity compared to competing methods.

We also thank all reviewers for their constructive feedback, which has further improved our paper. Below, we summarize the major revisions made in response to the reviewers' suggestions:

> - **Complete contributions [NfhL, VQZs]**: We clarify the infrastructure contributions, including the data-synthesis pipeline, fine-tuning framework, GRPO-R RL method for tool-integrated retouching, and A2L protocol. These components transform MLLMs into artistic agents, recognized by reviewers 3pvW and xmN. After the rebuttal, we addressed NfhL's concerns.
>
> - **Additional quantitative comparisons [NfhL, 3pvW]:** We compare commercial retouching tools using pixel-level and MLLM-based metrics, alongside user study. JarvisArt stands out for its: (1) instruction-driven design, (2) proficiency in artistic masking tools, and (3) transparent, interactive professional workflow.
>
> - **Discussion of potential intermediate errors [VQZs]:** We analyze intermediate errors from data generation and inference. A two-stage verification ensures accuracy during generation, while inference errors accumulate due to the lack of immediate feedback. We propose stepwise rewards in RL training to address this, with ongoing work focused on optimization.

We hope our pointwise responses above can clarify all reviewers' confusion and address the raised concerns. We thank all reviewers' efforts and time again.

Best, Authors

---

### Decision · Program_Chairs · 2025-09-17

**Decision:**

Accept (poster)

**Comment:**

This submission presents JarvisArt, a multimodal LLM-based agent for professional photo retouching. It is supported by the MMArt  and a two-stage training pipeline (SFT + GRPO-R). The reviewers praised its strong qualitative results, valuable dataset, comprehensive evaluation. I was most excited to see a practical integration with Lightroom.

The main concerns centered on modest methodological novelty, the narrow scope restricted to Lightroom-based edits, potential limitations in dataset robustness, and questions about fair comparisons against pixel-level generative models.

Overall, while not conceptually groundbreaking, the paper makes a meaningful contribution to the community.